# Dynamin1 long- and short-tail isoforms exploit distinct recruitment and spatial patterns to form endocytic nanoclusters

Anmin Jiang[1], Kye Kudo [1], Rachel S. Gormal [1], Sevannah Ellis [1], Sikao Guo[2], Tristan P. Wallis [1], Shanley F. Longfield[1], Phillip J. Robinson [3,6], Margaret E. Johnson [2,6], Merja Joensuu [1,5] ✉ & Frédéric A. Meunier [1,4,6] ✉

Endocytosis requires a coordinated framework of molecular interactions that ultimately lead to the fission of nascent endocytic structures. How cytosolic proteins such as dynamin concentrate at discrete sites that are sparsely distributed across the plasma membrane remains poorly understood. Two dynamin-1 major splice variants differ by the length of their C-terminal proline-rich region (short-tail and long-tail). Using sptPALM in PC12 cells, neurons and MEF cells, we demonstrate that short-tail dynamin-1 isoforms ab and bb display an activity-dependent recruitment to the membrane, promptly followed by their concentration into nanoclusters. These nanoclusters are sensitive to both Calcineurin and dynamin GTPase inhibitors, and are larger, denser, and more numerous than that of long-tail isoform aa. Spatiotemporal modelling confirms that dynamin-1 isoforms perform distinct search patterns and undergo dimensional reduction to generate endocytic nanoclusters, with short-tail isoforms more robustly exploiting lateral trapping in the generation of nanoclusters compared to the long-tail isoform.

Endocytosis is a critically important cellular process that allows cells to internalize part of their membrane, including receptors and their bound ligands, through the generation of endocytic vesicles[1]. Clathrin-mediated endocytosis (CME), which is by far the best characterized endocytic pathway, relies on sequential molecular interactions that are initiated by the recruitment of coat proteins from the cytosol into clusters on the inner leaflet of the plasma membrane. This coat assembly promotes further cargo recruitment which, in turn, induces membrane curvature to form invaginations known as clathrin-coated pits (CCPs)[2]. A previous study has reported the recruitment dynamics of a broad range of proteins of the endocytic machinery that are involved in CME with a temporal imaging resolution of 2s[3]. The scission of nascent endocytic structures relies on the constriction of the neck of the membrane invaginations, which occurs following the recruitment of the cytosolic protein called dynamin. In addition to coat proteins and dynamin1 (Dyn1), a plethora of other cytosolic proteins are recruited from the cytosol and sequentially assemble on endocytic sites[1,3,4]. These proteins have partially overlapping functions that define the steps of endocytosis. An outstanding question is how these cytosolic proteins are recruited in a timely fashion to discrete areas that are sparsely distributed across the plasma membrane.

The dynamin family of GTPase proteins are archetypical endocytic proteins recruited from the cytosol to the neck of nascent endosomes to promote membrane fission[5,6], which they mediate through their mechano-chemical enzymatic activity[7]. There are 3 dynamin genes, dynamin 1 (Dyn1), dynamin 2 (Dyn2) and dynamin 3 (Dyn3). Dyn1 is

[1]Clem Jones Centre for Ageing Dementia Research, Queensland Brain Institute, The University of Queensland, Brisbane, QLD 4072, Australia. [2]Department of Biophysics, Johns Hopkins University, 3400 N Charles St, Baltimore, MD 21218, USA. [3]Cell Signalling Unit, Children's Medical Research Institute, The University of Sydney, Sydney, NSW 2145, Australia. [4]School of Biomedical Sciences, The University of Queensland, Brisbane, QLD 4072, Australia. [5]Present address: Australian Institute for Bioengineering and Nanotechnology, The University of Queensland, Brisbane, QLD 4072, Australia. [6]These authors contributed equally: Phillip J. Robinson, Margaret E. Johnson, Frédéric A. Meunier. ✉e-mail: m.joensuu@uq.edu.au; f.meunier@uq.edu.au

neuronal specific, Dyn2 is ubiquitously expressed, and Dynamin 3 is expressed in a subset of cells including the testis[8,9]. Dyn1, which is predominantly expressed in the brain[8], has eight isoforms, each consisting of one of two different middle domains (either a or b) together with one of four C-terminal proline-rich tail regions (either a, b, c or d)[10]. The middle domain is essential for dynamin tetramerization and self-assembly into a helical structure[11], whereas the C-terminal region is vital for the recruitment of various binding partners involved in mediating endocytosis[12].

Dyn1 is known to be involved in CME, and has been reported to promote the assembly and maturation of CCP[13]. A recent study in neurons demonstrated that a fraction of long-tail Dyn1 isoforms (Dyn1xa, x representing either middle domain) is already present on the plasma membrane as pre-assembled clusters, enabling ultrafast endocytosis to occur[14]. Whether pre-clustering of other Dyn1 isoforms is also necessary for other forms of endocytosis remains unknown. Furthermore, it is unclear how pre-clustered Dyn1 facilitates endocytosis and how it interplays with Dyn1 that is newly recruited to the plasma membrane in response to stimulation.

Dimensional reduction is a key fundamental phenomenon that allows molecules to critically increase their concentration in space and time by moving from the 3D solution volume to the 2D membrane surface. This reduces their search area–accelerating search times and enhancing binding probabilities[15]. Pre-clustered Dyn1 could therefore represent a thermodynamically favorable mechanism that minimizes search effort, thus providing nanoscale recruitment hubs onto which newly recruited dynamin can be trapped to form larger and more numerous endocytic clusters. In this view, freshly recruited Dyn1 could therefore use pre-assembled clusters to fast track their 2D search pattern for future rounds of endocytosis.

The two major C-terminal Dyn1 splice variants in neurons are the long- and short-tail Dyn1xa and Dyn1xb isoforms, respectively[16]. The C-terminal proline-rich region of Dyn1 has a variety of binding sites for interacting partners which are essential for its recruitment to endocytic sites on the plasma membrane[17]. Dyn1 tail phosphorylation, mediated by cyclin-dependent kinase 5 (Cdk5) or Glycogen synthase kinase 3 (GSK3), and dephosphorylation, mediated by calcineurin, has been shown to be associated with the regulation of CME and activity-dependent bulk endocytosis[18–20], in response to calcium ($Ca^{2+}$)[21]. Among these enzymes, calcineurin has been reported to have a unique binding site on the Dyn1xb short-tail isoforms that is not present on the Dyn1xa long-tail isoforms[22]. Whether this unique interaction between calcineurin and Dyn1xb short-tail isoforms contributes to their recruitment to endocytic sites is unknown.

Here, we investigated the recruitment mechanisms of three structurally distinct Dyn1 isoforms (Dyn1aa, Dyn1ab and Dyn1bb) in response to stimulation. We utilized total internal reflection fluorescence microscopy (TIRF) combined with single-particle tracking photoactivated localization microscopy (sptPALM) in live neurosecretory pheochromocytoma (PC12) cells, Dyn1,2 conditional double knockout (DKO) mouse embryonic fibroblasts (MEFs) and mouse primary hippocampal neurons to quantify the activity-dependent recruitment patterns and nanoscale dynamics of individual Dyn1 molecules on the plasma membrane. Our results demonstrate that both short-tail Dyn1ab and Dyn1bb isoforms, have a similar bi-phasic recruitment pattern following secretagogue stimulation. An initial phase of recruitment occurred across the entire plasma membrane surface. This was followed by their lateral diffusion and trapping into discrete microscale areas on the plasma membrane. Such areas exhibit a high density of nanoscale clusters in which individual Dyn1 molecules are actively confined. In contrast, Dyn1aa exhibited a single-phase recruitment: following stimulation Dyn1aa is recruited to the PM similar to the initial recruitment step of short-tail isoforms, albeit at a lesser extent, and does not appear to undergo a subsequent concentration/clustering phase. The mobility of single Dyn1ab and Dyn1bb

molecules decreased significantly following stimulation, suggesting that these Dyn1 isoforms undergo activity-dependent lateral trapping possibly by binding to other effectors on the membrane. Conversely, Dyn1aa mobility was already low at rest and did not decrease further upon stimulation. We observed individual laterally diffusing Dyn1 molecules as they become actively-confined within nanoscale clusters –an event which occurred at a significantly higher rate for Dyn1bb and Dyn1ab isoforms. Together, these results provide further evidence that short-tail Dyn1 isoforms in particular exploit lateral trapping. Pharmacological inhibition of Dyn1 kinases and phosphatases, using staurosporine (Sta) and okadaic acid (OA) treatments respectively, showed that the second recruitment phase whereby Dyn1bb forms clusters on the plasma membrane was phosphorylation-dependent. These results were further supported by calcineurin inhibition through Cyclosporin-A (CysA) treatment, which also blocked the second phase of Dyn1ab and Dyn1bb recruitment. Further, Cyclosporin-A treatment had no effect on the recruitment of Dyn1aa nor on its nanoscale mobility. Our results indicate that different Dyn1 isoforms are recruited to the plasma membrane via different mechanisms, and that dephosphorylation by calcineurin plays a vital role in the lateral recruitment of Dyn1ab and Dyn1bb into nanoclusters. Our findings challenge the long-standing view that dynamin is recruited directly from the cytosol to the neck of nascent vesicles. We establish that Dyn1 short-tail isoforms are recruited *en masse* to the plasma membrane in an activity-dependent manner where they then undergo lateral trapping into endocytic structures. We found that this process is critically dependent on dephosphorylation of the proline-rich region of the short-tail isoforms. This exploitation of the plasma membrane 2D search pattern is far less pronounced for Dyn1aa, which instead undergoes a single-phase recruitment to the plasma membrane in response to secretagogue stimulation.

## Results

### Short- and long-tail Dyn1 isoforms exhibit distinct recruitment paradigms

To investigate the activity-dependent recruitment of Dyn1 short- and long-tail isoforms to the plasma membrane, green fluorescent protein (GFP)-tagged Dyn1ab, Dyn1bb and Dyn1aa isoforms were overexpressed in PC12 cells. Cells were imaged via TIRF microscopy and stimulated using strong secretagogue barium ($Ba^{2+}$) to promote exocytosis and subsequent compensatory endocytosis[23–26].

To analyze their recruitment kinetics, the mean fluorescence intensity (FI) on the plasma membrane was quantified using time-lapse imaging. Immediately following $Ba^{2+}$ secretagogue stimulation, a uniform increase in FI throughout the plasma membrane was observed for both Dyn1bb-GFP and Dyn1ab-GFP, which we termed "early phase recruitment" (Fig. 1a, b). Following this initial phase, Dyn1bb-GFP and Dyn1ab-GFP appeared to accumulate within discrete high FI areas (HFA) on the plasma membrane that continued to increase in FI (Fig. 1a, b). We termed this second phase the "concentration/clustering phase". The average fold change in FI of the entire plasma membrane elicited by stimulation was significantly higher for the short-tail isoforms Dyn1bb-GFP and Dyn1ab-GFP compared to that of long-tail isoform Dyn1aa-GFP (Supplementary Fig. 1), suggesting strong plasma membrane recruitment of the short-tail isoforms and a relatively moderate recruitment of the long-tail isoform (Fig. 1a–d). To further analyze the bi-phasic recruitment pattern exhibited by the short-tail isoforms, small regions of interest in both high (HFA) and low FI areas (LFA) (Fig. 1a, b) were selected, from which the kinetics, FI fold change, and time taken for FI to peak was quantified. While intensity increased with similar kinetics in both the LFA and HFA during the early recruitment phase, the LFA reached maximum intensity at an earlier timepoint than that of the HFA, which continued to rise (Fig. 1e, f). Statistical analysis showed that the average time to peak

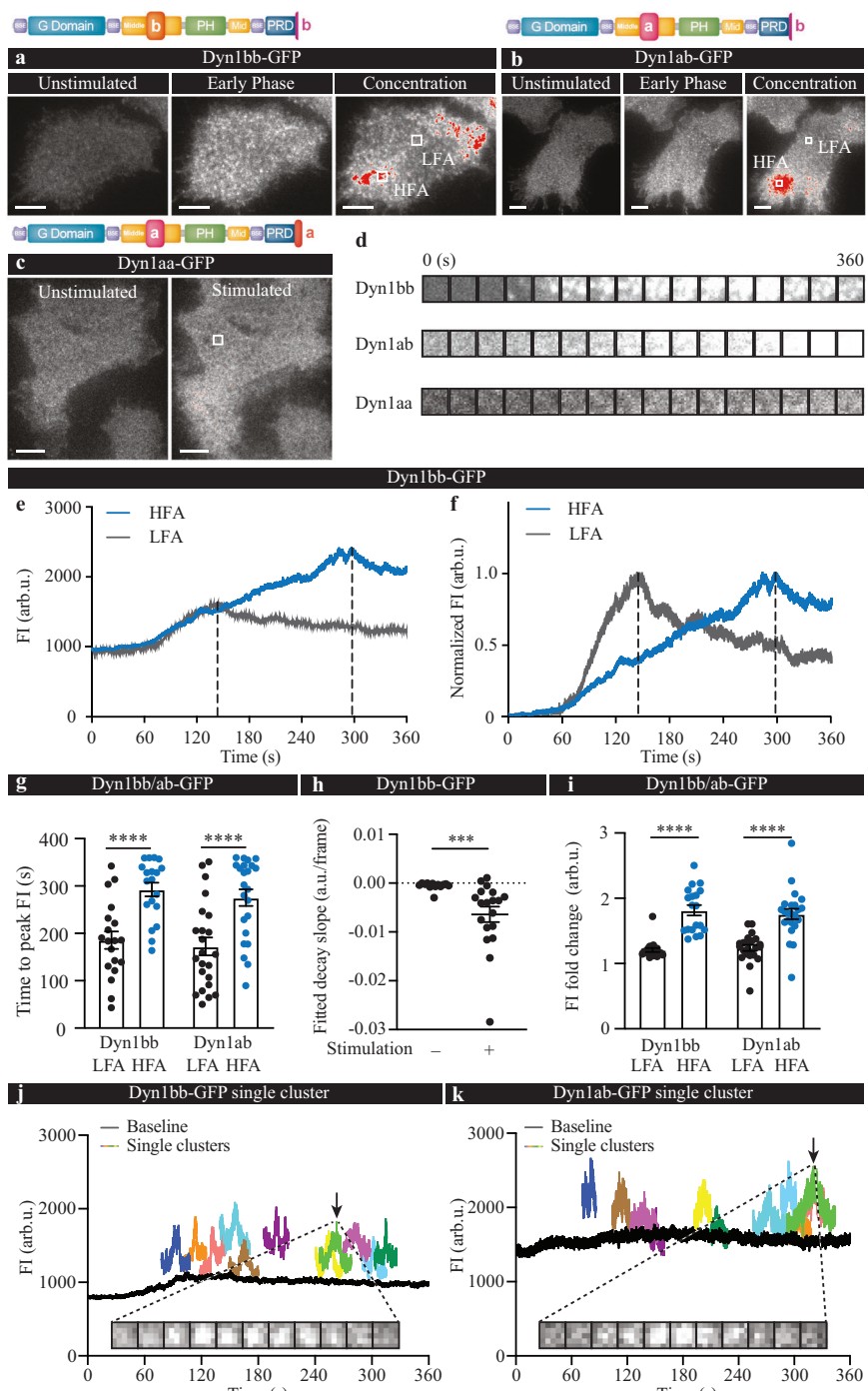

**Fig. 1 | Bi-phasic recruitment of Dyn1ab and Dyn1bb and single-phase recruitment of Dyn1aa.** PC12 cells were transfected with either Dyn1bb-GFP, Dyn1ab-GFP or Dyn1aa-GFP (domain structures illustrated) and imaged at 50 Hz in either unstimulated or stimulated (2 mM Ba$^{2+}$) conditions. Left panels show representative TIRF images of (**a**) Dyn1bb-GFP, (**b**) Dyn1ab-GFP, and (**c**) Dyn1aa-GFP before (unstimulated), and following stimulation (360 s), with early and concentration phases detected for the short-tail isoforms, as indicated. Scale bars, 5 μm. White box, representative region of interest (ROI) within the LFA and the HFA. Red areas represent the HFA for this frame. **d** TIRF time-lapse images at different time points (from 0–360 s) of indicated ROIs (white box outlines) in stimulated conditions. **e** A representative and (**f**) normalized trace of Dyn1bb-GFP FI (arbitrary units; arb.u.) taken from the high FI area (HFA; blue) and low FI area (LFA; dark gray) indicated in (**a**) upon stimulation (peak intensity points indicated by black dotted lines). **g** Time to peak was analyzed for the HFA and the LFA for both Dyn1bb-GFP and Dyn1ab-GFP (in seconds, s). **h** FI decay slope of the Dyn1bb-GFP LFA is significantly steeper than in the photo-bleached condition. **i** The FI fold change is significantly higher in the HFA than in the LFA for both Dyn1bb-GFP and Dyn1ab-GFP. **j**, **k** Representative FI curves of single clusters of Dyn1bb-GFP and Dyn1ab-GFP at different time points; arrows indicate the timepoint at which the below representative TIRF time-lapse images for a single cluster of Dyn1bb-GFP or Dyn1ab-GFP occurred. (For **g** and **i**, $n = 19$ cells for Dyn1bb, and $n = 23$ cells for Dyn1ab from 4 independent experiments; non-normally distributed paired data was analyzed with two-tailed Wilcoxon test, $p < 0.0001$; for **h**, $n = 13$ for control and $n = 19$ cells for stimulation, two-tailed, non-normally distributed unpaired data was analyzed with Mann–Whitney test, $p = 0.0003$; ***$p < 0.001$, ****$p < 0.0001$; Mean ± SEM are plotted). Source data are provided as a Source Data file.

following stimulation was ~280 s for the HFA and ~180 s for the LFA following stimulation (Fig. 1g).

Following the peak in fluorescent intensity, the LFA demonstrated a fluorescent decay. We fitted this fluorescence decay slope and found that it decayed at a much faster rate than the rate expected simply due to that of photobleaching (Fig. 1h). This indicates that following plasma membrane recruitment, short-tail Dyn1 molecules gradually diffuse from the LFA. Due to the concomitancy of the reduction of intensity in the LFA with the increase in intensity of the HFA during the later phase of clustering, the increased binding capacity of the HFA may act as a trap for diffusing Dyn1ab and Dyn1bb molecules. Indeed, the FI fold change was significantly higher in the HFA than in the LFA for both Dyn1ab and Dyn1bb (Fig. 1i), suggesting an uneven recruitment between these two areas. Further examination of the HFA revealed that it consisted of many small discrete clusters that appeared and disappeared at various times following stimulation, and had intensities that peaked at different times (Fig. 1j, k and Movie 1).

These results indicate that both Dyn1ab and Dyn1bb are recruited to high-density microscale areas on the plasma membrane through a bi-phasic mechanism. This includes an intial bulk recruitment phase to the plasma membrane, followed by a second concentration/clustering phase which may occur through the lateral movement and trapping of Dyn1 short-tailed isoforms on the plasma membrane. This bi-phasic recruitment pattern is at odds with the conventional view that dynamin molecules are directly recruited from the cytosol to the neck of endocytic sites during dynamin-dependent endocytosis[27–30]. In sharp contrast, long-tail isoform Dyn1aa-GFP exhibited a relatively uniform and modest increase in FI following stimulation. Interestingly, we could not detect the second concentration/clustering phase (Fig. 1c, d), indicating that Dyn1aa may be recruited on, or in, the immediate vicinity of endocytic sites via a moderate-level single-phase recruitment.

## Dyn1 clusters localize to sites of endocytosis

We next assessed whether the recruitment of Dyn1 corresponded to events of endocytosis by incubating PC12 cells expressing Dyn1bb-GFP, Dyn1ab-GFP or Dyn1aa-GFP with Transferrin-Alexa Fluor™ 647 (Tf-647, a known CME cargo). Following $Ba^{2+}$ stimulation, Dyn1bb-GFP and Dyn1ab-GFP co-localized strongly with Tf-647 (Supplementary Fig. 2). Long-tail isoform Dyn1aa on the other hand was co-localized with Tf-647 in both unstimulated and stimulated conditions, and had a higher level of co-localization in unstimulated condition compared to that of the short-tail isoforms (Supplementary Fig. 2). This suggests that in the absence of an evoked stimulus (at rest), long-tail isoform Dyn1aa is already present in clusters on the plasma membrane.

## Short- and long-tail Dyn1 isoforms exhibit distinct mobility patterns following recruitment to the plasma membrane

To directly assess whether lateral trapping occurs, we peformed sptPALM using Dyn1ab-mEos2, Dyn1bb-mEos2 and Dyn1aa-mEos2, which allowed us to determine the single-molecule mobilities of the long- and short-tail isoforms on the plasma membrane of live PC12 cells. Consistent with our previous results, upon $Ba^{2+}$ stimulation, a significant increase in the density of single-molecule trajectories of Dyn1ab-mEos2 and Dyn1bb-mEos2 was observed on the plasma membrane (Fig. 2a–c). Indeed, the recruitment and accumulation of single Dyn1ab-mEos2 and Dyn1bb-mEos2 molecules on the plasma membrane follows a bi-phasic pattern. In the early phase of stimulation, the short-tail isoforms are rapidly recruited to the plasma membrane (Supplementary Fig. 3a, b). However, during the second concentration phase in which the LFA decays in intensity whilst the HFA increases (100–200 s onwards, see Fig. 1e–g), the rate of molecular detections made across the entire plasma membrane plateaus (Supplementary Fig. 3a, b), therefore indicating that during the second concentration phase, the total content of Dyn1ab and Dyn1bb

on the plasma membrane remains approximately constant. These results confirm, that at the single-molecule level, Dyn1ab and Dyn1bb-mEos2 molecules are recruited from the cytosol to the plasma membrane following stimulation through a bi-phasic mechanism.

The representative super-resolved trajectory, diffusion coefficient, and average intensity maps of Dyn1bb-mEos2 (Fig. 2a) highlight the recruitment and confinement of the molecules on small discrete areas of the plasma membrane, which are suggestive of nanoclusters. Importantly, these clusters were also present in unstimulated cells albeit at a much lower density. The mobility of both Dyn1bb- and Dyn1ab-mEos2 decreased significantly following stimulation (Fig. 2b, c), confirming that a subpopulation of the Dyn1ab and Dyn1bb molecules were confined to nanoclusters following their recruitment to the plasma membrane. This was evidenced by the significant decrease in the mobile-to-immobile fraction (M/IMM) of molecules and Area Under the MSD Curve (AUC). Together with the recruitment results described in the previous section, it appears that these two short-tail isoforms share the same bi-phasic recruitment pattern, in that it leads to their lateral trapping into nanodomains on the plasma membrane following their intial recruitment to the plasma membrane.

The lack of a second recruitment phase for the long-tail isoform was confirmed using sptPALM on PC12 cells transfected with Dyn1aa-mEos2. As expected, while the number of tracked Dyn1aa-mEos2 molecules increased slightly on the plasma membrane following stimulation, their mobility, which was already low, did not decrease further (Fig. 2d). This supports the view that recruitment directly to, or nearby, discrete sites on the plasma membrane had occurred. This also suggests that Dyn1aa is already confined on the plasma membrane prior to stimulation, possibly in the form of pre-clustered assemblies that are independent of activity. This mechanism supports the conventional view that dynamin molecules are directly recruited from the cytosol to the endocytic sites with the caveat that these nanoclusters likely pre-exist on the plasma membrane as previously suggested[14]. These results suggest that pre-clustered Dyn1aa could either be used as seeds to generate productive oligomeric assemblies of dynamin at the neck of nascent endocytosis vesicles, or serve as ready-to-use molecules required for specific endocytic events such as ultra-fast endocytosis[14].

## Short- and long-tail Dyn1 isoforms exhibit distinct nanoclustering modalities

To gain a greater understanding of the dynamic nanoscale organization of dynamin isoforms upon stimulation of exo-endocytosis, such as nanocluster size and lifetime, we next investigated their nanoclustering metrics. Current clustering analysis methods primarily focus on the detection of spatial nanoclusters termed "hotspots" based on molecular density, however such approaches neglect important temporal information such as cluster lifetime[31]. To address this gap, we previously developed a cluster analysis method called NAnoscale Spatio-Temporal Indexing Clustering (NASTIC), which is able to quantify both spatial and temporal components of nanoclusters[31]. The sptPALM data for all Dyn1 isoforms, were analyzed using this tool.

A representative Dyn1bb-mEos2 cluster map is shown for both unstimulated control (Ctrl) and stimulated conditions (Supplementary Fig. 4a). The indicated nanoclusters (white dotted outline magnified in the inset) were analyzed in space and time in Supplementary Fig. 4b, highlighting its temporal clustering dynamics and how they were impacted by stimulation. The overall spatiotemporal clustering of Dyn1bb-mEos2 before and after $Ba^{2+}$ stimulation is represented in Fig. 3a–c, which shows the activity-dependent recruitment and iterative clustering of Dyn1bb-mEos2 molecules in long-lasting hotspots on the plasma membrane. Spatiotemporal cluster analysis demonstrated that before stimulation, the cluster lifetime of short-tail isoforms Dyn1ab and Dyn1bb and long-tail isoform Dyn1aa were all on average 6 s (Fig. 3d–f). Upon stimulation the lifetime of short-tail isoform

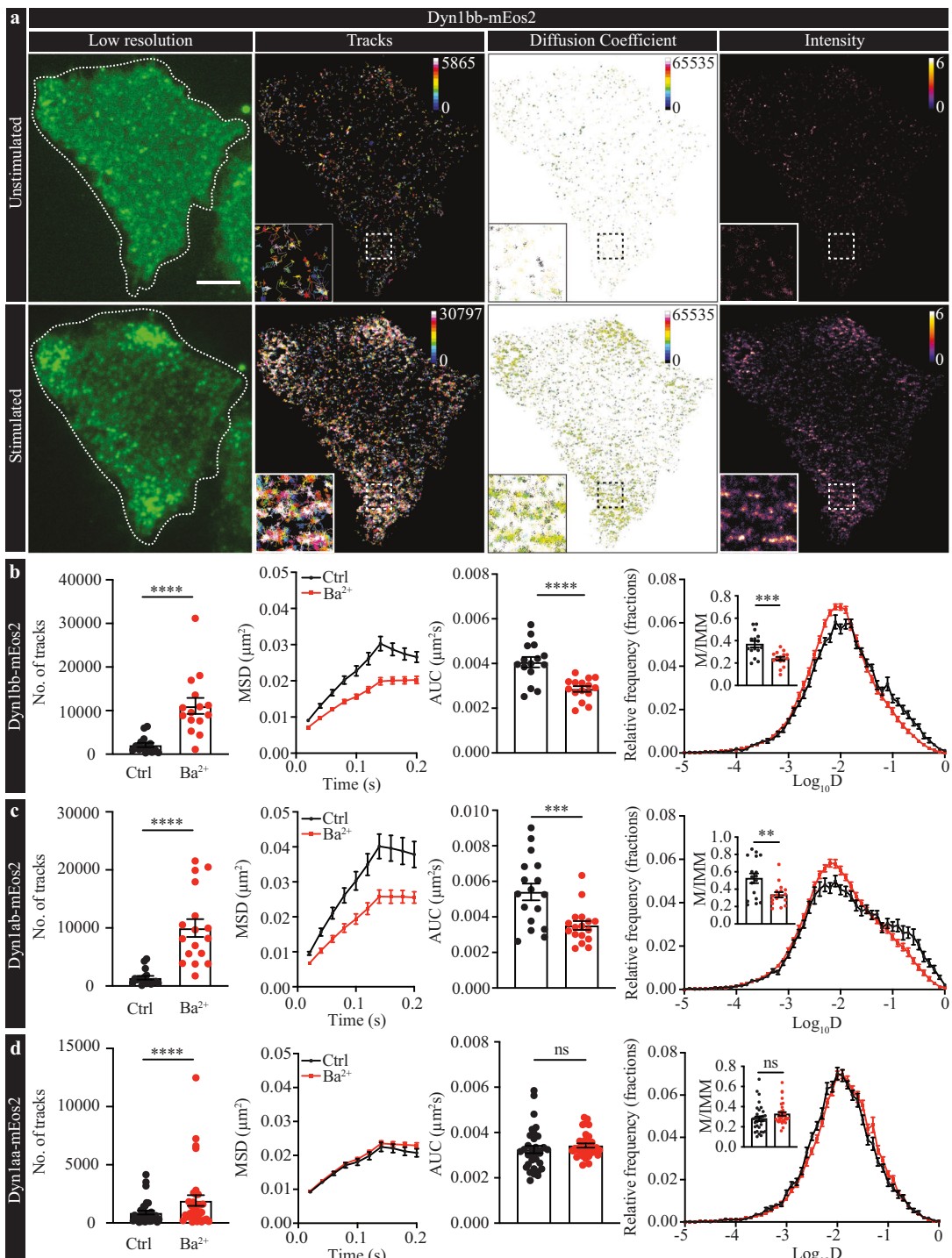

**Fig. 2 | Single molecule mobility dynamics of Dyn1bb, Dyn1ab, and Dyn1aa on the plasma membrane of PC12 cells.** PC12 cells were transfected with either Dyn1bb-mEos2, Dyn1ab-mEos2 or Dyn1aa-mEos2 and imaged at 50 Hz in both unstimulated and stimulated (2 mM $Ba^{2+}$) conditions. **a** Representative low-resolution images, sptPALM trajectory maps, diffusion coefficients and average intensities of Dyn1bb-mEos2 in both unstimulated (upper panel) and stimulated (lower panel) conditions. Dashed lines in the low-resolution image show the cell outline. Scale bars, 5 μm. Single molecule mobilities of (**b**) Dyn1bb-mEos2, (**c**) Dyn1ab-mEos2, and (**d**) Dyn1aa-mEos2 in unstimulated and stimulated PC12 cells was compared. For each isoform, from left to right are the average total number of tracks, average MSD (μm²), AUC of the MSD (μm²s), and average $Log_{10}D$ (Diffusion coefficient, D) frequency distribution (fractions) with an inset showing the mobile-to-immobile ratio (M/IMM). ($n = 15$ cells for Dyn1bb, $n = 17$ cells for Dyn1ab, $n = 33$ cells for Dyn1aa from 4 independent experiments; normally distributed paired data was analyzed with two-tailed paired Student's $t$ test for Dyn1bb and non-normally distributed paired data was analyzed with Wilcoxon test for Dyn1ab and Dyn1aa; ns non-significant $p > 0.05$, **$p < 0.01$, ***$p < 0.001$, ****$p < 0.0001$; Mean ± SEM are plotted). (**b**: No. of Tracks $p < 0.0001$, AUC $p < 0.0001$, M/IMM $p = 0.0001$; **c**: No. of Tracks $p < 0.0001$, AUC $p = 0.0005$, M/IMM $p = 0.0011$; **d**: No. of Tracks $p < 0.0001$, AUC $p = 0.1641$, M/IMM $p = 0.0626$). Source data are provided as a Source Data file.

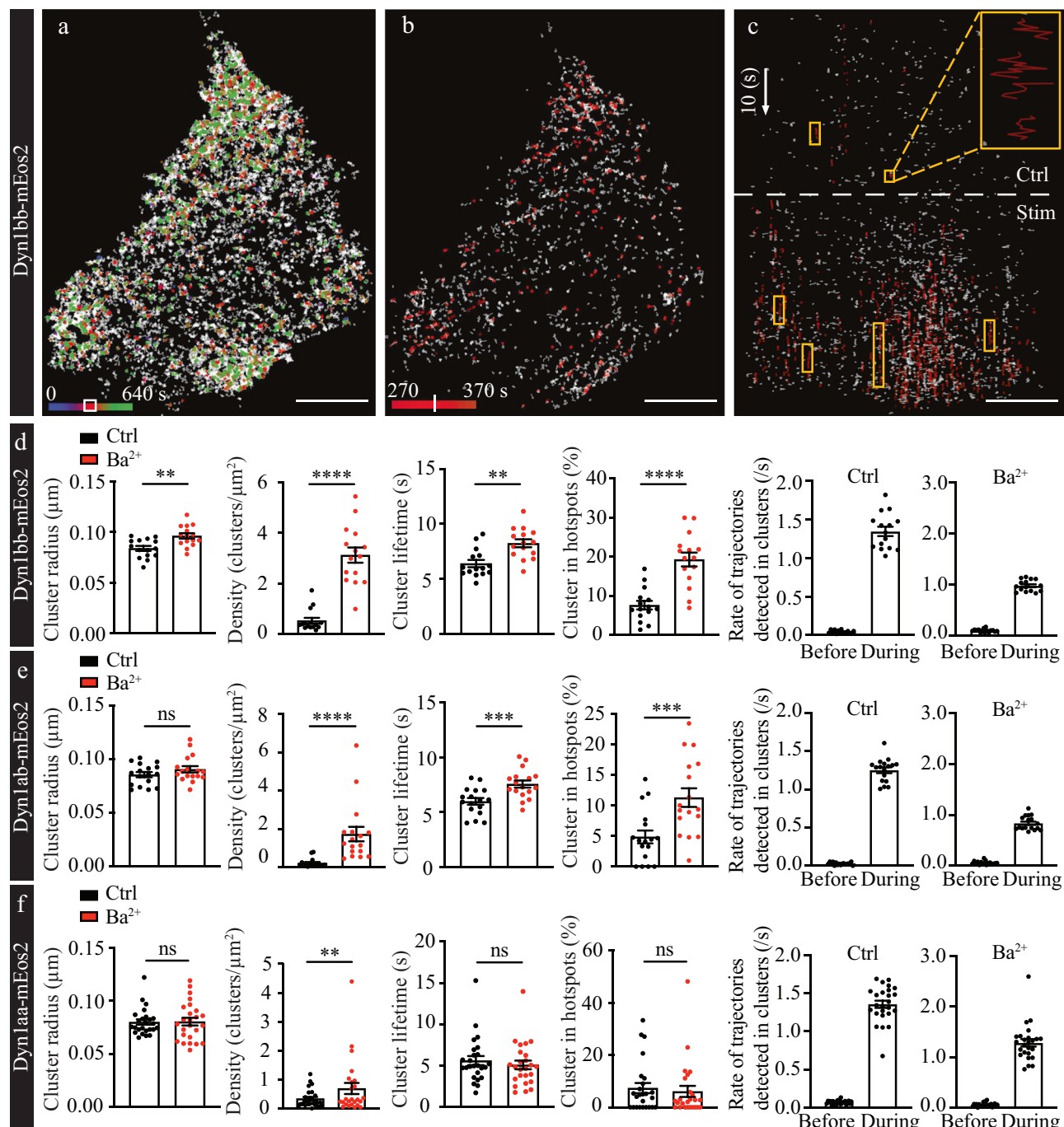

**Fig. 3 | Distinct nanoclustering of Dyn1bb, Dyn1ab, and Dyn1aa on the plasma membrane.** SptPALM data of Dyn1bb-mEos2, Dyn1ab-mEos2 and Dyn1aa-mEos2 were analyzed with the NAnoscale SpatioTemporal Indexing Clustering (NASTIC) program. **a** Representative nanoclusters of Dyn1bb-mEos2 in transfected PC12 cells prior to stimulation (Ctrl) and following stimulation (2 mM Ba$^{2+}$). White box outline on the time bar indicates the time window selected for (**b**). The white vertical line on the time bar represents the stimulation time point indicated by a horizontal white dotted line in (**c**) with Ctrl above and Stim below. **c** shows the same plot rotated so that the timescale is along the *y*-axis. Yellow box outlines show examples of nanocluster trajectories, with a magnified inset on the top right. Scale bars, 5 μm. **d**–**f** Cluster radius (μm), density of clusters (clusters/μm$^2$), cluster lifetime (s), percentage of clusters within hotspots. For each cluster, we detected the number of trajectories that occurred during the lifetime of the cluster (/s). We used the same time window to determine the number of trajectories detected at this location immediately before the formation of the cluster. These analyses were carried out before and after stimulation ($n = 15$ cells for Dyn1bb, $n = 17$ cells for Dyn1ab, $n = 25$ cells for Dyn1aa, from 3 independent experiments; normally distributed paired data was analyzed with two-tailed paired Student's *t* test and non-normally distributed paired data was analyzed with two-tailed Wilcoxon test; ns non-significant $p > 0.05$, **$p < 0.01$, ***$p < 0.001$, ****$p < 0.0001$; Mean ± SEM are plotted). (*p* values from left to right for **d**: $p = 0.0062$, $p < 0.0001$, $p = 0.0046$, $p < 0.0001$; **e**: $p = 0.1349$, $p < 0.0001$, $p = 0.0003$, $p = 0.0005$; **f**: $p = 0.8532$, $p = 0.0031$, $p = 0.2752$, $p = 0.4145$). Source data are provided as a Source Data file.

clusters increased to around 8 s (Fig. 3d, e)—an effect not detected with the long-tail isoform (Fig. 3f). Importantly, we computed the number of trajectories that occurred immediately before and during cluster formation using the lifetime of each cluster as the time window for

trajectory detection. We found that the rate of trajectory detection significantly increased during spatiotemporal clustering for all tested isoforms (Fig. 3d–f). Additionally, the cluster radius of all isoforms was ~80 nm prior stimulation (Fig. 3d–f), suggesting that both short-tail

and long-tail Dyn1 isoforms are involved in housekeeping CME at rest, and were pre-clustered on the plasma membrane. The low density of clusters for all Dyn1 isoforms before stimulation (Fig. 3d–f) suggests a limited level of CME and/or precursor coated pits. In comparison, the significantly increased density of clusters for all Dyn1 isoforms following stimulation (Fig. 3d–f) reflects an increased level of compensatory endocytosis and confirms the recruitment of Dyn1 molecules to these clusters, either directly from cytosol (Dyn1aa), or via lateral trapping from the plasma membrane (Dyn1bb and ab). Importantly, short-tail isoform Dyn1bb exhibits a significantly increased cluster lifetime (Fig. 3d), as well as an increased trend in cluster radius (Fig. 3d), compared to long-tail isoform Dyn1aa (Fig. 3f). Additionally, repeated clustering in the same area of the plasma membrane, defined as a cluster hotspot, increased significantly for Dyn1bb and Dyn1ab in response to stimulation (Fig. 3d, e). This was not observed for Dyn1aa (Fig. 3f). Although the short-tail isoforms (Dyn1bb and ab) formed clusters of a lower density compared to long-tail isoform Dyn1aa prior to stimulation, their ability to respond to stimulation was greatly enhanced by lateral diffusion—with the density of clusters significantly increasing in response to stimulation, far exceeding that of Dyn1aa.

### Dyn1 single molecules exhibit lateral trapping into nanoclusters in neurosecretory cells and neurons

We have demonstrated that Dyn1xb (short-tail isoforms) are recruited to the HFA of the plasma membrane following secretagogue stimulation through a bi-phasic mechanism (Fig. 1a, b), and that they undergo activity-dependent nanoclustering (Fig. 3d, e). This data suggests Dyn1xb isoforms are first recruited to the plasma membrane and then are laterally trapped into nanoclusters. To provide direct evidence of lateral trapping, we sought to quantify moments in which laterally diffusing Dyn1 molecules become actively-confined within nanoscale clusters. First, spatiotemporal nanoclusters were derived using a Density-Based Spatial Clustering of Applications with Noise (DBSCAN) algorithm with $n = 3$ dimensions. Neighborhood proximity was established by transforming the temporal component of each detection into a pseudo-spatial component (using a spatiotemporal scalar unit) such that the neighborhood radius ($\varepsilon$) applies in both space and transformed time. Via this approach, clusters are derived based on both the spatial and temporal proximity of detections between trajectories. Following spatiotemporal clustering, trajectories were filtered to isolate those that first appear within the evanescent field of excitation, intercept a nanocluster, and are subsequently confined within the nanocluster for the duration of its lifetime (see "Methods" for full description).

Trajectories with this specific two-dimensional movement pattern were exhibited by all Dyn1 isoforms (Fig. 4a). Following entry into a spatiotemporal nanocluster, the mobility of trajectories was significantly reduced (Fig. 4b–d). Therefore, individual Dyn1 molecules can diffuse laterally and become actively confined within nanoclusters, where they exhibit significantly lower mobility. Both short-tail isoforms exhibited a significantly greater proportion of trajectories which undergo two-dimensional sampling followed by active confinement within nanoclusters in comparison to the long-tail Dyn1aa isoform (Fig. 4e). These results provide further evidence that active lateral trapping into nanoclusters is fundamental to the recruitment of short-tail Dyn1 isoforms to endocytic sites.

As neurons exhibit a wide range of endocytic modes[14], we next tested the single molecule mobility dynamics of both long- and short-tail Dyn1 isoforms in cultured murine primary hippocampal neurons. Dyn1aa-mEos2, Dyn1ab-mEos2, or Dyn1bb-mEos2 was co-transfected along with Synaptotagmin1-pHluorin (to identify active synapses). SptPALM was performed on neurons in low K$^+$ buffer and imaged before and during electrical stimulation to elicit compensatory endocytosis (10 Hz, 30 s) (Fig. 5a–c). The mobility of the three Dyn1 isoforms in synapses was higher compared to that found in PC12 cells but

displayed the same activity dependent changes (Fig. 5a–c). The mobility of short-tail isoforms Dyn1bb-mEos2 and Dyn1ab-mEos2 decreased significantly following electrical stimulation (Fig. 5a, b). As observed in PC12 cells, Dyn1aa-mEos2 mobility did not significantly change following electrical stimulation in synapses (Fig. 5c). Similarly, following potassium stimulation in hippocampal neurons, Dyn1bb-mEos2 mobility was significantly decreased, whereas Dyn1aa-mEos2 mobility was not significantly altered (Supplementary Fig. 5). Taken together, these data suggest that short-tail Dyn1 isoforms cluster through plasma membrane recruitment, followed by lateral trapping into nanoclusters.

We also performed spatiotemporal cluster analysis on Dyn1 within the synapses of electrically stimulated hippocampal neurons. Similarly, to that we observed in PC12 cells, we found that Dyn1 isoforms exhibited lateral trapping (Fig. 5d and Supplementary Fig. 6), whereby upon entering a spatiotemporal nanocluster, the mobility of Dyn1 molecules is significantly reduced (Fig. 5e).

### Calcineurin controls the second concentration/clustering phase of short-tail Dyn1 isoforms

Dynamin short-tail isoforms respond to stimulation by significantly increasing nanocluster density and lifetime. Synaptic activity is known to promote dynamin dephosphorylation, a step that is critical for compensatory endocytosis at the synapse[17]. We therefore tested the effect of phosphorylation/dephosphorylation in promoting the activity-dependent clustering of dynamin short-tail isoform Dyn1bb. Activity-dependent mobility changes of Dyn1bb-mEos2 were assessed following either okadaic acid (OA) treatment (1 μM, 30 min), or staurosporine (Sta) treatment (1 μM, 30 min). OA strongly inhibits protein phosphatases, whilst Sta inhibits protein kinases[32,33]. Treatment with OA blocked the previously observed activity-dependent decrease in Dyn1bb-mEos2 mobility (Supplementary Fig. 7a). Interestingly, following Sta-treatment, Dyn1bb-mEos2 molecules exhibited a low mobility prior stimulation, which significantly increased following stimulation (Supplementary Fig. 7b). This low mobility of Dyn1bb prior to stimulation could stem from its phosphorylation status, whereby in the presence of Sta, it is confined into nanoclusters. Sta has a high affinity against, and inhibits, a broad selection of protein kinases. To determine the specific phosphatase responsible for Sta-induced Dyn1bb mobility changes, we investigated the role of Calcineurin, a phosphatase previously known to interact with dynamin[17].

Calcineurin is an important synaptic phosphatase known to dephosphorylate dynamin[17]. Short-tail Dyn1xb isoforms exhibit a unique calcineurin binding site at their C-terminus[22] that could be responsible for their activity-dependent clustering. This calcineurin binding site is not present in long-tail isoform Dyn1aa[22]. We next assessed the dependency of Dyn1bb clustering on calcineurin binding by using calcineurin inhibitor cyclosporin-A[34]. PC12 cells transfected with either Dyn1bb-GFP (TIRF) or Dyn1bb-mEos2 (sptPALM) were pre-treated with cyclosporin-A (40 μM, 10 min) prior to imaging. Cyclosporin-A treatment did not affect the first phase of Dyn1bb-GFP recruitment (Fig. 6a, b). However, it strongly prevented the formation of high FI areas (HFA) during the second recruitment phase following stimulation (Fig. 6a–c). Following cyclosporin-A treatment, we found that Dyn1bb-mEos2 molecules exhibited a very low mobility at rest, indicating that the inhibition of calcineurin activity could confine Dyn1bb on the plasma membrane (Fig. 6d–f). Importantly, Ba$^{2+}$-stimulation did not further decrease Dyn1bb-mEos2 mobility in the presence of cyclosporin-A. Calcineurin inhibition therefore prevented the previously observed activity-dependent decrease in Dyn1bb-mEos2 mobility elicited by stimulation (Fig. 2b). These results suggest that calcineurin has a central role in the lateral trapping of Dyn1bb into nanoclusters during the second "concentration/clustering phase".

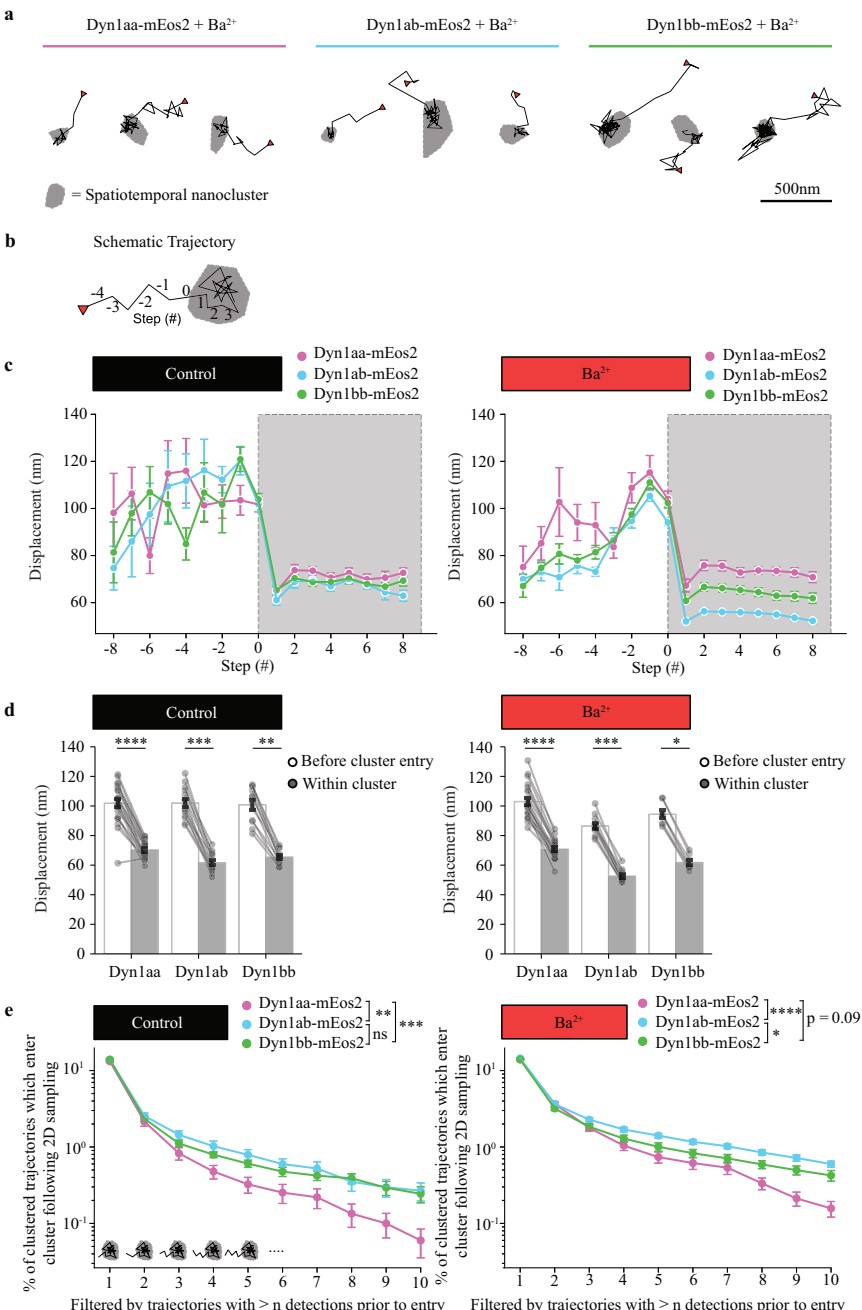

**Fig. 4 | Nanoscale two-dimensional sampling followed by active cluster confinement of Dyn1aa, Dyn1ab and Dyn1bb isoforms. a** Representative trajectories from Dyn1aa-mEos2, Dyn1ab-mEos2 and Dyn1bb-mEos2 following $Ba^{2+}$ secretagogue stimulation which undergo sampling of an approximate two-dimensional space (defined by the evanescent field) followed by active confinement within a spatiotemporal nanocluster (represented by gray area; red triangle represents first detection in trajectory). **b** Schematic of a trajectory indicating step number relative to the step of entry (step = 0) into a nanocluster. **c** Frame-to-frame displacements with respect to step number of those trajectories which enter a nanocluster during their lifetime (gray region represents steps made following nanocluster entry; Mean ± SEM). **d** Comparison of the frame-to-frame displacements of trajectories before cluster entry and following cluster entry; each point represents the average displacements for an individual cell (two-tailed pairwise Wilcoxon signed-rank test followed by Bonferroni correction; $p$ value from left to right for control: $p < 0.0001$, $p = 0.0002$, $p = 0.0015$, for $Ba^{2+}$: $p < 0.0001$, $p = 0.0007$, $p = 0.0234$; Mean ± SEM).

**e** Percentage of clustered trajectories which enter a spatiotemporal nanocluster following approximate two-dimensional sampling, where percentages were calculated after filtering those trajectories which exhibited ≥ $n$ detections prior to entry (nparLD implementation of the non-parametric mixed effects model[61]; F1-LD-F1; whole-plot factor = isoform, sub-plot factor = $n$ prior filter). Control: ANOVA-Type statistic (isoform), $p = 0.0009$; demonstrating an effect of isoform. Stimulation: ANOVA-Type statistic (isoform), $p < 0.0001$; demonstrating an effect of isoform (plots show nparLD model-associated pairwise comparisons, $p$ values adjusted for with the Bonferroni correction; for control: aa/ab $p = 0.0022$, aa/bb $p = 0.0003$, ab/bb $p = 0.9139$, for $Ba^{2+}$: aa/ab $p < 0.0001$, aa/bb $p = 0.0304$, ab/bb $p = 0.0116$; Mean ± SEM). For (**c**–**e**): control Dyn1aa; $n = 23$ cells, Dyn1ab; $n = 15$ cells, Dyn1bb; $n = 12$ cells: $Ba^{2+}$ Dyn1aa; $n = 22$ cells, Dyn1ab; $n = 13$ cells, Dyn1bb; $n = 8$ cells. ns, non-significant $p > 0.05$; *$p < 0.05$; **$p < 0.01$; ***$p < 0.001$; ****$p < 0.0001$. Source data are provided as a Source Data file.

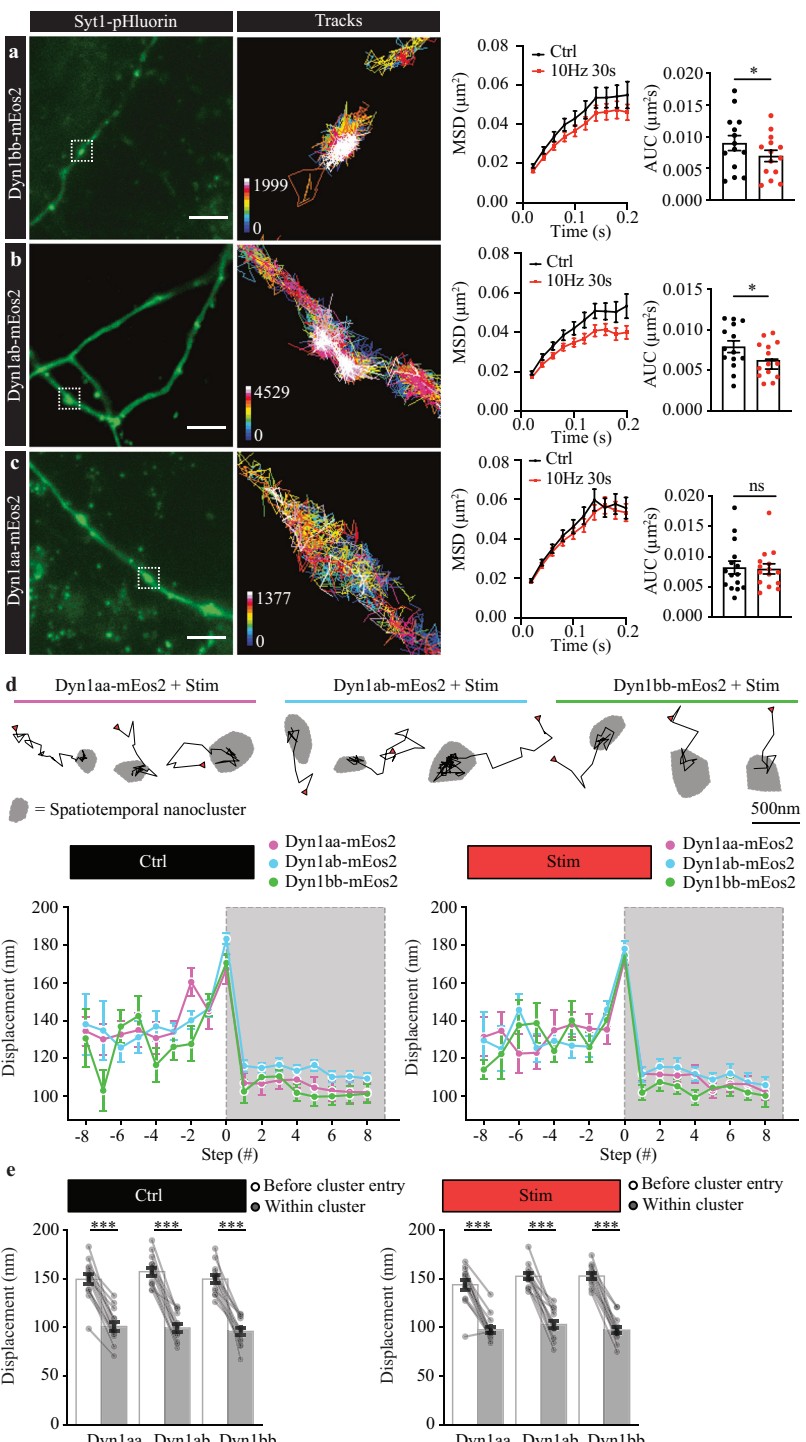

**Fig. 5 | Single molecule mobility dynamics of Dyn1bb, Dyn1ab and Dyn1aa in mouse primary hippocampal neurons.** Dyn1bb-mEos2, Dyn1ab-mEos2, or Dyn1aa-mEos2 was co-transfected with Synaptotagmin1-pHluorin and imaged at 50 Hz in either low K$^+$ (control) or subsequent 10 Hz 30 s (electrical stimulation) conditions in mouse primary hippocampal neurons. From left to right, representative Syt1-pHluorin images with white dotted boxes show magnified sptPALM trajectory maps, average MSD ($\mu m^2$) and the AUC of the MSD ($\mu m^2 s$) for (**a**) Dyn1bb-mEos2, (**b**) Dyn1ab-mEos2, and (**c**) Dyn1aa-mEos2 in a low K$^+$ condition. Scale bar, 5 µm.
**d** Frame-to-frame displacements with respect to step number of those trajectories which enter a nanocluster during their lifetime (gray region represents steps made following nanocluster entry; Mean ± SEM). **e** Comparison of the frame-to-frame

displacements of trajectories before cluster entry and following cluster entry; each point represents the average displacements for an individual cell (paired Wilcoxon signed-rank test followed by Bonferroni correction; p value from left to right for control: $p = 0.0002$, $p = 0.0004$, $p = 0.004$, for Ba$^{2+}$: $p = 0.0004$, $p = 0.0004$, $p = 0.0004$; Mean ± SEM). ($n = 15$ neurons for Dyn1aa, $n = 14$ neurons for Dyn1ab, $n = 14$ neurons for Dyn1bb from 5 independent experiments; normally distributed paired data was analyzed with two-tailed paired Student's $t$ test for Dyn1aa, $p = 0.7615$, and non-normally distributed paired data was analyzed with two-tailed Wilcoxon test for Dyn1bb, $p = 0.0289$, and Dyn1ab, $p = 0.0247$; ns, non-significant $p > 0.05$; *$p < 0.05$; ***$p < 0.001$; Mean ± SEM are plotted). Source data are provided as a Source Data file.

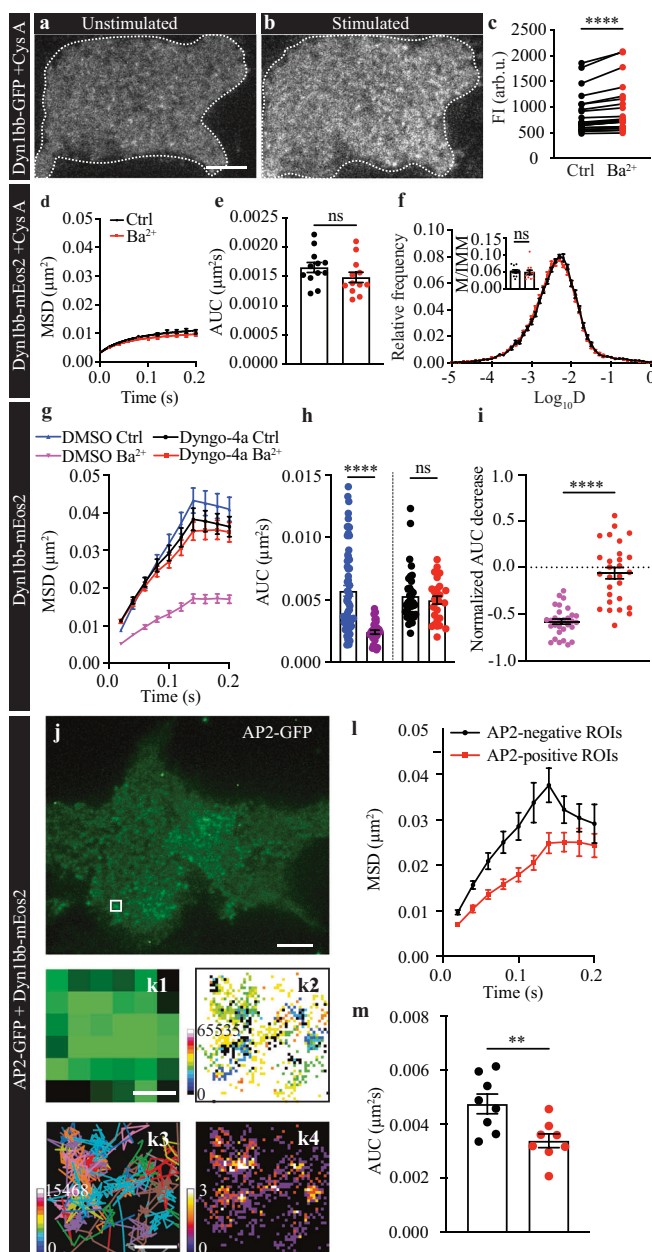

**Fig. 6 | Dyn1bb intrinsic GTPase activity and calcineurin activity are essential to mediate clustering of short-tail Dyn1bb in AP2-positive sites. a–f** PC12 cells expressing either Dyn1bb-GFP or Dyn1bb-mEos2 were pre-treated with cyclosporin-A (CysA; 40 μM for 10 min), and imaged at 50 Hz in both unstimulated and stimulated (2 mM Ba$^{2+}$) conditions. Representative TIRF images of CysA pre-treated PC12 cells expressing Dyn1bb-GFP in (**a**) unstimulated and (**b**) Ba$^{2+}$-stimulated conditions. Scale bar, 5 μm. Outlines of the cells are indicated for clarity. **c** Significant increase in average FI (arbitrary units; arb.u.) across the entire plasma membrane of PC12 cells transfected with Dyn1bb-GFP following stimulation (non-normally distributed paired data was analyzed with two-tailed Wilcoxon test, $n = 24$ cells from 3 independent experiments, $p < 0.0001$). The (**d**) average MSD (μm$^2$), (**e**) AUC of the MSD (μm$^2$s), and (**f**) mean frequency distribution of Log$_{10}$D with an inset showing the M/IMM fraction, of Dyn1bb-mEos2 single molecule mobility compared between unstimulated and stimulated PC12 cells treated with CysA. ($n = 12$ cells from 3 independent experiments; normally distributed paired data was analyzed with two-tailed paired Student's $t$ test (**e**), $p = 0.0778$, or non-normally distributed paired data was analyzed with two-tailed Wilcoxon test (**f**), $p = 0.3804$; ns non-significant $p > 0.05$, ****$p < 0.0001$; Mean ± SEM are plotted). **g–m** Dyn1bb-mEos2 was imaged at 50 Hz in either unstimulated or stimulated (2 mM Ba$^{2+}$) conditions in transfected PC12 cells treated with either DMSO or Dyngo-4a (30 μM for 30 min). **g** The average MSD (μm$^2$), and (**h**) AUC of the MSD (μm$^2$s) of Dyn1bb-mEos2 was compared between unstimulated and stimulated PC12 cells that were treated with either DMSO or Dyngo-4a. **i** The decrease in the AUC of the MSD following stimulation was compared between DMSO and Dyngo-4a conditions. ($n = 62$ cells for DMSO Ctrl and $n = 30$ cell for DMSO Ba$^{2+}$, $n = 34$ cells for Dyngo-4a Ctrl and $n = 28$ cells for Dyngo-4a Ba$^{2+}$, from 3 independent experiments; non-normally distributed unpaired data was analyzed with two-tailed Mann–Whitney test for (**h**), $p < 0.0001$ for DMSO and $p = 0.8387$ for Dyngo-4a; normally distributed unpaired data was analyzed with two-tailed unpaired Student's $t$ test for (**i**), $p < 0.0001$; ns non-significant $p > 0.05$, ****$p < 0.0001$; Mean ± SEM are plotted). **j** Representative image of AP2-GFP in transfected PC12 cells, with a white box showing a representative AP2-positive cluster magnified in (**k**). Scale bar, 5 μm. (k1) Representative low-resolution image, (k2) sptPALM diffusion coefficient map (darker points are more confined), (k3) trajectory map (color coded as they appear in time) and (k4) average intensity map (brighter points indicate higher intensity) of Dyn1bb-mEos2 in the region highlighted in (**j**). The (**l**) average MSD (μm$^2$), and (**m**) AUC of the MSD (μm$^2$s), of Dyn1bb-mEos2 were compared in AP2-positive and -negative regions of interest (ROIs) in unstimulated PC12 cells. ($n = 8$ for both AP2-positive and -negative ROIs from 2 independent experiments; normally distributed unpaired data was analyzed with two-tailed unpaired Student's $t$ test, $p = 0.0086$; **$p < 0.01$; Mean ± SEM are plotted). Source data are provided as a Source Data file.

## Lateral trapping of Dyn1 short-tail isoforms is controlled by dynamin GTPase activity

Short-tail isoforms are involved in CME, as well as other endocytic routes[22]. During CME, dynamin molecules oligomerize at the neck of nascent endosomes to promote fission[35]. Dynamin GTPase activity is enhanced by dynamin oligomerization[36]. In addition, the assembly of dynamin is thought to occur exclusively at the membrane, mediated by other factors such as lipid binding. We therefore tested whether the activity-dependent lateral trapping observed for the Dyn1 short-tail isoforms depends on the GTPase activity of dynamin. For this, we used a small molecule dynamin inhibitor Dyngo-4a, which blocks the GTPase activity of dynamin and prevents dynamin-dependent endocytosis[37,38]. Dyngo-4a treatment (30 μM for 30 min) in PC12 cells transfected with Dyn1bb-mEos2 prevented the activity-dependent decrease in Dyn1bb single-molecule mobility (Fig. 6g–i). This indicates that the GTPase activity of the short-tail Dyn1bb isoform contributes to its observed lateral trapping into nanoclusters.

## Short-tail isoform Dyn1bb forms nanoclusters in AP2-positive clusters

To study whether the observed decrease in the single-molecule mobility of Dyn1bb-mEos2 occurs within nanoclusters that define endocytic sites, we performed dual-color imaging of PC12 cells co-transfected with GFP-tagged adaptor protein 2 (AP2-GFP) and Dyn1bb-mEos2. AP2-GFP positive clusters, which indicate the locations of nascent endocytic structures, were used to select regions of interest (ROIs) for sptPALM analysis of Dyn1bb-mEos2 (Fig. 6j). Representative super-resolved trajectory, average intensity, and average diffusion coefficient maps of Dyn1bb-mEos2 in AP2-GFP ROIs (Fig. 6k) provide an example of the confinement of dynamin molecules within those areas. Our results showed that the mobility of Dyn1bb-mEos2 was significantly lower within AP2-GFP-positive clusters (dark colors in k2 diffusion maps and hot colors in k4 intensity map) compared to its mobility outside of these regions (Fig. 6l, m). This strongly suggests that following recruitment to the plasma membrane, Dyn1bb molecules are trapped and confined within endocytic structures.

## Endogenous Dynamin is pre-clustered on the plasma membrane of MEFs and PC12 cells

The presence of long-tail Dyn1 isoform pre-assembled clusters (at rest), was recently demonstrated to be essential for ultrafast

endocytosis, suggesting that these clusters are required for the rapid availability of Dyn1xa isoforms to promote the fission of nascent recycling endosomal vesicles[14]. Our results suggest that in neurons and neurosecretory cells, the formation of pre-assembled clusters is not limited to the long-tail isoforms, but also occurs for the short-tail isoforms—with cluster membership increasing upon stimulation (Fig. 3). However, these experiments were done in an overexpression background, which is subject to potential artifacts. To investigate whether endogenous dynamin nanoclusters are also present on the plasma membrane, we used a genetically modified Mouse Embryonic Fibroblast (MEF) conditional double knock-out (DKO) cell line in which Dyn1 and 2 can be depleted (MEF$^{Dyn1,2}$ DKO)[39]. MEF cells endogenously express Dyn1 and 2[39], but Dyn3 is not expressed at detectable levels[39]. Upon treatment with 4-hydroxy-tamoxifen (4OH-TMX), Dyn1 and 2 are depleted and imaged after 6 days (untreated = Ctrl). We took advantage of Fluorescent intrabody Localization Microscopy (FiLM)[40] to express an anti-dynamin single-domain nanobody raised against GTP-loaded Dyn1 and 2[41] genetically tagged with mEos2 (DynNB-mEos2). This technique provides a unique opportunity to capture the dynamic nanoscale organization of endogenous proteins in live cells[40]. Upon

expression of DynNB-mEos2 in untreated MEF$^{Dyn1,2}$ DKO cells (Ctrl), we could detect clusters of endogenous dynamin on the plasma membrane (Fig. 7a) that were reminiscent of those found in PC12 cells using sptPALM (Fig. 2). The 4OH-TMX treated and untreated MEF cells were electroporated to express DynNB-mEos2. As expected, the number of detections of endogenous dynamin significantly decreased in 4OH-TMX treated cells (Fig. 7a–c). To quantify the nanoscale organization of endogenous dynamin we performed NASTIC analysis and revealed that dynamin nanoclusters were of similar size and lifetime to that found in PC12 cells expressing DynNB-mEos2 (Supplementary Table 1). Our results demonstrate that endogenous dynamin is present and organized in nanoclusters on the plasma membrane of both fibroblasts and neurosecretory cells.

Serum-starvation followed by re-supplementation triggers endocytosis in MEF cells[42]. We therefore investigated whether re-supplementation of serum-starved MEF$^{Dyn1,2}$ DKO cells could trigger additional lateral trapping of Dyn1, in a similar way to secretagogue stimulation in PC12 cells. For this, 4OH-TMX-treated MEF$^{Dyn1,2}$ DKO cells re-expressing Dyn1bb-mEos2 were serum-starved in normal culture medium for 30 min to decrease the rate of endocytosis. Serum was

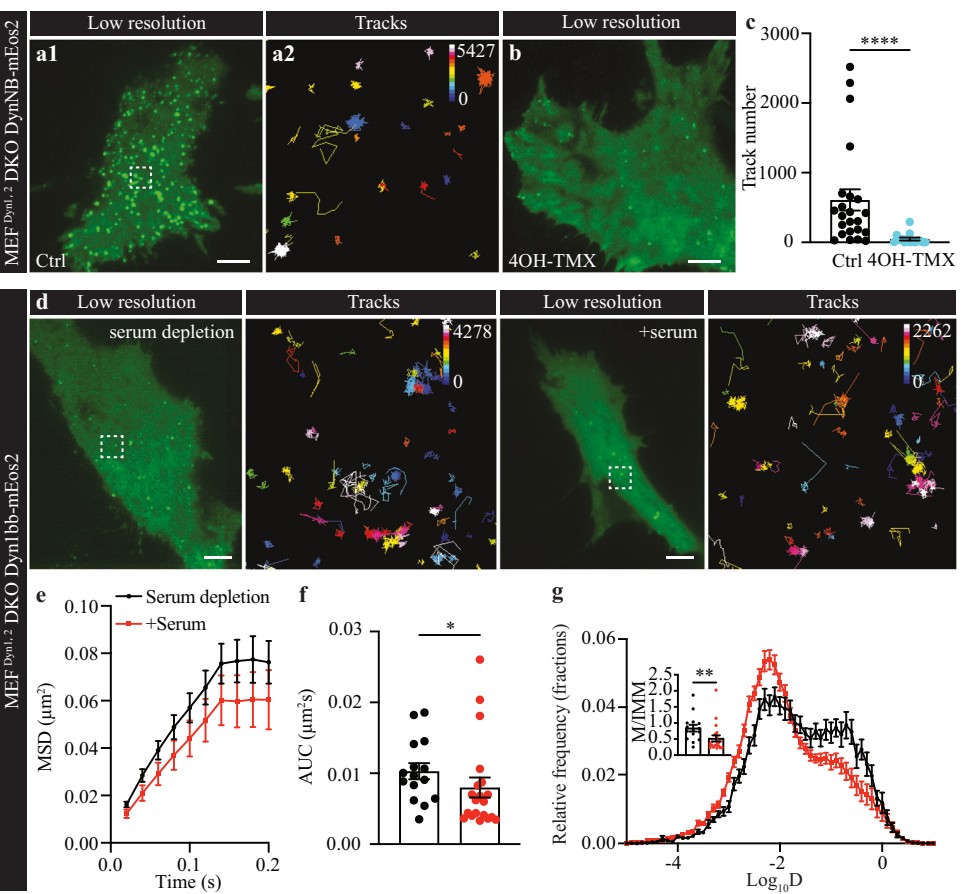

**Fig. 7 | Dyn1bb-mEos2 mobility decreased during induced endocytosis in MEF$^{Dyn1,2}$ DKO cells. a–c** DynNB-mEos2 tracks were imaged at 50 Hz in transfected conditional MEF$^{Dyn1,2}$ DKO cells that were either treated with 4OH-TMX to induce the Dyn1 and 2 conditional double knockout (DKO), or left untreated (Ctrl). Representative low-resolution image of DynNB-mEos2 in MEF$^{Dyn1,2}$ DKO cells either (**a**) left untreated (Ctrl), with the trajectory map corresponding to the white dashed box ROI shown magnified on the right, or (**b**) treated with 4OH-TMX. Scale bar, 5 μm. **c** Number of tracks obtained for DynNB-mEos2 in 4OH-TMX treated and untreated MEF$^{Dyn1,2}$ DKO cells (non-normally distributed unpaired data was analyzed with two-tailed Mann−Whitney test, $p < 0.0001$). **d–g** Dyn1bb-mEos2 mobility was imaged at 50 Hz upon starvation and following serum-supplementation in transfected MEF$^{Dyn1,2}$ DKO cells. **d** From left to right, representative low-resolution

images with trajectory maps corresponding to the white dashed box outline shown magnified on the right for Dyn1bb-mEos2 in starvation, and serum re-supplementation conditions, respectively. Scale bar, 5 μm. The (**e**) average MSD (μm²), (**f**) AUC of the MSD (μm²s), and (**g**) mean frequency distribution of Log$_{10}$D (Diffusion coefficient, D) with an M/IMM fraction inset, for Dyn1bb-mEos2 in starvation and serum-supplementation conditions. ($n = 15$ cells for starvation condition and $n = 20$ cells for serum-supplemented condition, from 3 independent experiments; non-normally distributed unpaired data was analyzed with two-tailed Mann−Whitney test, $p = 0.0302$ for (**f**) and $p = 0.0030$ for (**g**); *$p < 0.05$, **$p < 0.01$, ****$p < 0.0001$; Mean ± SEM are plotted). Source data are provided as a Source Data file.

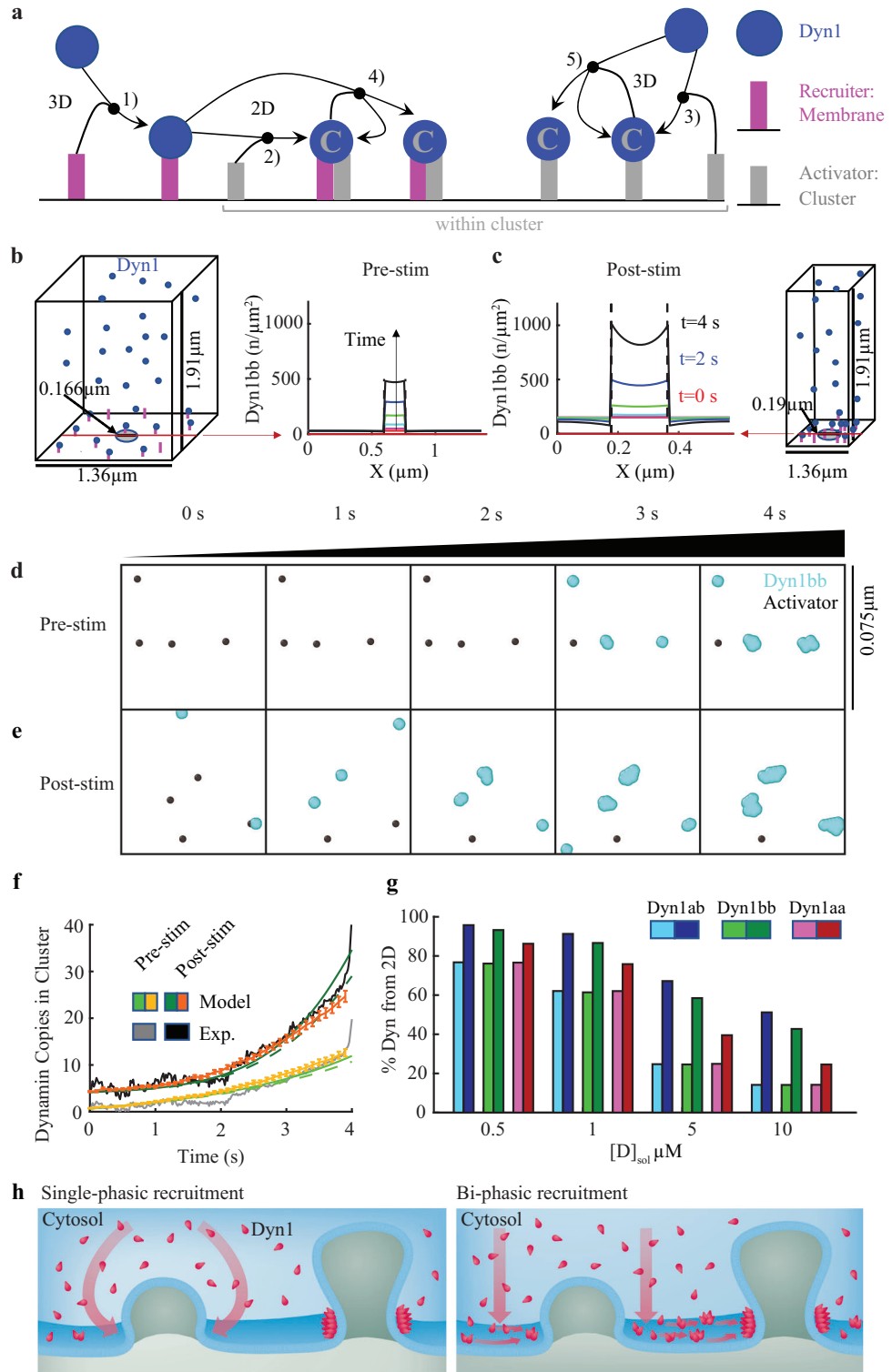

then added back to the cells to promote endocytosis, and imaged immediately by sptPALM. Representative low-resolution TIRF images and sptPALM trajectory maps of Dyn1bb-mEos2 in both serum depleted and +serum conditions are shown in Fig. 7d. As expected, the mobility of Dyn1bb-mEos2 decreased significantly after the supplementation of serum and recommencement of endocytosis (Fig. 7e–g). In addition, Dyn1bb-mEos2 single molecules are able to form clusters in the absence of any other Dyn isoforms and become more immobilized during endocytosis. This decrease in mobility suggests that

lateral trapping is a general mechanism involved in the clustering of short-tail Dyn1 isoforms.

### Resident dynamin clusters facilitate further dynamin clustering upon stimulation

A number of studies have suggested that dynamin could be present at the plasma membrane of neurons at all times, to control fusion pore dynamics during exocytosis[43–45] and ultrafast endocytosis[14]. Our study demonstrates that pre-clustering is a general mechanism that also

**Fig. 8 | A spatial reaction-diffusion model of dynamin recruitment to clusters reproduces experimental growth, showing lateral recruitment from the membrane. a** Our minimal reaction network for Dyn1 (blue circle) and two types of recruitment molecules on the membrane surface that bind Dyn1 (magenta and gray rods). Recruiters are either initially well-mixed/homogeneous across the membrane surface (with concentration $[R]_{mem}$, magenta rods) or are restricted to the cluster sites on the surface (with concentration $[A]_{clus}$, gray rods). Reactions 2, and 3 involve only one Dyn1 and use the association rate $k_{mem}$. Reactions 4, 5 involve two copies of Dyn1 and use the association rate $k_{dyn}$. Reaction 1 is a reversible binding between one Dyn1 and one uniform recruiter ($R$) with rates $k_{frev}$ and $k_b$. **b** The system geometry is designed based on the measured density of clusters, illustrated here based on 0.54 clusters/$\mu m^2$ for Dyn1bb pre-stimulation. Assuming that the clusters are evenly distributed on the surface, we can then model recruitment to a single cluster rather than the entire surface, as the identical dynamics would play out to adjacent clusters. In addition to the cluster density, we also specify the volume $V$ to area $A$ ratio of the cell, which we set at $V/A = 1.9\,\mu m$, a representative value for a mammalian cell type[15]. The plot reports result from the PDE simulations of one optimal model, showing how the density of Dyn1 increases within the cluster radius as time progresses while staying relatively flat outside of the cluster site. **c** The same model post-stimulation, where the system geometry is now designed based on the increased cluster density of 3.1 clusters/$\mu m^2$. The initial density of Dyn1bb on the membrane is higher, following the experiments. The cluster density also increases sharply within the cluster, with more depletion simultaneously occurring now outside of the cluster due to the higher density of clusters on the membrane (smaller area per cluster). Although the measured diffusion on the membrane is relatively slow at 0.02–0.05 $\mu m^2$/s, it is sufficient to support a majority of recruitment from the

membrane after stimulation. Snapshots from stochastic reaction-diffusion simulation trajectories (**d**) pre-stimulation and (**e**) post-stimulation. The frame is zoomed in to part of the cluster region where Dyn1 has stochastically assembled (side length 75 nm). The gray points are the activators, the cyan molecules are Dyn1. The membrane recruiters in magenta are not visible as they are bound to the Dyn1 molecules. Each Dyn1 has 4 sites to recruit additional Dyn1 within the cluster. **f** The total copies in the cluster are compared to the experimental growth curve, that is scaled by the measured increases in single-particle tracks. The same optimal model for Dyn1bb is shown as from (**b**, **c**). The non-spatial model is in solid lines, the spatial PDE model is in dashed lines, and the stochastic spatial model is in orange, averaged for each system over 48 trajectories with standard deviation shown. This optimized model has initial concentrations for Dyn1 as $[D]_{mem} = 30$ copies/$\mu m^2$ and $[D]_{sol} = 10\mu M$, where the subscript denotes localization on membrane or in solution. Uniform recruiters are initially concentrated at $[R]_{mem} = 4.85$ copies/$\mu m^2$, and "activators" are $[A]_{clus} = 207$ copies/$\mu m^2$. Optimal rates are $k_{mem} = 0.017\ \mu M^{-1}\,s^{-1}$, $k_{frev} = 0.09\ \mu M^{-1}\,s^{-1}$, $k_{memPost} = 0.0024\mu M^{-1}\,s^{-1}$, $k_{dyn} = 0.035\ \mu M^{-1}\,s^{-1}$, $k_b = 0.15\ s^{-1}$, $h = 10$ nm. For the stochastic spatial model, $k_{memPost} = 0.006\mu M^{-1}\,s^{-1}$. **g** For all isoforms, the recruitment proceeds in part from the lateral 2D surface, but for Dyn1ab and Dyn1bb, the membrane-bound density following stimulation rises significantly higher, and thus there is a larger change in 2D recruitment after stimulation. For Dyn1aa, the changes following stimulation are more modest, and thus the recruitment is more similar before and after stimulation. We fix $[D]_{mem} = 10$ copies/$\mu m^2$ for all these models, meaning that the copies in solution always out-number the membrane copies, for $[D]_{sol} = 10\mu M$ it is by 1000-fold. Source data are provided as a Source Data file. **h** Schematic of single-phasic recruitment of Dyn1aa and bi-phasic recruitment mechanism of Dyn1ab/bb.

occurs in neurosecretory and MEF cells. To better understand the role of pre-clustered dynamin, we developed a spatial reaction-diffusion model and recapitulated the kinetics of dynamin assembly into clusters on the plasma membrane.

The observed local increase in Dyn1 concentration on the membrane following stimulation could indicate either a higher density of Dyn1 clusters, or existing clusters with larger sizes or more sustained recruitment. Our data reveals that the first mechanism occurs: significantly more clusters formed in response to stimulation (Fig. 3), whereas the lifetime and sizes of the clusters only changed by modest factors of ~1–1.3 (Fig. 3d–f). We developed a spatial reaction-diffusion model using differential equations to quantify mechanisms of Dyn1 recruitment to clusters from the cytosol and membrane, both before and after stimulation (Fig. 8a–c). We defined a minimal model with three components to limit the number of free parameters as much as possible while still capturing the observed experimental dynamics: (1) Dyn1, (2) a uniformly distributed membrane recruiter and (3) a cluster-associated recruiter or "activator" that is spatially restricted to cluster sites. The membrane recruiter represents a population of proteins (such as a variety of SH3-domain containing proteins[46]) and lipids (such as PI(4,5)P$_2$) already on the plasma membrane that recruit Dyn1. The cluster-associated "activator" mimics Dyn1-binding proteins assembled at sites of clathrin-coated pits (such as amphiphysin and intersectin)[3]. As illustrated in the reaction network in Fig. 8a, Dyn1 binds the membrane recruiters from 3D solution, and binds the activators either directly from 3D or laterally from 2D after binding the membrane recruiters. Only when bound to an activator can Dyn1 bind to other Dyn1 molecules (either from 3D or 2D), which facilitates enrichment in the cluster sites (Fig. 8a and see "Methods" for full model description).

For each isoform, our model is constrained by several experimental observables quantified in Supplementary Table 2: (1) the kinetics of the fluorescence within forming clusters (Fig. 1j, k), (2) the relative increase in densities of dynamin in the cluster, (3) the relative increase in dynamin density on the membrane following stimulation, (4) the density of clusters formed on the membrane, both before and following stimulation, and (5) the diffusion constants of dynamin on the membrane. The free parameters of our models are the species

concentrations and binding rates, where we use physically reasonable values as ranges (see Supplementary Table 3 and "Methods") over which we allow optimization. For each isoform we optimize specific values of concentrations and rates using stochastic optimization with genetic algorithms to maximize agreement between the modeled kinetics of Dyn1 accumulation in clusters compared to the same experimental observable (Fig. 8f and Supplementary Fig. 8). We further recapitulated these same kinetics with a structure-resolved stochastic reaction-diffusion model that explicitly incorporated Dyn1 self-assembly[47] (Fig. 8d, e and Movie 2 and 3).

The model shows that while recruitment of Dyn1 to clusters prior to stimulation can proceed via a combination of 3D binding from the cytosol, and 2D lateral recruitment, following stimulation, recruitment via 2D lateral recruitment is always increased (Fig. 8g). Notably, the increase in 2D recruitment is most pronounced for the Dyn1bb and Dyn1ab short-tail isoforms. They form clusters predominantly from 2D following stimulation (Fig. 8g), driven by the significant increase of 5–7x in cluster density (Fig. 3d–f). The Dyn1aa isoform shows a more modest increase in 2D recruitment that is consistent with its more modest overall changes of ~2x cluster density following stimulation. We show that 2D recruitment plays an important role in cluster formation even when the cytosolic (solution) Dyn1 concentration, $[D]_{sol}$, gives rise to 1000x more copies of Dyn1 than are localized to the membrane. Although diffusion in solution is at least 100 times faster, the Dyn1 localized to the membrane can exploit dimensional reduction, or the reduced search space of 2D, to more efficiently bind to the cluster[15,48]. The model predicts that with a lower $[D]_{sol}$, or with a higher concentration of membrane bound Dyn1, more of the recruitment proceeds from the membrane (Fig. 8g).

For the model to reproduce the relatively sharp increase in Dyn1 density following a very slow initial growth as detected in Fig. 1j, k, we needed to introduce a cooperative effect. Dyn1 not only had to bind to the "activators" or cluster-associated proteins[3], but it had to subsequently recruit itself to the clusters via dimerization (continuum differential equations) or oligomerization (stochastic reaction-diffusion), mimicking assembly of a metastable dynamin helix. By explicitly capturing Dyn1 self-assembly in the stochastic simulations using the same optimized rates as the ordinary differential equations, we can directly

see that once a single Dyn1 is "activated" it helps recruit multiple additional Dyn1 (from 1–4 copies in our model) to cluster sites (Movie 4). It is important to note that dynamin oligomerises in two ways, as rings or as helices. Dyngo-4a specifically inhibits helical formation states of dynamin but does not affect ring formation[37]. Our model is therefore in good agreement with the inhibitory effect of Dyngo-4a on the activity-dependent reduction in mobility (Fig. 6). Further, our model indicates that Dyn1 recruitment is not a rate-limiting step. If that were so, the increased Dyn1 density on the membrane following stimulation would accelerate Dyn1 recruitment, but since that is not the case, this indicates that the sizes and lifetimes of the Dyn1 clusters are controlled by the nucleators that form them. These nucleators thus must also be more abundant on the surface to increase cluster densities following stimulation.

Our model showed that the formation of clusters following stimulation resulted in more local reorganization of Dyn1 from the membrane, and higher molecular membership within the clusters, for the Dyn1ab and Dyn1bb isoforms (Fig. 8c). This could perhaps contribute to the "hotspots" (repeated clustering) observed experimentally, where locations of previous clusters were more likely to nucleate new clusters due to inhomogeneity in Dyn1 density. In contrast, prior to stimulation, Dyn1 density remained relatively uniform and sparse across the surface, reducing the bias toward nucleating at the same positions (Fig. 8b).

We note that despite the slow diffusion on the membrane of 0.02 $\mu m^2$/s, that is still sufficiently fast to support an average displacement of 0.57 $\mu m$ across the membrane surface during the 4 s of cluster formation ($\triangle r \sim \sqrt{4D\triangle t}$). From the perspective of a cluster site, that means that all Dyn1 molecules within that 0.57 $\mu m$ distance can potentially contribute to cluster formation from 2D (Fig. 8b, c). The experiments above show a population of diffusing Dyn1 on the membrane, and our model shows that they will collide with cluster sites. Even with moderate association rates predicted by the model of ~$10^4 M^{-1}s^{-1}$ (i.e., not all collisions result in binding events), the Dyn1 molecules ultimately bind to the cluster sites and one another from 2D. Our model allows 2D association because it is in general not physically justified to prohibit 2D association and only allow association from 3D. The only way to eliminate 2D association is to assert that the molecular conformations of Dyn1 are oriented so that the binding interfaces cannot form in 2D but can still form a binding interaction with a Dyn1 molecule recruited from 3D. Although we cannot reject this possibility of inhibited 2D association, it renders the membrane population of Dyn1 effectively useless for cluster formation, which seems wasteful given how efficiently they recruit to cluster sites.

## Discussion

### Distinct recruitment mechanisms of Dyn1 long- and short-tail isoforms

Previous studies have observed the recruitment of dynamin molecules to endocytic sites, and have suggested that this recruitment is essential for both the early formation of CCPs, as well as the final membrane scission process[3,13,49–51]. However, the dynamics of how cytosolic dynamin molecules eventually reach the neck of the nascent endocytic structure remain unclear. Our study challenges the long-standing view that dynamin molecules are recruited directly from the cytosol to the neck of newly forming vesicles (Fig. 8h). It has been previously demonstrated that CCPs can form repeatedly at predefined sites at varying times (hotspots), regulated by accessory proteins like AP2 that serve as nucleation factors[52]. However, the mechanisms by which dynamin is recruited to these pre-defined sites are unclear. Here, we demonstrate that although both short-tail and long-tail Dyn1 isoforms are recruited to the plasma membrane in response to secretagogue stimulation, the modality of their recruitment and their subsequent clustering behavior largely differs. The Dyn1bb/ab short-tail isoforms displayed a bi-phasic recruitment and a significant reduction in their

mobility in response to stimulation, which was associated with their ability to undergo dimensional reduction; to be broadly recruited to the plasma membrane surface, undergo lateral diffusion, and subsequently become actively-confined into discrete nanoclusters within high-density microscale areas on the plasma membrane.

We found that the mechanism underpinning the bi-phasic recruitment and clustering of the short-tail isoforms is regulated by their unique ability to undergo dephosphorylation by calcineurin, as inhibition of calcineurin activity was found to enhance the confinement of these isoforms into nanoclusters, and inhibit the activity-dependent decrease in their single molecule mobility. It is important to note that other recruitment factors, such as membrane curvature, may also play a role in the assembly of the endocytic machinery including dynamin. In the absence of clathrin, the induced membrane curvature is thought to be sufficient to recruit CME proteins and induce endocytosis[53]. In addition, actin also plays a role in dynamin recruitment during CME[49,50]. The formation of dynamin helices also affects dynamin recruitment, which has been shown to be mediated by lipid binding (such as PI(4,5)P_2), and interaction with SH3 containing proteins (such as endophilin and amphiphysin)[7]. These important interactions likely contribute to the mechanism of clustering, oligomerisation, and successful fission.

The speed requirements for ultrafast endocytosis to occur within a millisecond timeframe, have recently been revealed following the discovery of pre-assembled clusters present in neurons at rest[14]. It is interesting to note that pre-clustering of the long-tail isoform (Dyn1aa) is critically involved in this process[14]. In support of this finding, we found that, at rest, the Dyn1aa long-tail isoform is organized in nanoscale clusters (~80 nm radius). Further, upon stimulation, the rate of Dyn1 trajectories detected in clusters largely increases (by 10–25-fold, see Fig. 3d–f) suggesting that pre-organized clusters could potentially recruit additional molecules. Further experiments are needed to assess this possibility. The density of the clusters themselves increases in response to stimulation suggesting that de novo formation of Dyn1aa nanoclusters is also occurring. We therefore suggest that the cluster organization of long-tail Dyn1aa on the plasma membrane could facilitate a fast and efficient oligomerization of nanoscale platforms that provide a ready-to-go status conducive to ultrafast endocytosis.

Importantly, the trapping of newly recruited Dyn1xb molecules within nanoclusters also depends on dynamin GTPase activity suggesting that these clusters are functionally involved in endocytosis. Inhibiting dynamin GTPase activity prevents the activity-dependent mobility reduction showing that the GTPase activity is critical for its lateral trapping. The presence of clusters when tracking endogenous dynamin with DynNB, a nanobody that preferentially binds to the GTP-hydrolytic state of dynamin, further supports this view. We also found that short-tail isoforms are present as pre-assembled clusters on the plasma membrane at rest (Fig. 3 and Supplementary Fig. 4). Short-tail Dyn1 isoforms have been reported to mediate activity-dependent bulk endocytosis[22]. To meet the high demand for bulk membrane retrieval, short-tail Dyn1 isoforms undoubtedly need a robust recruitment mechanism capable of sustaining the fission of large nascent endosomes. Binding to the plasma membrane followed by nanoclustering is a hallmark of dimensional reduction which greatly reduces the concentration requirement and increases the speed of dynamin organization. Our finding of a bi-phasic recruitment allows for large capacity binding to the plasma membrane, followed by lateral trapping into larger and more numerous nanoclusters compared to the long-tail isoform. Our work further explains the need for pre-clustered dynamin on the plasma membrane as it greatly facilitates the subsequent clustering of dynamin short-tail isoforms following their translocation to the plasma membrane. Such clusters have previously been described but were often dismissed as overexpression artifacts[15]. We therefore expanded our investigation into the role of these

pre-assembled clusters by demonstrating that they are also present in MEF and PC12 cells using single-molecule imaging of endogenous dynamin molecules with an anti-dynamin1/2 nanobody[41] expressed as intrabody in these cells[40]. Not only were these pre-assembled clusters present in fibroblasts (MEF cells), but Dyn1bb mobility was reduced in starved cells following serum re-supplementation suggesting a general mechanism by which a high local concentration of dynamin can be achieved through large capacity recruitment to the plasma membrane followed by lateral clustering. This high-capacity mechanism mainly relies on the superior ability of the short-tail isoform to undergo activity-dependent recruitment compared to that of the long-tail isoform. Mathematical modeling revealed that the core function of pre-clustered dynamin is that they allow for the timely enlargement of dynamin nanoclusters to reach the critical concentration threshold required for productive endocytosis to occur in response to stimulation.

The dogma stipulating the direct recruitment of Dyn1 to the neck of nascent endosomes would require a very high cytosolic concentration of Dyn1 to achieve the timely recruitment to these discrete sites on the plasma membrane[54] (Fig. 8h). Our results provide evidence that rather, long- and short-tail isoform Dyn1 molecules are recruited to the plasma membrane and are subsequently trapped in endocytic sites with distinct search patterns. Dimensional reduction is thermodynamically favorable and therefore does not require a high concentration of cytosolic molecules to achieve critical concentration on the plasma membrane. Resident Dyn1 pre-assembled clusters (at rest) could therefore be acting as seeds to further recruit additional molecules (including other Dyn1 isoforms) following their activity-dependent recruitment to the plasma membrane—thus affording fast formation of endocytic vesicles critically needed for maintaining neurotransmission between neurons.

## Methods

### Plasmids and inhibitors
For Dyn1bb-mEos2, Dyn1bb was amplified from pEGFP-N1-hDyn1bb with the following primers: Forward −5′ TCGAATTCTGATGGGCAA CCGCGGC 3′ and Reverse – 5′ GTGGATCCCGGGGGTCACTGATAG 3′. PCR products and pmEos2-N1[23] were digested with EcoRI (New England Bioscience, #R0101) and BamHI (New England Bioscience, #R0136), gel extracted and ligated with T4 ligase (New England Bioscience, #M0202) to make pmEos2-N1-hDyn1bb as similarly generated[55].

pEGFP-N1-hDyn1bb, pmCerulean-N1-hDyn1aa, and pmCerulean-N1-hDyn1ab plasmids were a generous gift from Phil Robinson[22,56]. pEGFP-N1-DynNB was a generous gift from Aurélien Roux[41]. pmCerulean-N1-hDyn1(aa or ab), pEGFP-N1-DynNB, and pmEos2-N1 were digested with BglII (New England Bioscience, #R0144) and EcoRI (New England Bioscience, #R0101), gel extracted and ligated with T4 ligase (New England Bioscience, #M0202) to make pmEos2-N1-hDyn1 (aa and ab) and pmEos2-N1-DynNB. To generate Dyn1ab-EGFP and Dyn1aa-EGFP, Dyn1 were created as above, but T4 ligation with pEGFP-N1 backbone (Clontec). Positive clones were verified by sanger sequencing at the Australian Genome Resource Facility (AGRF, Brisbane).

AP2-GFP was a generous gift from Dr Giuseppe Balistreri. Dynamin inhibitor Dyngo-4a (30 μM, 30 min) was a generous gift from Prof Phil Robinson[37]. Staurosporine (Sigma-Aldrich, #S4400) and okadaic acid (Sigma-Aldrich, #O9381) were used as inhibitors for kinase and phosphatase activity respectively (1 μM, 30 min for both). Cyclosporin-A (Sigma-Aldrich, #30024) was used as a calcineurin inhibitor (40 μM, 10 min). Transferrin-Alexa Fluor™ 647 (Invitrogen™, #T23366, 10 μg/ml, 5 min).

### MEF cell 4OH-TMX treatment and transfection
Dyn1 and 2 conditional double knockout mouse embryonic fibroblasts cells (MEF[Dyn1,2] DKO)[39] were a generous gift from Dr Giuseppe Balistreri.

Cells were maintained in Dulbecco's Modified Eagle Medium (DMEM) (Life technologies Gibco, #11995-065) containing 10% heat inactivated Fetal Bovine Serum (FBS) (Bovogen, #SFBS-HI) and 1% GlutaMax 100 X (Life technologies Gibco, #35050-061), at 37 °C in 5% $CO_2$. To achieve the Dyn1 and 2 conditional double knockout, 3 μM of 4OH-TMX (Sigma-Aldrich, #H7904-5MG) was applied to the media of MEF[Dyn1,2] DKO cells on day 1 upon splitting (1:4); media was then replaced on day 2 with fresh media containing 3 μM 4OH-TMX. On day 3, 0.3 μM of 4OH-TMX within fresh media was applied, and this concentration was kept until the MEF cells were ready to be used in experiments (normally from day 6 to day 9).

Electroporation was used to transfect MEF cells with Dyn1bb-mEos2. The Amaxa NHDF Nucleofector Kit (Lonza, #VPD-1001) and Amaxa Nucleofector II (Lonza AAB-1001) electroporation machine was used for the electroporation, for which Lonza's Optimized Protocols for Amaxa Nucleofector Technology were followed. On day 4, $2 \times 10^6$ MEF cells and 100 μl Nucleofector solution were placed into an Amaxa certified cuvette and transfected in the Amaxa Nucleofector II using the T-20 program. Immediately after electroporation, cells were transported to pre-warmed MEF media containing 0.3 μM 4OH-TMX, and subsequently replated evenly into 10 glass-bottom dishes (Cellvis, CA, USA, #D29-20-1.5N) that were pre-treated with 1 μg/ml of Fibronectin (Merck, #F0635). Imaging was performed from day 6 to day 9. During 4OH-TMX treatment, MEF cells were maintained as normal and passaged when they reached 80–90% confluency.

### PC12 cell culture and transfection
PC12 cells were cultured in DMEM (Gibco, Life Technologies, #11995-065) supplemented with 5% heat-inactivated FBS (Bovogen, #SFBS-HI), 5% heat-inactivated horse serum (Gibco, Life Technologies, #26050088) and 0.5% GlutaMAX (Gibco, Life Technologies #35050061); the cells were maintained at 37 °C in 5% $CO_2$. $4.5 \times 10^5$ cells in 6 well plate (or 3.5 cm dish) were plated a day before transfection (~60–80% confluent). Transfections were performed using Lipofectamine LTX and Plus Reagent (Thermo Fisher Scientific, #A12621) as per the manufacturer's instructions. In brief, transfection was prepared in Optimem (serum-reduced medium) containing 1–3 μg of Plasmid DNA, 1 μl/μg DNA Plus Reagent and 6.75 μl LTX for each 35 mm dish. Twenty-four hours after transfection, cells were re-plated into poly-D-lysine (PDL, Sigma-Aldrich, #P7886-100MG) coated glass-bottomed culture dishes (Cellvis, CA, USA, #D29-20-1.5N). Live-cell imaging was done on day 3–4 post-transfection.

### Primary hippocampal neuron culture and transfection
Embryonic day 18 (E18) embryos were removed from euthanised C57BL/6J mice and dissected for hippocampal neurons as previously described[57]. A total of $1 \times 10^5$ neurons were seeded within the 29 mm glass bottom dishes (Cellvis, CA, USA, #D29-20-1.5N) coated with poly-L-lysine (PLL, Sigma-Aldrich, #P2636-100MG). Following incubation (37 °C and 5% $CO_2$ for 2 h), plating media was replaced with culturing media (100 U/ml penicillin μg/ml streptomycin, 1X GlutaMAX supplement, 1X B27 in neurobasal media). Hippocampal neurons were transfected in 14–16 days-in-vitro (DIV 14–16) with Dyn1aa-mEos2 or Dyn1bb-mEos2 using Lipofectamine 2000 (Thermo Fisher Scientific, #11668019), following the manufacturer's instructions. In brief, transfection was prepared in Neurobasal medium with either 1–3 μg of Plasmid DNA, or 1 μl/μg Lipofectatmine2000 was used per 35 mm dish. Imaging was done in DIV 18–20.

### Time-lapse TIRF microscopy
Transfected PC12 cells were bathed in Buffer A (145 mM NaCl, 5 mM KCl, 1.2 mM $Na_2HPO_4$, 10 mM D-glucose and 20 mM HEPES, pH 7.4). The cells were then visualized using an inverted Roper Scientific TIRF microscope equipped with a perfect focus system and an iLas2 double-laser illuminator (Roper Scientific). The microscope was fitted with a

Nikon CFI Apo TIRF × 100 (1.49 NA) oil objective (Nikon Instruments) and an Evolve 512 delta EMCCD camera (Photometrics). Metamorph software was used for movie acquisition (Metamorph 7.7.8, Molecular Devices) at 50 Hz with 16,000 frames acquired for each cell kept at 37 °C. A 491 nm laser was used to photoactivate the cells expressing Dyn1aa-GFP, Dyn1ab-GFP, and Dyn1bb-GFP in both control (before) and 2 mM $Ba^{2+}$ (during) stimulation conditions. TIRF angle was calibrated each imaging session and TIRF critical angle was ~70°. PC12 cells were selected based on cell morphology (proper attachment to the cover-glass and presence of filipodia).

The LFA refers to areas with comparatively less Dyn1-GFP fluorescence observed throughout the acquisition period, compared to that of HFA which exhibited high intensity Dyn1-GFP. Since the distribution of HFAs and LFAs varied dynamically, ROIs of equal size were meticulously chosen for each movie to ensure they were within either HFAs or LFAs throughout the duration of the acquisition.

### SptPALM

Time-lapse TIRF imaging of live MEF cells, PC12 cells and mouse primary hippocampal neurons was conducted on a TIRF microscope (Roper Technologies) equipped with an iLas2 double-laser illuminator (Roper Technologies), a CFI Apo TIRF 100× (1.49-NA) objective (Nikon), and two Evolve512 delta EMCCD cameras (Photometrics). Image acquisition was performed using Metamorph software (version 7.7.8; Molecular Devices). Cells were bathed at 37 °C in Buffer A (145 mM NaCl, 5 mM KCl, 1.2 mM $Na_2HPO_4$, 10 mM D-glucose, and 20 mM HEPES, pH 7.4). Single molecule movies were captured at 50 Hz (16,000 frames by image streaming) and 20 ms exposure. A quadruple beam splitter (LF 405/488/561/635-A-000-ZHE; Semrock) and a QUAD band emitter (FF01-446/510/581/703-25; Semrock) were used to isolate the mEos2 signal from autofluorescence and background noise signals. A 405 nm laser (0.005 W/cm$^2$ at sample) was used simultaneously with a 561 nm laser (82 W/cm$^2$ at sample) to photoconvert and excite the photoconverted molecules, respectively. MEF cells were imaged in both serum-starved conditions (control) and serum re-supplementation conditions (stimulation). PC12 cells were imaged in Buffer A (control), and following the addition of 2 mM $Ba^{2+}$ (stimulation). Presynaptic boutons from well-transfected mature hippocampal neurons were chosen for imaging. During the imaging experiments, hippocampal neurons were maintained in low $K^+$ solution and stimulated with either extracellular high $K^+$ or field stimulation electrodes (RC-21BRFS; Warner Instruments, Holliston, MA). Neurons were imaged before and immediately after the addition of high $K^+$ solution. Alternatively, to electrically induce compensatory endocytosis, neurons were stimulated with 1 ms pulses delivered at 10 Hz for 30 s (100 mA) and imaged before (Ctrl) and during/following (10 Hz 30 s) electrical stimulation.

For dual-color imaging, PC12 cells were co-transfected with AP2-GFP and Dyn1bb-mEos2, and time-lapse TIRF live imaging was conducted as above, with the exception of an additional camera to separate green (491 nm) and red (561 nm) channels for simultaneous acquisition. AP2-GFP was imaged simultaneously with Dyn1bb-mEos2 for the entire acquisition (320 s). The acquired movies were split into green (AP2-GFP) and red (Dyn1bb-mEos2) emission channels. An average z-projection of the AP2-GFP movie was created to identify regions of AP2 clusters. AP2-positive (Green clusters) and negative (no GFP signal) ROIs of the same size were selected. Those ROIs were then used to perform single-particle tracking on Dyn1 molecules deemed "positive" or "negative" AP2 areas.

### Tracking

Information regarding the localization and dynamics of Dyn1-mEos2 single molecules were extracted from the 16,000 frames TIRF movies as previously described[58]. Wavelet segmentation was used to detect and track single molecules, and simulated annealing was used to optimize multi-frame object correspondence. A custom-written program for Metamorph software (Molecular Devices) named PALM-Tracer[59], was used to localize and track the dynamics of Dyn1-mEos2 single molecules. The diffusion coefficient (D) distribution was sorted into two groups, named mobile and immobile as previously reported[26]. Analysis was done as previously described[23]. All tracks shorter than 8 frames were excluded from the tracking process to minimize non-specific background. ImageJ (2.0.0-rc-43/1.50e; National Institutes of Health) was used for the color-coding of the super-resolved images. The analyzed data from PALM-Tracer was then filtered and analyzed using QBI SPT Auto Analysis software (v.2.1.2) to generate the average MSD, AUC, $Log_{10}D$, and mobile-to-immobile fraction, https://github.com/QBI-Software/AutoAnalysis_SPT.

### Fluorescence intensity analysis

The Fluorescence intensity (FI) analysis was performed in Fiji (ImageJ) (2.0.0-rc-43/1.50e; National Institutes of Health) by plotting the Z-axis profiles of each movie using either the whole cell, high FI area (HFA), or low FI area (LFA), as the regions of interest (ROIs).

### NASTIC analysis

NASTIC analysis was performed as previously described[31]. The parameters used in this study were: for PC12 and MEF cells: Acquisition time (s) = 320, Frame time (s) = 0.02, Time threshold (s) = 20, Radius factor = 1.2, Cluster threshold = 3, Cluster size screen (μm) = 0.15, MSD screen = off.

### Analysis of two-dimensional sampling followed by nanocluster confinement

A 3-Dimensional (spatial and temporal) Density-Based Spatial Clustering of Applications with Noise (3D DBSCAN) algorithm was utilized to obtain spatiotemporal nanoclusters (BOOSH; GitHub: https://github.com/tristanwallis/smlm_clustering/blob/main/boosh_gui.py) due to its ability to gain more fine-grained clustering information in comparison to NASTIC. These spatiotemporal nanoclusters were obtained using the following parameters: for hippocampal neurons: minimum trajectory length = 8, maximum trajectory length = 100, acquisition time = 60 s, frame time = 0.02 s, epsilon = 0.05 μm, minimum points = 8, time window = 10 s, cluster size screen = 0.3 μm, MSD screen = off; for PC12 cells: minimum trajectory length = 8, maximum trajectory length = 100, acquisition time = 320 s, frame time = 0.02 s, epsilon = 0.05 μm, minimum points = 3, time window = 10 s, cluster size screen = 0.15 μm, MSD screen = off. PC12 cells with single-particle trajectories greater than 50 trajectories/μm$^2$ (higher detection densities induce mistracking), or with significant drift during acquisition were rejected for analysis. Custom-written python scripts were developed to obtain trajectories which enter spatiotemporal nanoclusters during their lifetime. In brief, the core function loops through each spatiotemporal nanocluster to obtain trajectories that are accordant with the following: (1) the first detection of the trajectory is not within the spatiotemporal cluster, (2) the last detection of the trajectory is within the spatiotemporal cluster, and (3) the trajectory does not intercept the boundary of the spatiotemporal cluster prior to its step of entry. Each frame-to-frame displacement (with respect to entry step) from all such trajectories were pooled in a single group for calculation of cell means. For before cluster entry versus within cluster, all steps from all such trajectories were pooled in a single group for calculation of cell means. Statistical tests and sample sizes are included within the figure legend.

### Analysis of single-particle tracking rate of cumulative count

Cumulative counts of trajectories were calculated per μm$^2$ of cell-surface (determined by hand-drawn region of interest). Rates of increase in the cumulative counts were calculated via chunking the

cumulative counts into blocks of 30 s to avoid under-sampling, followed by use of the gradient function of NumPy.

## Dyn1 and transferrin co-localization analysis

PC12 cells were plated in 3.5 cm dishes with poly-D-lysine (PDL, Sigma-Aldrich, #P7886-100MG, 0.1 mg/ml) coated glass coverslips, and transfected with Dyn1aa-GFP, or Dyn1ab-GFP, or Dyn1bb-GFP as described above. For control conditions, transfected PC12 cells were incubated with 10 μg/ml Tf-647 for 5 min. For stimulated conditions, transfected PC12 cells were incubated with 10 μg/ml Tf-647 for 5 min total, with 2mM Ba$^{2+}$ applied at 3 min. PC12 cells were fixed at 5 min with 4% PFA for 20 min, washed with 1xPBS 3 times, and mounted on glass slides for imaging. Photomicrographs were acquired at a resolution of $20.313 \times 20.313 \times 260$ nm, using an Olympus UPLXAPO 100x/1.45 NA oil-immersion objective, a SoRa disk and 3.2x magnifier super-resolution configuration on a spinning disk confocal microscope (SpinSR10; Olympus, Japan) built around an Olympus IX3 body and equipped with two ORCA-Fusion BT sCMOS cameras (Hamamatsu Photonics K.K., Japan) and controlled by Olympus cellSens software. Images were deconvolved using Huygens Professional version 22.10 (Scientific Volume Imaging, The Netherlands) using the CLME algorithm with manual background value of 100 for Dyn1-GFP and 10 for Tf-647. Images were analyzed using Imaris Image Analysis Software version 10.1. Intensity-based co-localization analysis was performed on images of Dyn1 and Tf, and a co-localization channel built. A surface was created of the co-localized channel to obtain the area of Dyn1 and Tf co-localization for each cell.

## Model description

**Model reaction network.** We track 6 species in this model. Three unbound species are: $D_{sol}$: dynamin in solution, $R_{mem}$: a membrane recruiter uniformly distributed on the membrane, and $A_{clus}$: an "activating" recruiter that is localized to the membrane cluster region. The activating recruiter represents proteins that have already nucleated endocytic sites, and thus can now recruit dynamin to directionally assemble, which we treat as irreversible binding. Three membrane-bound dynamin species are: 1) $D_{mem}$: dynamin localized to the membrane after binding the recruiter $R_{mem}$, 2) $D_{clus}$: dynamin localized to the cluster site and arriving via 3D diffusion, and 3) $D_{2dclus}$: dynamin localized to the cluster site and arriving via 2D diffusion. We treat $D_{clus}$ and $D_{2dclus}$ as distinct species only to be able to differentiate mechanisms of arrival at the membrane; they are otherwise identical.

We initialize the densities of species $D_{sol}$, $R_{mem}$, and $D_{mem}$ to be at an equilibrium steady-state by choosing the off-rate $k_b$ such that $\frac{[D]_{sol,0}[R]_{mem,0}}{[D]_{mem,0}} = \frac{k_b}{k_{frev}}$. As a result, the change in dynamin density on the membrane is driven entirely by the appearance of a cluster with a non-zero $A_{clus}$ density. If no cluster appears, no changes in copies occur. The 7 reactions with rates indicated are:

$$D_{sol} + R_{mem} \rightleftharpoons D_{mem} : k_{frev}, k_b \tag{1}$$

$$D_{mem} + A_{clus} \rightarrow D_{2dclus} : k_{mem}^{2D} \tag{2}$$

$$D_{sol} + A_{clus} \rightarrow D_{clus} : k_{mem} \tag{3}$$

$$D_{mem} + D_{2dclus} \rightarrow 2D_{2dclus} : k_{dyn}^{2D} \tag{4}$$

$$D_{sol} + D_{clus} \rightarrow 2D_{clus} : k_{dyn} \tag{5}$$

$$D_{mem} + D_{clus} \rightarrow D_{2dclus} + D_{clus} : k_{dyn}^{2D} \tag{6}$$

$$D_{sol} + D_{2dclus} \rightarrow D_{2dclus} + D_{clus} : k_{dyn} \tag{7}$$

Reactions 1–5 are illustrated in Fig. 8a, where reactions 6–7 are the same as 4–5 and are necessary because we separately track dynamin in the cluster from 3D and 2D. Reactions 2, 4, and 6 are purely in 2D, and the rates are thus given by $k_{dyn}^{2D} = k_{dyn}/h$, $k_{mem}^{2D} = k_{mem}/h$, where $h$ is a nanoscopic length scale that controls the conversion from 3D to 2D rate constants. This length scale can be thought of as approximately the height of space occupied by the molecules at the surface, and is necessary to describe reaction dynamics restricted to 2D, as explored in previous work[15,48]. Time-resolved solutions are shown in Supplementary Fig. 8.

All code and model inputs are openly available in the github repository, along with parameter sets and representative movies: https://github.com/mjohn218/dynamin_model.

**Changes to the model parameters following stimulation.** Following stimulation, we keep as many parameters as possible identical to prior. To mimic the changes that occur experimentally following stimulation, we make 3 primary changes. First, the density of clusters has increased, and thus the occupied area/cluster is lower (Supplementary Table 2). Second, the initial density of dynamin on the membrane has increased as measured by single-particle tracking (SPT) (see Supplementary Table 2). We increase the pre-stimulation density by this factor and thus correspondingly decrease the amount of solution dynamin, such that total dynamin copies are preserved. Third, we allow the rate of dynamin binding to the activator to change, to mimic any changes in the kinetics of assembling and recruiting to these nucleation sites. Hence there is a value of $k_{memPre}$ and another value of $k_{memPost}$. The $k_{dyn}$ and $k_{frev}$ are unchanged, and we keep the density of the (unbound) $R_{mem}$ and initial $A_{clus}$ the same as pre-stimulation.

**Continuum reaction-diffusion simulations.** The deterministic reaction-diffusion model was simulated using Virtual Cell[15]. This PDE solver used a fully implicit finite volume, regular grid, with a variable time step that was maximally 0.1 s. Absolute error tol: $10^{-9}$, relative error: $10^{-7}$. The mesh used default values, with side lengths of 0.015 μm in x/y and 0.04 μm in z. The system was initialized using two geometries. The first geometry was based on an experimental cluster density for Dyn1bb of 0.54 clusters/μm$^2$. For regularly spaced clusters, this results in each cluster within a 2D space of $1.36 \times 1.36$ μm, producing a membrane area of 1.85 μm$^2$. The cluster was placed in the center of this square and modeled as a disk of radius 0.083 μm consistent with experiment. The height of the simulation volume was chosen as 1.9 μm, to produce a V/A ratio of 1.9 μm which is representative of a mammalian cell[15]. The second geometry was based on 3 clusters/μm$^2$, having a 2D space of $0.568 \times 0.568$ μm and a cluster having a radius of 0.096 μm.

Dynamin in solution $D_{sol}$ was initialized as well-mixed throughout the volume. It was given a diffusion coefficient of 7 μm$^2$/s, which is a value assuming a hydrodynamic radius of 10 nm and a factor of 3 slow-down relative to pure water. Dynamin on the membrane $D_{mem}$ was initialized as well-mixed across the surface, including within the cluster due to the initial intensity of dynamin being essentially uniform on the surface prior to increases at the cluster site, with a diffusion coefficient of 0.045 or 0.02 μm$^2$/s (Supplementary Table 2). The dilute recruiter $R_{mem}$ was initialized as well-mixed on the surface, outside of the cluster site, also diffusing at the same rate as dynamin. The activating recruiter $A_{clus}$ was initialized as within the cluster site only, and non-diffusing.

Aside from the discrete localization of $A_{clus}$, $D_{clus}$, and $D_{2dclus}$ being restricted only to within the cluster, the change in dynamin density on the membrane surface over the seconds time-scale showed

some spatial variation, but it was a relatively minor gradient at the perimeter of the cluster, as visible in Fig. 8b, c. This is due to the sufficiently fast 2D diffusion on the surface over the length-scales needed to reach and grow the cluster site. Also, the binding rates were not diffusion-limited (i.e., they were ≪ $10^9$ M$^{-1}$s$^{-1}$). These results are consistent with the recently quantified spatial dependence of dimerization kinetics in 2D vs. 3D[48]. The density of $D_{mem}$ that was within the cluster boundaries did drop compared to outside of the cluster, and this spatial variation was accounted for in quantifying the total change in dynamin density within the cluster. We note that although the spatial model did have slightly slower kinetics of recruitment to the clusters at later times (Fig. 8d), this could be "corrected" by using the same model parameters from the non-spatial ODE model except for a higher value of $k_{Dydy}$, which is largely responsible for the steepness of growth at the later times.

**Stochastic structure-resolved reaction-diffusion simulations.** We also performed structure-resolved stochastic reaction-diffusion simulations using NERDSS software[47]. Briefly, each molecule has coordinates in 3D (solution) or 2D (membrane) space that propagate in space and time according to Brownian motion. Each dynamin has 6 interfaces, one to bind the membrane recruiter, one the cluster activator, and four sites to bind up to four additional dynamins. When two reaction partners (e.g., dynamin and the cluster activator) collide, they can react with a probability controlled by the reaction rate. NERDSS software has been extensively validated for reaction kinetics in 3D, 2D, and between 3D to 2D, producing the exact association kinetics for a reaction-diffusion system of volume excluding particles.

Aside from giving each component specific interfaces to mediate reactions, and initializing species randomly either in 3D or 2D, all the parameters are the same as for the continuum PDE reaction-diffusion model except two modifications to rates. The reactions 4 and 5 involve binding between two dynamin to convert one inactive dynamin to an activated dynamin, producing two activated dynamin available for further binding events. This reaction is not straightforward to implement in NERDSS. A natural mimic, however, is to allow this same forward reaction, but the product is a bound dimer of dynamin. For this bound dimer to be able to continue activating further dynamin, they both must have additional free interfaces for association with additional dynamin. Hence we give each dynamin 4 independent binding sites that can react with dynamin (once activated) with a rate $k_{dyn}/4$. The ensures a similar effective rate between a $D_{mem} + D_{2dclus}$ as used in the PDE solver, and it allows the dimer of dynamin to continue growing into a higher-order oligomer. This one modification was sufficient to reproduce the pre-stimulation kinetics, but the post-stimulation kinetics were a bit too slow. We found this was due to the slower creation of the pool of activated dynamin in the stochastic simulations, most likely because the continuum models can amplify from fractional concentrations whereas the stochastic simulations require two copies encounter one another. We therefore increased the rate of $k_{memPost}$ by 3-fold higher. Physically this indicates that following stimulation, dynamin is more rapidly binding to molecules within the cluster, although the rate is modest at $0.006\,\mu$M$^{-1}$s$^{-1}$. Lastly, to avoid simulating a large solution volume with thousands of molecules, we truncate the height of the simulation volume. To retain the same ratio of volume to surface molecules, we keep the solution concentration at the fixed initial concentration of $10\,\mu$M via a 0th order titration reaction at $100\,\mu$M/s and a 1st order degradation reaction at $10\,$s$^{-1}$ as was done in work on clathrin assembly[60]. This assumes that the clustering on the membrane does not appreciably change the solution concentration, which is accurate for these systems.

**Concentration and kinetic parameters.** Several parameters in the model are not strictly constrained by the experiment, so we allow their initial values to vary over reasonable ranges as we optimize the model against experiments. We allow $0.1 < [D]_{sol} < 10\,\mu$M and $10 < [D]_{mem} < 80$ copies/$\mu$m$^2$ (Supplementary Table 3). The density range of membrane bound dynamin are chosen such that the copies within the cluster are in the range of 0.2–2 copies prior to clustering, and that once the cluster forms, the copies rise according to the single-particle tracks by 20 to 40-fold (Fig. 3d–f). Thus, the dynamin copies within the clusters is ~5 to 80 copies at peak, which is the order-of-magnitude of copies needed to form a constriction filament; we find parameter sets including the most likely 20–40 copy range. For reference, the lipid PI(4,5)$P_2$ has a density of ~20,000 copies/$\mu$m$^{2,15}$. Total Dynamin is thus given by $D_{totCopies} = V[D]_{sol} + A[D]_{mem}$. So even for the lowest solution concentrations of Dynamin, at least 2x more copies are in solution than on the surface. Rates are chosen to mimic reasonable ranges for protein-protein interactions, and the length scale $h$ is set to 10 nm, which is comparable to a size of a dynamin molecule and the expected magnitude for converting a 3D rate to a 2D rate constant[15].

**Corresponding ODE model.** We implemented a non-spatial version of our model. Because of the ability of components to transition from 3D volume to a 2D surface, we solve for all species in units of $\mu$M, which requires that all 2D reactions are then multiplied by a dimensionality factor $DF = V/(Ah)$[15]. The initial densities of membrane bound species were first converted to copy numbers by multiplying by either the total membrane area or the cluster area, and then converted to concentrations via the system volume. The ODE model produces similar kinetics as the spatial model, which is reflective of the rate-limited reaction-rates and the relatively short length scales that the molecules have to traverse to reach the cluster site over the simulated timescales. The ODE model does not capture any decrease in $[D]_{mem}$ that occurs locally within the cluster, and that contributes to some difference in the relative intensity from the ODE relative to the spatial model. The ODEs were solved using built-in functions for stiff systems in MATLAB and Python.

**Optimization of model parameters to experiment.** The model is constrained by several experimental observables (Supplementary Table 2). First, the kinetics of the fluorescence within forming clusters (from Fig. 1j, k and fitted in Supplementary Fig. 9). Second, the relative increase in densities of dynamin in the cluster before it forms and at its peak. Third, the relative increase in dynamin density on the membrane following stimulation. Fourth, the density of clusters formed on the membrane, both before and following stimulation. And fifth, the diffusion constants of dynamin on the membrane.

We quantified the relative increase in dynamin density within the cluster using: Rel Density$(t) = \frac{[D]_{mem}(t) + [D]_{clus}(t) + [D]_{2dclus}(t)}{[D]_{mem}(0)}$. Alternatively, we measure the increase in dynamin copies via Dyn_copies$(t) = ([D]_{mem}(t) + [D]_{clus}(t) + [D]_{2dclus}(t))*A_{clus}$, where $A_{clus} = \pi R_{clus}^2$, and the concentrations here have been converted to per area. The experimental measurement is Rel Intensity$(t) = \frac{I_{cluster}(t) - \Delta}{I_{cluster}(0) - \Delta}$. $I_{cluster}$ is the measured fluorescence intensity in arbitrary units, and $\Delta$ would be an offset due to background.

Here we choose $\Delta$ such that the increase in relative intensity matches the increase in single-particle tracks measured within clusters before and during their appearance for each isoform. We note that the experimental kinetic data of Cluster Intensity vs. time is similar for distinct isoforms, so we use the same growth shape vs. time for all isoforms to fit our parameters. However, the relative scale of density increase and the starting density are isoform-dependent.

We optimized the model parameters to minimize the $\chi^2$ distance between the simulated and observed relative intensity, weighted by the measured experimental errors at each time-point. The simulated intensity was solved from the ODE version of the model. To find optimal parameter sets over the ranges defined previously (Supplementary Table 3), we performed stochastic sampling of parameter sets via a genetic algorithm implemented in python via the DEAP package[15].

## Statistical analysis

Statistical tests were performed as described in the corresponding figure legends. Data was first analyzed to determine its distribution; Normalized data was analyzed using a Student's $t$ test (of paired Student's $t$ test for samples analyzed before and after stimulation). Non-normally distributed data was analyzed by Mann–Whitney or Wilcoxon test (paired data). Data were considered significant if they had a $p$ value < 0.05. Data are shown as mean ± SEM unless otherwise stated.

## Reporting summary

Further information on research design is available in the Nature Portfolio Reporting Summary linked to this article.

## Data availability

Raw data generated in this study are available for downloading from the publicly accessible institutional data repository of The University of Queensland (UQ eSpace): https://espace.library.uq.edu.au/view/UQ:5d9371d. Source data are provided with this paper.

## Code availability

The work of NASTIC analysis have been described by Wallis et al.[31]. All Python codes are available at the Github repository https://github.com/tristanwallis/smlm_clustering.

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

## Acknowledgements

We thank Dr Rumelo Amor and his team at Queensland Brain Institute's (QBI's) Advanced Microscopy Facility for their assistance with imaging and image analysis. We thank Dr JB Sibarita for providing the PALM-tracer analysis software. We thank Dr Alex McCann for critical appraisal of the manuscript. We thank Dr Jing Xue and Dr David Cardoso for providing the Dyn1 plasmids. We thank Prof Aurélien Roux for providing the DynNB plasmid. We thank Dr Giuseppe Balistreri for providing the AP2-GFP plasmid and the MEF^Dyn1,2 DKO cell line. We thank Dr Christopher Small, Dr Barbara Duda, and Anusha Malapaka for the mouse hippocampal neuron dissections. This work was also supported by an Australian Research Council (ARC) Discovery Project Grant (DP190100647), ARC LIEF Grant (LE130100078) and The National Health and Medical Research Council (NHMRC) Senior Research Fellowship (1155794) to F.A.M.. M.J. is supported by an ARC Discovery Early Career Researcher Award (DE190100565) and The University of Queensland Early Career Researcher Grant (UQECR2057309). A.J. is supported by the Research Training Program (RTP) Scholarship and QBI top-up Scholarship. M.E.J. gratefully acknowledges funding from a US National Institutes of Health MIRA Award R35GM133644.

## Author contributions

F.A.M. conceived the project with the help of P.J.R., M.E.J. and M.J.. A.J., F.A.M. and M.J. planned the experiments. A.J. performed all tissue culture, transfection, TIRF recruitment and sptPALM experiments including dual-color imaging. A.J., T.P.W. and K.K. performed spatiotemporal cluster analysis. K.K. performed all lateral trapping analyses. A.J. and R.S.G. carried out the cloning. S.E. performed the Transferrin imaging and colocalization analysis. S.F.L. set up the electrical stimulation paradigm. M.E.J. and S.G. performed the modeling. F.A.M. and A.J. wrote and edited the manuscript with the help of all authors.

## Competing interests

The authors declare no competing interests.
