## [Peer Review File · Nature Communications]

Dynamin1 long- and short-tail isoforms exploit distinct recruitment and spatial patterns to form endocytic nanoclustersREVIEWER COMMENTS

Reviewer #1 (Remarks to the Author):

In this manuscript, Jiang, et al., explored single-particle behaviors of three splice variants of Dynamin 1 (aa, ab, and bb) in PC12 cells, MEF, and mouse hippocampal neurons. Authors demonstrated that ab and bb variants are recruited to the plasma membrane and then confined to smaller areas in conditions that induce secretion and compensatory endocytosis. Single-particle analysis shows a marked decrease in diffusion kinetics of ab and bb, but not aa, molecules upon stimulus. This change is controlled by the phosphorylation status of ab and bb molecules. These data along with computer simulations are taken to suggest that ab and bb molecules have biphasic behaviors during endocytosis: recruitment and lateral diffusion/trapping. The data are of high quality. However, there are several concerns.

Major:

1. Molecular recruitment during endocytosis.

It is not clear how the observed biphasic behaviors of Dynamin 1 ab and bb relate to actual endocytic events. The authors should consider adding pHluorin-based assays or transferrin uptake assay to demonstrate that they are looking at molecular behaviors around bona fide endocytic pits.

2. Novelty

The puncta formation during endocytosis is well-known phenomenon (Merrifield et al. 2003 PMID: 12198492) (Taylor et al. 2011 PMID: 21445324) (Grassart et al. 2014 PMID: 24891602) (Srinivasan et al 2018 PMID: 29668686). Several recent studies also show similar results (Cocucci et al. PMID: 25232009) and even suggest that trapping of dynamin within puncta allows further recruitment of molecules at endocytic pits (Claverie et al. 2019 doi:10.1017/S1431927619006937). Authors may consider measuring the diffusion coefficient at the endocytic pits with finer ROIs and see if there are two populations of diffusional states of molecules (scaffolding vs diffusion). Authors should also address if dynamin become dissociated from endocytic pits after endocytosis is complete (Taylor et al. 2011 PMID: 21445324).

3. Lateral diffusion of Dyn1xb in PC12 cells.

In Figure 1, authors suggested that Dynamin 1ab and bb molecules translocate from low to high fluorescence intensity area. Authors took these data to suggest lateral diffusion, but it is equally possible that molecules are dissociated from the plasma membrane and re-recruited to the plasma membrane (Taylor et al. 2011 PMID: 21445324) (Cocucci et al. PMID: 25232009). More convincing data are necessary.

3. Differences between Dyn1 variants in PC12 cells.

Based on Figure 1d, there is a marked difference between Dynamin 1bb and Dynamin 1ab: ab seems recruited on the entire surface. The decrease in MSD may simply come from slower diffusion coefficient of ab. Please clarify this point.

4. Endogenous Dyn1.

Authors tried to address potential artifacts by overexpression using nanobody against GTP-loaded dynamins, dynab (Galli et al 2017 eLife). However, dynab recognizes both Dynamin 1 and 2 (also mentioned in Line 344). Thus, it is uncertain if this experiment has addressed the issue.

5. Localization of Dyn1xb in cultured neurons.

For neurons, it is not clear if molecules imaged are within presynapses or not. Presynaptic markers would be helpful.

6. Recruitment of additional molecules at pre-organized clusters.

Authors claim that pre-organized clusters of Dyn1aa recruit additional molecules (line 444-448). However, the current data are not convincing. For this claim, it requires simultaneous imaging of Dyn1aa and xb in PC12 cells and in neurons.

7. Experimental procedures are all too brief.

Please describe experiment in more detail. When did authors stimulate cells? What criteria were used to select ROIs? How is MSD calculated within and outside of AP2 puncta? Was AP2 imaged once or simultaneously with Dynamin? How were HFA and LFA defined? What was the density of neurons/cells? How were they transfected? How did the TIRM imaging performed on neurons when synapses are typically found outside the evanescent field? What was the exit angle of the laser and estimated d value? These are just a few examples.

Minor:

1. In line 406-409, authors claim that dynamin molecules on membrane are recruited to the clusters via oligomerization. This is based on the results from Dyngo-4a experiments in Figure 6. However, it is not clear if Dyngo-4a inhibit oligomerization of dynamin in the cells based on one of authors previous work (McCluskey et al. 2013, PMID: 24025110)

2. Please add a reference for line 474-476.

Reviewer #2 (Remarks to the Author):

In the manuscript titled “Dynamin1 long- and short-tail isoforms exploit distinct recruitment and spatial patterns to form endocytic nanoclusters,” the authors have explored the activity-dependent formation of Dynamin 1 (Dyn 1) nanoclusters at the plasma membrane of various cell types using SMLM techniques and have identified the mechanisms of different Dynamin 1 isoform recruitment, which differ in their C-terminal PRD. Overall the manuscript elaborates on how dynamin recruitment may match the timescale of endocytosis, borrowing ideas from thermodynamically favorable information compression through dimensional reduction, which is insightful and compelling. However, the manuscript needs further improvements detailed below.

Major concerns:

1. In the abstract, the authors concluded that “we demonstrate that short-tail Dyn1 isoforms Dyn1ab and Dyn1bb display an activity-dependent recruitment to the membrane, promptly followed by concentration into nanoclusters.” This can be better viewed in Fig. 1c-f. The authors explain the decline of fluorescence in LFA as the source for the increase of fluorescence in HFA, and thus interpret the result as dynamin movement from LFA to HFA. However, it could be possible that dynamin in LFA falls off the membrane whereas dynamin in HFA continues to remain as clusters and increase in intensity. Can the authors exclude this possibility?
2. In figures 2a and b, it is surprising that even though the authors detected such a large number of tracks, the MSD increased almost linearly with time under both stimulated and unstimulated conditions. Had it been active confinement, the stimulated curve should have reached a plateau ideally. The reduced slope of the red curve means that the molecules are in general moving slowly, which could be simply because of crowding than active confinement to the pre-assembled clusters. The authors should plot the distribution of the power factor or the scaling factor of time (α) to verify confinement.
3. Interestingly, the authors analyzed the time scale of these clusters. I am curious, what is the average time length through which they can follow the same track? Can the authors calculate the average fallout rate of these tracks, which should be less if the authors compare it to another molecule that moves in 3D? If they follow a single track long enough, do they see ‘cage hopping’ phenomena i.e. repeated confinement and release from clusters? Does the average tracking time for individual tracks and their fall-out rate change with stimulation?
4. Why does the cluster radius reduce on stimulation of hippocampal neurons while it increases in PC12 cells? Probably, KCl-mediated stimulation is a harsh method of neuronal stimulation, and the authors may want to use action potential stimulation. In this context, if the cluster parameters don’t change but the MSD values of a single molecule track slow down, it means a crowding effect. I am not sure if this data should stay at the main figure.
5. The authors have not clarified much about how the GTPase activity of dynamin may help in their lateral trapping. Perhaps they should discuss more.
6. The authors did not show a readout of clathrin-mediated endocytosis other than AP2 puncta, and thus should not claim its relation with clathrin-mediated endocytosis.
7. The photoswitchable protein mEOS2 has a native green fluorescence. I wonder how the authors distinguish the light coming from GFP and mEOS2 that are transfected in the same cell in some experiments.

Minor concerns:

1. A schematic of Dyn1aa, ab, bb would be of great help for readers.
2. If indeed the purpose of dimensional reduction is to reduce search time and area, why are the Dyn1aa recruited to the membrane but does not search for pre-formed clusters? In lines numbers 196-204, the authors indeed agree that Dyn1aa can be recruited directly, probably through interactions with other molecules. This means to form a cluster, a 2D search is not always necessary. Can the authors discuss this further?
3. Dynamin pre-assembly is a requirement for ultrafast endocytosis. Does the stimulation used by authors (barium and high potassium) leads to ultrafast endocytosis?
4. It has been shown before that dynamin 1aa could be involved in forming the preformed clusters (lines

66-68). Also, it is known that most of the clustered dynamin is dephosphorylated. However, the longer version of Dyn 1 can not bind to calcineurin and indeed, their data suggest that dyn1aa does not cluster despite their recruitment to the plasma membrane. Can the authors explain how dyn1aa might form a cluster?

5. The authors should show the distribution of the data in 1g and 1i. If the data is not normally distributed, they should consider showing the median along with its standard deviation. Especially since in 1h stimulation increases the variance of the decay indicating the appearance of two or multiple populations.

6. In figure 6e LUT bars are missing. In some of the figures such as 2a, 3a, 4a, and 6e, what exactly the different colors of tracks represent is not clear.

7. The manuscript suffers from many typos and grammatical and punctuational errors that need to be corrected.

8. Fig. 2a, cell seems too big as estimated with the scale bar (5 μm). can the authors check the scale bar?

Reviewer #3 (Remarks to the Author):

The manuscript titled “Dynamin1 long- and short-tail isoforms ...” by A. Jiang et. al. describes a discovery of a two-stage concentration of Dyn1 short-tail isoforms (Dyn1xb) to micro clusters on the plasma membrane, which is supported by low- and high-resolution microscopy, computer vision software, and mathematical modeling. This finding brings certain novelty to the field of CME and molecular condensation. A major revision is recommended to improve the MS.

Major concerns:

1. The complete picture of the two-stage concentration is still confusing. The authors try to convince that Dyn1xb is recruited to the plasma membrane (pm) first, then attracted to a nearby micro cluster instead of undergoing 3D diffusion. The direct evidence to back this claim is the directed motion of single Dyn1xb molecules toward a given center, which really describes a trap. Whether such directed motion exists is the key. Such motions might be extracted from the current data collection. The subtle changes in MSD and diffusion coefficients are nice additional evidence but insufficient to support the 2D trapping model because these quantities only describe random motions.

2. The proposed phenomenon has multiple time scales that are hard to interpret. A time series of global intensity describes the global translocation of Dyn1xb at a time scale of minutes. However, the detailed lateral diffusion is measured over merely 0.2 seconds (why not longer?). It is reluctant to correlate these two processes with such a large disparity in time scale. The authors need to clarify the full picture of Dyn1xb condensation. It seems that Dyn1xb turnovers much faster in low-intensity areas but much slower in high-intensity areas. Or Dyn1xb can hardly be recruited to the low-intensity areas? Have these chemical turnovers much to do with the lateral physical motions? The relation between the slow change in global intensity pattern and the fast local motion of the Dyn1xb molecules is unclear.

3. After clearing up the full picture, the authors are strongly recommended to provide an interpretable

schematic of the two-stage versus the single-stage condensation.

4. The images with tracks, e.g. Fig2.a, are very crowded. It is not fully convincing that these reflect single molecule mobility. Are the tracks reflecting single Dyn1xb molecules or the superstructures of Dyn1xb oligomers? It is mentioned in the modeling session that these are likely dynamin rings. But no ring structure is seen. Can the intensity information from the tracking help clarify the observed structures? For example, the intensity distribution of the tracks could be applied to identify and filter large structures. This will help convince that the slower diffusion is due to trapping but not due to tracking larger objects. Also, a comparison between the intensity before and after stimulation is helpful. In other words, the correlation between the lateral diffusion and the intensity of the tracks needs clarification. Ideally, the diffusion should not correlate with the intensity.

5. The definition of 'single clusters', 'single molecules' and 'areas' needs clarification. It is hard to digest these concepts, understand how they are measured, and compare them to the model. Again, this is also a part of the full picture.

6. For the model, a more comprehensive description of the variables, key rate parameters, and how the variables and parameters are related to the experiments are needed in the main text. Also, in addition to the average concentration, a snapshot or the dynamics of the intensity pattern of Dyn would be helpful to visually compare the model with the experiments and to directly support the two-stage condensation.

Minor concerns:

1. The red dots and white boxes in Fig1ab are confusing. Are these examples of ROI? They need to be described in the main text for clarity.

2. Other recruitment factors, such as membrane curvature, should be discussed, e.g. <https://doi.org/10.1083/jcb.202109013>.

3. References regarding CME hotspot need to be added, e.g. <https://doi.org/10.1111/j.1600-0854.2011.01273.x>.

4. A little more clarification on Dyn1 vs Dyn2 in different cell types is needed in the introduction before jumping to Dyn1.

5. The selection criteria of FI areas need more clarification in the main text.

6. 'AUC' is not intuitive and not explained.

7. Terms like [D] need clarification when first introduced.

REVIEWER COMMENTS

Reviewer #1 (Remarks to the Author):

In this manuscript, Jiang, et al., explored single-particle behaviors of three splice variants of Dynamin 1 (aa, ab, and bb) in PC12 cells, MEF, and mouse hippocampal neurons. Authors demonstrated that ab and bb variants are recruited to the plasma membrane and then confined to smaller areas in conditions that induce secretion and compensatory endocytosis. Single-particle analysis shows a marked decrease in diffusion kinetics of ab and bb, but not aa, molecules upon stimulus. This change is controlled by the phosphorylation status of ab and bb molecules. These data along with computer simulations are taken to suggest that ab and bb molecules have biphasic behaviors during endocytosis: recruitment and lateral diffusion/trapping. The data are of high quality. However, there are several concerns.

We thank the reviewer for their kind assessment of our work.

Major:

1. Molecular recruitment during endocytosis.

It is not clear how the observed biphasic behaviors of Dynamin 1 ab and bb relate to actual endocytic events. The authors should consider adding pHluorin-based assays or transferrin uptake assay to demonstrate that they are looking at molecular behaviors around bona fide endocytic pits.

We thank the reviewer for their suggestion – we have now performed a set of experiments to assess the co-localisation of dynamin isoforms with transferrin. For these experiments, PC12 cells were transfected with either Dyn1aa, ab, or bb tagged with mEos2, incubated with Alexa Fluor 647-labelled transferrin (Tf-647), briefly stimulated with BaCl₂ to induce compensatory endocytosis, and fixed for imaging. Imaging was performed using Super-Resolution Spinning-Disk Confocal Microscopy using Optical Photon Reassignment (SoRa; Yokogawa), after which co-localisation was assessed using Imaris. We found that dynamin isoforms co-localised to regions of transferrin uptake, and that the level of co-localisation increased upon secretagogue stimulation, therefore demonstrating that dynamin clusters on the plasma membrane are localized to bona fide endocytic pits. These new results are now included as a supplementary figure (Extended Fig. 2). Importantly, in our original manuscript, we had also performed dual-colour imaging of AP-2-GFP (localizing to clathrin-coated endocytic pits) with Dyn1bb-mEos2, and found that dynamin mobility was significantly lower within these endocytic areas (Fig. 6). Together, these results provide evidence that the clustering behaviour of dynamin that we quantify throughout the manuscript corresponds to bona fide endocytic pits.

2. Novelty

(i) The puncta formation during endocytosis is well-known phenomenon (Merrifield et al. 2003 PMID: 12198492) (Taylor et al. 2011 PMID: 21445324) (Grassart et al. 2014 PMID: 24891602) (Srinivasan et al 2018 PMID: 29668686). Several recent studies also show similar results (Cocucci et al. PMID: 25232009) and even suggest that trapping of dynamin within puncta allows further recruitment of molecules at endocytic pits (Claverie et al. 2019 doi:10.1017/S1431927619006937). We thank the reviewer for highlighting these seminal papers on dynamin-mediated endocytosis. We have now incorporated the following references into both the introduction and discussion sections of the manuscript:

“A previous study has reported the dynamics of a broad range of endocytic machinery proteins that are involved in the scission process of CME with a temporal resolution of 2s (Taylor, Perrais et al. 2011) (line 49-51).”

“Dyn1 is known to be involved in CME, and has been reported to promote the assembly and maturation of CCPs(Srinivasan, Burckhardt et al. 2018) (line 69-70).”

“Previous studies have observed the recruitment of dynamin molecules to endocytic sites, and have suggested that this recruitment is essential for both the early formation of CCPs, as well as the final membrane scission process (Merrifield, Feldman et al. 2002, Taylor, Perrais et al. 2011, Cocucci, Gaudin et al. 2014, Grassart, Cheng et al. 2014, Srinivasan, Burckhardt et al. 2018). However, the dynamics of how cytosolic dynamin molecules eventually reach the neck of the nascent endocytic structure remain unclear. (line 532-535)”

“In addition, actin has also been shown to play a role in dynamin recruitment during CME (Merrifield, Feldman et al. 2002, Grassart, Cheng et al. 2014) (line 556-557).”

These selected manuscripts did not explore the contributions of dynamin1 long- and short- tail isoforms, nor the biophysical mechanism of their recruitment to endocytic sites. In our experimental design, we have been able to specifically examine the behaviour of dynamin1 isoforms as they are recruited to the plasma membrane and laterally trapped at endocytic sites at millisecond time scale. We have also examined the spatiotemporal nanocluster dynamics of dynamin 1 isoforms using our new nanocluster spatiotemporal analysis program (Wallis, Jiang et al. 2023). Our data expands upon the previous results showing puncta formation, to explain the kinetics of lateral trapping of Dyn1 and the requirement for dimensional reduction of short-tail isoforms for fast and efficient assembly at endocytic sites. Our data further supports previous work that highlighted a role of long-tail isoforms in ultra-fast endocytosis by confirming that Dyn1aa isoform constitutively resides at the plasma membrane in sub-diffractive clusters.

The work of Cocucci et al. PMID: 25232009 reported that Dyn2 is recruited to clathrin-coated pits in two sequential phases. The first is associated with coated pit maturation; the second, with fission of the membrane neck of a coated pit. These two phases are largely different from the bi-phasic recruitment mechanism described in our study which involves the recruitment to the plasma membrane followed by lateral trapping in endocytic sites.

We thank the reviewer for pointing to Claverie et al., 2019 which is a conference proceeding. This study indeed characterised the mobility of dynamin1 (not sure which isoform) during fission, showing that it has reduced mobility at CCP sites which is in good agreement with our data in Fig. 6. We have now cited this work in our manuscript.

(ii) Authors may consider measuring the diffusion coefficient at the endocytic pits with finer ROIs and see if there are two populations of diffusional states of molecules (scaffolding vs diffusion). Authors should also address if dynamin become dissociated from endocytic pits after endocytosis is complete (Taylor et al. 2011 PMID: 21445324).

Our work significantly expands upon the previous studies, we isolate not only the mobility and behaviour of Dyn1 molecules at CCP sites as well characterising the events of Dyn1 confinement into nanoclusters by lateral trapping (Fig. 3-6). In Fig. 6 we did finer ROI in AP2-GFP cluster and found that indeed Dyn1bb mobility was significantly lower in these clusters indicating the nascent endocytic sites. Expanding our study to further examine post-fission events such as the disassembly of dynamin oligomeric structures is, we feel, outside the scope of this manuscript. However, as stated above, we have discussed this aspect in our manuscript and have referenced Taylor et al., (2011).

3. Lateral diffusion of Dyn1xb in PC12 cells.

In Figure 1, authors suggested that Dynamin 1ab and bb molecules translocate from low to high fluorescence intensity area. Authors took these data to suggest lateral diffusion, but it is equally possible that molecules are dissociated from the plasma membrane and re-recruited to the plasma membrane (Taylor et al. 2011 PMID: 21445324) (Cocucci et al. PMID: 25232009). More convincing data are necessary.

We thank the reviewer for this fair criticism. We have now performed two additional analyses to provide further evidence of lateral diffusion by Dyn1ab and Dyn1bb. Firstly, we analysed the cumulative count of single-particle trajectories (trajectories/ μm^2) for each mEos2-tagged Dyn1 isoform at the membrane of PC12 cells as a function of time (new Extended Data Fig. 3a-b). We observed that during control treatment, the cumulative count of Dyn1aa, Dyn1ab and Dyn1bb trajectories increased at an approximately linear rate (constant rate of cumulative count) over the duration of imaging. For Dyn1ab and Dyn1bb, upon Ba^{2+} stimulation, the gradient of the cumulative count rose rapidly, indicating a major recruitment onto the plasma membrane. However, during the second concentration phase in which the low fluorescent intensity area (LFA) decays whilst high fluorescent intensity area (HFA) rises (100 – 200 s onwards, see Fig. 1e-g). The rate of molecular detections plateaus across the entire plasma membrane (new Extended Data Fig. 3a-b), therefore indicating that during the second concentration phase, the total content of Dyn1ab and Dyn1bb remains approximately constant. This analysis demonstrates that in the second phase, when the re-organization of the diffraction-limited fluorescent intensity is observed, the total membrane concentration of Dyn1ab and Dyn1bb remains constant and therefore supports the lateral trapping hypothesis.

Secondly, we have extensively expanded the analysis of our single-particle tracking experiments. Here, we utilize our spatiotemporal clustering algorithm to derive regions of spatiotemporal clustering (i.e., finer regions of interest), and analysed those trajectories which intercept and enter nanoclusters in both space and time. Specifically for this analysis, we opted for measuring frame-to-frame displacement rather than diffusion coefficients, which allowed for the characterization of the stepwise changes in the displacements as trajectories enter within a derived spatiotemporal nanocluster. We observe that Dyn1 isoforms can transition from a rapidly moving state, in which the molecule samples two-dimensional space of the plasma membrane, to a highly confined state in which the molecule becomes ‘trapped’ in a spatiotemporal nanocluster. This new analysis was performed on PC12 cells (new Fig. 4) and in hippocampal neurons (new Fig. 5). In addition, we found that the proportion of clustered trajectories in which molecules laterally diffuse prior to entering the nanocluster (and having its last detection localized within) is significantly greater for short-tail Dyn1bb and Dyn1ab isoforms compared to long-tail Dyn1aa. This strongly suggests that Dyn1bb and Dyn1ab exploit lateral trapping to a greater extent than Dyn1aa (new Fig. 4e). This demonstrates that molecules can undergo lateral trapping into spatiotemporal nanoclusters, and that isoforms which exhibit bi-phasic recruitment undergo these behaviours to a greater degree.

3. Differences between Dyn1 variants in PC12 cells.

Based on Figure 1d, there is a marked difference between Dynamin 1bb and Dynamin 1ab: ab seems recruited on the entire surface. The decrease in MSD may simply come from slower diffusion coefficient of ab. Please clarify this point.

To clarify, both Dyn1ab and bb isoforms are recruited to the entire surface of the plasma membrane during the initial phase (Fig. 1a-b, Early Phase). It is important to note that the images for each Dyn1 isoform in Fig. 1d are taken from a single representative region of interest (ROI) that only encompasses a small area of the plasma membrane (ROI shown in Fig. 1a, b and c), and is intended as a visual representation of the overall temporal recruitment dynamics of dynamin. Quantification of Dyn1ab and bb isoform recruitment dynamics using numerous ROIs is depicted in Fig. 1g, which shows that they exhibit similar recruitment kinetics. The MSDs and diffusion coefficients of Dynamin molecules is quantified in Fig. 2, which shows that both Dyn1ab and bb isoforms display an activity-dependent confinement upon stimulation, with similar MSDs (Fig. 2b-c).

4. Endogenous Dyn1.

Authors tried to address potential artifacts by overexpression using nanobody against GTP-loaded dynamins,

dynab (Galli et al 2017 eLife). However, dynab recognizes both Dynamin 1 and 2 (also mentioned in Line 344). Thus, it is uncertain if this experiment has addressed the issue.

It is true that the Dynamin nanobody (DynNB-mEos2) can target both Dynamin1 and 2. However, the primary role of Dynamin 2 is in receptor mediated endocytosis, whereas Dynamin 1 is the primary isoform involved in activity-dependent endocytosis. Since we do not have access to a nanobody that is only specific to Dyn1, we feel that this was the best biosensor available on which to test the hypothesis that endogenous dynamin forms nanoclusters. Using this approach, we were able to observe endogenous dynamin clusters that were similar to those observed in PC12 cells and Neurons transfected with Dyn1-mEos2. It is important to note that we have demonstrated in the DynDKO-MEF cells that the DynNB-mEos2 is specific for Dynamin, as the number of DynNB-mEos2 trajectories on the plasma membrane was significantly reduced following TMX-induced KO indicating that the nanobody was not targeted to the membrane non-specifically.

5. Localization of Dyn1xb in cultured neurons.

For neurons, it is not clear if molecules imaged are within presynapses or not. Presynaptic markers would be helpful.

To clarify, the analysis of Dyn1 mobility within hippocampal neurons was performed using axonal and presynaptic regions of interest that were identified based on morphology (we have added this information into the figure, which is now Extended Fig. 5). However, the reviewer is correct that without proper presynaptic markers it is somehow unclear whether dynamin nanoclusters are present at the presynapse. To address this issue, we have now performed a new set of experiments in hippocampal neurons co-transfected Syt1-pHluorin, a synaptic vesicle marker highly enriched at the presynapse to identify presynaptic regions of interest. The results of these experiments carried out with the 3 Dyn1 isoforms confirmed our previous analysis of an activity-dependent decrease in mobility of the short-tail Dyn1 isoforms (new Fig. 5).

6. Recruitment of additional molecules at pre-organized clusters.

Authors claim that pre-organized clusters of Dyn1aa recruit additional molecules (line 444-448). However, the current data are not convincing. For this claim, it requires simultaneous imaging of Dyn1aa and xb in PC12 cells and in neurons.

The review is correct that co-imaging Dyn1aa and xb would be ideal to draw such conclusion. To perform these experiments the two dynamin isoforms needed to be spectrally distinct. We therefore generated halo-tag Dyn1bb as this tag can be used for single molecule imaging using a variety of bright, cell permeant fluorescent ligands. Unfortunately, swapping mEos for halotag on Dyn1bb completely blocked the activity-dependent recruitment to the plasma membrane suggesting loss of Dyn1 function. Despite our best efforts, we could not get the halo tag constructs to work and opted to discontinue this avenue. We have however rewritten this section to clarify our message:

“In support of this finding, we found that, at rest, the Dyn1aa long-tail isoform is organised in nanoscale clusters (~80 nm radius). Further, upon stimulation, the rate of Dyn1 trajectories detected in clusters largely increase (by 10-25-fold, see Fig. 3d-f) suggesting that pre-organised clusters could potentially recruit additional molecules. Further experiments are needed to assess this possibility”.

7. Experimental procedures are all too brief.

Please describe experiment in more detail. When did authors stimulate cells? What criteria were used to select ROIs? How is MSD calculated within and outside of AP2 puncta? Was AP2 imaged once or simultaneously with Dynamin? How were HFA and LFA defined? What was the density of neurons/cells?

How were they transfected? How did the TIRM imaging performed on neurons when synapses are typically found outside the evanescent field? What was the exit angle of the laser and estimated d value? These are just a few examples.

We apologize for the lack of the detail in some section of the methods, we have now added the following in the method section:

1. For live cells imaging PC12 cells or hippocampal neurons were imaged immediately following the addition of Ba^+ , or K^+ , or electrical stimulation (10 Hz for 30 s, 100 mA).
2. The LFA refers to areas where there was comparatively less observed accumulation of Dyn1-GFP fluorescence throughout the acquisition period compared to the HFA which exhibited high intensity Dyn1-GFP. Since the distribution of HFAs and LFAs varied dynamically, ROIs of equal size were chosen for each recorded movie to ensure they were within either HFAs or LFAs.
3. AP2-GFP was imaged simultaneously with Dyn1bb-mEos2 for the entire acquisition (320 s). The acquired movies were split into green (AP2-GFP) and red (Dyn1bb-mEos2) emission channels. An average z-projection of the AP2-GFP movie was created to identify AP2 discrete clusters. AP2-positive (Green clusters) and negative (no GFP signal) ROIs of the same size were selected. Those ROIs were then used to perform single-particle tracking on Dyn1 molecules deemed 'positive' or 'negative' AP2 areas.
4. For PC12 cells: 4.5×10^5 cells in 6 well plate (or 3.5cm dish) were plated a day before transfection (~60-80% confluent). Transfections were performed using Lipofectamine LTX and Plus Reagent (Thermo Fisher Scientific, #A12621) as per the manufacturer's instructions. In brief, transfection was prepared in Optimem (serum-reduced medium) containing 1-3 mg of Plasmid DNA, 1mL/mg DNA Plus Reagent and $6.75 \mu\text{L}$ LTX for each 35mm dish. For hippocampal neurons: A total of 1×10^5 neurons were seeded within the 29 mm glass bottom dishes coated with poly-L-lysine. They were transfected in 14-16 days-in-vitro (DIV16) with Dyn1aa-mEos2 or Dyn1bb-mEos2 using Lipofectamine 2000 (Thermo Fisher Scientific, #11668019), following the manufacturer's instructions. In brief, transfection was prepared in Neurobasal medium with either 1-3 μg of Plasmid DNA, or $1 \mu\text{L}/\mu\text{g}$ Lipofectatmine2000 was used per 35mm dish.
5. Presynaptic boutons present in the field of view on the cover-glass within the evanescent field were chosen for imaging from mature hippocampal neurons. During the imaging experiments, hippocampal neurons were maintained in low K^+ solution and stimulated with either extracellular high K^+ or field stimulation electrodes (RC-21BRFS; Warner Instruments, Holliston, MA). Neurons were imaged before and immediately after the addition of high K^+ solution. Alternatively, to electrically induce compensatory endocytosis, neurons were stimulated with 1ms pulses delivered at 10 Hz for 30 s (100mA) and imaged before (Ctrl) and during/following (10 Hz 30 s) electrical stimulation.
6. TIRF angle was calibrated each imaging session and TIRF critical angle was approximately 70° .
7. Other details have also been added into the method section for completeness including all new analysis section (spatiotemporal cluster analysis, single cluster analysis, transferrin experiment and SoRa imaging).

Minor:

1. In line 406-409, authors claim that dynamin molecules on membrane are recruited to the clusters via oligomerization. This is based on the results from Dyngo-4a experiments in Figure 6. However, it is not clear if Dyngo-4a inhibit oligomerization of dynamin in the cells based on one of authors previous work (McCluskey et al. 2013, PMID: 24025110)

We have now rewritten this part to clarify our message and interpretation of our model:

“For the model to reproduce the relatively sharp increase in Dyn1 density following a very slow initial growth as detected in Fig. 1j,k, we needed to introduce a cooperative effect. Dyn1 not only had to bind to the ‘activators’ or cluster-associated proteins³, but it had to subsequently recruit itself to the clusters via dimerization (continuum differential equations) or oligomerization (stochastic reaction-diffusion), mimicking assembly of a metastable dynamin helix. By explicitly capturing Dyn1 self-assembly in the stochastic simulations using the same optimized rates as the ordinary differential equations, we can directly see that once a single Dyn1 is ‘activated’ it helps recruit multiple additional Dyn1 (from 1-4 copies in our model) to cluster sites (Movies 3). It is important to note that dynamin oligomerises in two ways, as rings or as helices. Dyngo-4a specifically inhibits helical formation states of dynamin but does not affect ring formation (McCluskey et al. 2013). Our model is therefore in good agreement with the inhibitory effect of Dyngo-4a on the activity-dependent reduction in mobility (Fig. 6).”

2. Please add a reference for line 474-476.

As requested, we have now rephrased the sentence and added a reference for it:

“The dogma stipulating the direct recruitment of Dyn1 to the neck of nascent endosomes would require a very high cytosolic concentration of Dyn1 to achieve the timely recruitment to these discrete sites on the plasma membrane (Praefcke and McMahon 2004) (Fig. 8h).” (Line 604-606)

Reviewer #2 (Remarks to the Author):

In the manuscript titled “Dynamin1 long- and short-tail isoforms exploit distinct recruitment and spatial patterns to form endocytic nanoclusters,” the authors have explored the activity-dependent formation of Dynamin 1 (Dyn 1) nanoclusters at the plasma membrane of various cell types using SMLM techniques and have identified the mechanisms of different Dynamin 1 isoform recruitment, which differ in their C-terminal PRD. Overall the manuscript elaborates on how dynamin recruitment may match the timescale of endocytosis, borrowing ideas from thermodynamically favorable information compression through dimensional reduction, which is insightful and compelling. However, the manuscript needs further improvements detailed below.

We thank the reviewer for their kind assessment of our work.

Major concerns:

1. In the abstract, the authors concluded that “we demonstrate that short-tail Dyn1 isoforms Dyn1ab and Dyn1bb display an activity-dependent recruitment to the membrane, promptly followed by concentration into nanoclusters.” This can be better viewed in Fig. 1c-f. The authors explain the decline of fluorescence in LFA as the source for the increase of fluorescence in HFA, and thus interpret the result as dynamin movement from LFA to HFA. However, it could be possible that dynamin in LFA falls off the membrane whereas dynamin in HFA continues to remain as clusters and increase in intensity. Can the authors exclude this possibility?

We thank the reviewer for their alternative model to explain our experimental data. This concern has been addressed in the response to Reviewer 1 (please see the response to Reviewer 1, Comment 3 copied below).

“We thank the reviewer for this fair criticism. We have now performed two additional analyses to provide further evidence of lateral diffusion by Dyn1ab and Dyn1bb. Firstly, we analysed the cumulative count of single-particle trajectories (trajectories/ μm^2) for each mEos2-tagged Dyn1 isoform at the membrane of PC12

cells as a function of time (new Extended Data Fig. 3a-b). We observed that during control treatment, the cumulative count of Dyn1aa, Dyn1ab and Dyn1bb trajectories increased at an approximately linear rate (constant rate of cumulative count) over the duration of imaging. For Dyn1ab and Dyn1bb, upon Ba²⁺ stimulation, the gradient of the cumulative count rose rapidly, indicating a major recruitment onto the plasma membrane. However, during the second concentration phase in which the low fluorescent intensity area (LFA) decays whilst high fluorescent intensity area (HFA) rises (100 – 200 s onwards, see Fig. 1e-g). The rate of molecular detections plateaus across the entire plasma membrane (new Extended Data Fig. 3a-b), therefore indicating that during the second concentration phase, the total content of Dyn1ab and Dyn1bb remains approximately constant. This analysis demonstrates that in the second phase, when the re-organization of the diffraction-limited fluorescent intensity is observed, the total membrane concentration of Dyn1ab and Dyn1bb remains constant and therefore supports the lateral trapping hypothesis.

Secondly, we have extensively expanded the analysis of our single-particle tracking experiments. Here, we utilize our spatiotemporal clustering algorithm to derive regions of spatiotemporal clustering (i.e., finer regions of interest), and analysed those trajectories which intercept and enter nanoclusters in both space and time. Specifically for this analysis, we opted for measuring frame-to-frame displacement rather than diffusion coefficients, which allowed for the characterization of the stepwise changes in the displacements as trajectories enter within a derived spatiotemporal nanocluster. We observe that Dyn1 isoforms can transition from a rapidly moving state, in which the molecule samples two-dimensional space of the plasma membrane, to a highly confined state in which the molecule becomes ‘trapped’ in a spatiotemporal nanocluster. This new analysis was performed on PC12 cells (new Fig. 4) and in hippocampal neurons (new Fig. 5). In addition, we found that the proportion of clustered trajectories in which molecules laterally diffuse prior to entering the nanocluster (and having its last detection localized within) is significantly greater for short-tail Dyn1bb and Dyn1ab isoforms compared to long-tail Dyn1aa. This strongly suggests that Dyn1bb and Dyn1ab exploit lateral trapping to a greater extent than Dyn1aa (new Fig. 4e). This demonstrates that molecules can undergo lateral trapping into spatiotemporal nanoclusters, and that isoforms which exhibit bi-phasic recruitment undergo these behaviours to a greater degree.”

2. In figures 2a and b, it is surprising that even though the authors detected such a large number of tracks, the MSD increased almost linearly with time under both stimulated and unstimulated conditions. Had it been active confinement, the stimulated curve should have reached a plateau ideally. The reduced slope of the red curve means that the molecules are in general moving slowly, which could be simply because of crowding than active confinement to the pre-assembled clusters. The authors should plot the distribution of the power factor or the scaling factor of time (α) to verify confinement.

Our MSD and diffusion co-efficient analysis is performed on tracks that are detected for at least 8 frames. Despite having a large number of tracks, most of these detections are very short lived due to the short lifetime of photoconverted mEos2. The MSD curve shown in the revised figures has now been extended to show values of up to 10 frames, or 0.2 seconds, noting that some tracks do not last this long and the statistical strength for these additional data points is weaker. The MSD slope now show a plateau which is partially indicative of confinement, in conjunction with our analysis of the mobile/immobile ratio. To further assess the confinement of Dyn1 molecules, we have now used a novel spatiotemporal clustering algorithm (BOOSH) to delineate discrete regions of spatiotemporal clustering to analyse trajectories which intercept these spatiotemporal nanoclusters. We observe that Dyn1 isoforms can transition from a rapidly moving state, in which the molecule samples a two-dimensional space (defined by the evanescent field of the TIRF imaging), to a tightly confined state in which the molecule becomes ‘trapped’ in the spatiotemporal nanocluster, leading to dramatically reduced step-wise displacements. This ‘trapping’ of a laterally diffusing molecule into spatiotemporal clusters is in direct alignment with an active confinement into nanoclusters.

These data, including representative trajectories displayed in Figure 4, indicate that the reduced mobility of dynamin can be attributed to its confinement into clusters and not simply due to molecular crowding.

3. Interestingly, the authors analyzed the time scale of these clusters. I am curious, what is the average time length through which they can follow the same track? Can the authors calculate the average fallout rate of these tracks, which should be less if the authors compare it to another molecule that moves in 3D? If they follow a single track long enough, do they see ‘cage hopping’ phenomena i.e. repeated confinement and release from clusters Does the average tracking time for individual tracks and their fall-out rate change with stimulation?

As discussed above, the average track length is very short and we have not been able to observe the hopping behaviour between nanoclusters. However, as discussed above (please also refer to our response to Reviewer 1 Comment 3), we have generated a novel analysis now presented in Figure 4 and 5 which characterises the trapping of dynamin1 molecules into nanoclusters. In brief, molecules that were diffusing on the membrane prior to their confinement into nanoclusters was analysed for all isoforms in PC12 and Hippocampal neurons. We also calculated the average track length for all isoforms. Although some changes are significant, the effect size is small may not be reflective of true retention times of Dyn1 within the clusters. See results below.

4. Why does the cluster radius reduce on stimulation of hippocampal neurons while it increases in PC12 cells? Probably, KCl-mediated stimulation is a harsh method of neuronal stimulation, and the authors may want to use action potential stimulation. In this context, if the cluster parameters don't change but the MSD values of a single molecule track slow down, it means a crowding effect. I am not sure if this data should stay at the main figure.

We have now completed a set of experiments that includes electrical stimulation of hippocampal neurons as requested by the reviewer. We used electrical stimulation (10 Hz for 30 s), and analysed the mobility of Dyn1 isoforms before and during/following stimulation and found that, similar to PC12 cells, stimulating neurotransmitter release and compensatory endocytosis dramatically reduced the mobility of the short tail isoforms. This data is now included in new Figure 5. As requested by the reviewer, we have relocated the high K⁺ results into Extended data Figure 5, noting that the analysis was performed on the entire neuron (axons and pre-synapses), which could account for the discrepancy. We also performed spatiotemporal cluster analysis (focusing on pre-synapses only) following electrical stimulation and this data also showed that Dyn1bb cluster size increased upon activity, but in this paradigm the data did not reach statistical significance. The detection density following electrical stimulation also resulted in an increase in trajectory density but this was not as pronounced in PC12 cells. The protein abundance alone is not enough to alter the

mobility of molecules, unless they are interacting in discrete sites. Our model attempts to differentiate these principles.

5. The authors have not clarified much about how the GTPase activity of dynamin may help in their lateral trapping. Perhaps they should discuss more.

We have now added some additional discussion points regarding the potential role of GTPase activity of in the nanoclustering and lateral trapping of Dynamin:

“Importantly, the trapping of newly recruited Dyn1xb molecules within nanoclusters also depends on dynamin GTPase activity suggesting that these clusters are functionally involved in endocytosis. Inhibiting dynamin GTPase activity prevents the activity-dependent mobility reduction showing that the GTPase activity is critical for its lateral trapping. Further, the presence of clusters when tracking endogenous dynamin with DynNB that binds preferentially to the GTP-hydrolytic state of dynamin further supports this view.” (line 574-579).

6. The authors did not show a readout of clathrin-mediated endocytosis other than AP2 puncta, and thus should not claim its relation with clathrin-mediated endocytosis.

We have modified the manuscript to state that AP2 cluster is a marker of ‘nascent’ endocytic structures. To further address this question, we have now included a set of experiments that show the level of colocalization between Dynamin1 isoforms and labelled Transferrin, a known clathrin-mediated endocytosis cargo (new Extended Data Figure 2).

7. The photoswitchable protein mEOS2 has a native green fluorescence. I wonder how the authors distinguish the light coming from GFP and mEOS2 that are transfected in the same cell in some experiments.

In our experimental set up, the brightness of unconverted mEos2 is much dimmer than that of GFP and we solely rely on the red-shifted photoconverted species for tracking Dyn1 molecules. To ensure that our AP2-GFP fluorescence was reliably used to generate ROIs we performed a set of acquisitions where we imaged PC12 cells that were transfected with either AP2-GFP or Dyn1bb-mEos2 using identical imaging settings (described below), recording both the green and red channels (while simultaneously inducing mEos2 photo conversion with 405nm laser).

In this recording example, the mean fluorescence intensity (FI) of AP2-GFP was around 1091 (a.u.) with a background ~464 (a.u.) while the mean FI of Dyn1bb-mEos2 is ~471 (a.u.) with a background ~391(a.u.). The FI of AP2-GFP is therefore ~7.8x higher than the Dyn1bb-mEos2. It is important to note, that in our analysis in Fig. 6 takes into account this brightness difference.

We have now included the following description of the analysis in the methods section of the manuscript: “AP2-GFP was imaged simultaneously with Dyn1bb-mEos2 for the entire acquisition (320 s). The acquired movies were split into green (AP2-GFP) and red (Dyn1bb-mEos2) emission channels. An average z-projection of the AP2-GFP movie was created to identify AP2 discrete clusters. AP2-positive (Green clusters) and negative (no GFP signal) ROIs of the same size were selected. Those ROIs were then used to perform single-particle tracking on Dyn1 molecules deemed ‘positive’ or ‘negative’ AP2 areas.”

AP2-GFP_5% 491_80% 561_20ms 20Average

Mean FI is ~1091 (a.u.) with a background ~464 (a.u.) for AP2-GFP.

Dyn1bb-mEos2_5% 491_80% 561_20ms 20Average

Mean FI is ~471 (a.u.) with a background ~391(a.u.) for Dyn1bb-mEos2.

Minor concerns:

1. A schematic of Dyn1aa, ab, bb would be of great help for readers.

We thank the reviewer for their suggestion, we have now incorporated schematics depicting the functional domains of Dyn1aa, ab and bb into Fig. 1 for added clarity.

2. If indeed the purpose of dimensional reduction is to reduce search time and area, why are the Dyn1aa recruited to the membrane but does not search for pre-formed clusters? In lines numbers 196-204, the authors indeed agree that Dyn1aa can be recruited directly, probably through interactions with other molecules. This means to form a cluster, a 2D search is not always necessary. Can the authors discuss this further?

Our new analysis has shown that Dyn1aa molecules can also undergo a 2D search for pre-formed clusters on the plasma membrane, however this occurs to a much lesser extent compared to Dyn1ab and bb. Our data shows that the short-tail isoforms Dyn1ab/bb showed a biphasic recruitment mechanism, and the dimensional reduction and 2D search is optimally utilized. We have included a paragraph in the discussion about the different recruitment mechanism of the long- and short-tail isoforms and how they might link to different endocytic pathways: “Short-tail Dyn1 isoforms have been reported to mediate activity-dependent bulk endocytosis²². To meet the high demand for bulk membrane retrieval, short-tail Dyn1 isoforms undoubtedly need a robust recruitment mechanism capable of sustaining the fission of large nascent endosomes. Binding to the plasma membrane followed by nanoclustering is a hallmark of dimensional reduction which greatly reduces the concentration requirement and increases the speed of dynamin organisation. Our finding of a bi-phasic recruitment allows for large capacity binding to the plasma membrane, followed by lateral trapping into larger and more numerous nanoclusters compared to the long-tail isoform. Our work further explains the need for pre-clustered dynamin on the plasma membrane as it greatly facilitates the subsequent clustering of dynamin short-tail isoforms following their translocation to the plasma membrane.”

3. Dynamin pre-assembly is a requirement for ultrafast endocytosis. Does the stimulation used by authors (barium and high potassium) leads to ultrafast endocytosis?

To the best of our knowledge there are no reports in the literature of ultrafast endocytosis occurring in PC12 cells. Likewise, it is unlikely that the High K⁺ stimulation induce ultrafast endocytosis in hippocampal neurons. Electrical stimulation, added in the revised version of this paper could potentially elicit this process.

4. It has been shown before that dynamin 1aa could be involved in forming the preformed clusters (lines 66-68). Also, it is known that most of the clustered dynamin is dephosphorylated. However, the longer version of Dyn 1 can not bind to calcineurin and indeed, their data suggest that dyn1aa does not cluster despite their recruitment to the plasma membrane. Can the authors explain how dyn1aa might form a cluster?

A recent publication has indeed described the assembly of Dyn1xa isoforms as pre-assemblies using STED microscopy (Imoto, Raychaudhuri et al. 2022). The authors suggest that the likely mechanism mediating such pre-assembled clusters is through the binding of Dyn1xa to Syndapin1 via the generation of biomolecular condensates. To clarify, we used single molecule imaging to characterise the dynamics of Dyn1aa clusters (See Fig 3). While Dyn1aa does not have a calcineurin binding site, dephosphorylation requires the binding between calcineurin and the short tail isoform Dyn1xb (Xue, Graham et al. 2011). The clustering of Dyn1aa may therefore require the recruitment of other binding partners, or maybe some other Dyn1 isoforms. We have expanded our discussion to include potential factors that might modulate cluster formation, including membrane curvature, actin interaction, lipid binding, etc.

5. The authors should show the distribution of the data in 1g and 1i. If the data is not normally distributed, they should consider showing the median along with its standard deviation. Especially since in 1h stimulation increases the variance of the decay indicating the appearance of two or multiple populations.

We have now added the individual data points for all bar graphs within the manuscript to show the distribution of the data. The data in Figure 1g and 1i are not normally distributed and the appropriate statistical tests have been used (non-normally distributed paired data was analysed with Wilcoxon test).

6. In figure 6e LUT bars are missing. In some of the figures such as 2a, 3a, 4a, and 6e, what exactly the different colors of tracks represent is not clear.

We have now added the missing LUT bars to Fig. 6, and have included the relevant information in the figure legend as well as in the results sections for clarity. We have now added the detail: “sptPALM diffusion coefficient map (darker points are more confined), trajectory map (colour coded as they appear in time) and average intensity map (brighter points indicate higher intensity).”

7. The manuscript suffers from many typos and grammatical and punctuational errors that need to be corrected.

We apologise for these errors, and to the best of our knowledge, have now corrected these errors.

8. Fig. 2a, cell seems too big as estimated with the scale bar (5 μm). can the authors check the scale bar?

We have now amended the scale bar in Fig. 2a, and have double-checked the scale bars for all figures.

Reviewer #3 (Remarks to the Author):

The manuscript titled “Dynamin1 long- and short-tail isoforms ...” by A. Jiang et. al. describes a discovery of a two-stage concentration of Dyn1 short-tail isoforms (Dyn1xb) to micro clusters on the plasma membrane, which is supported by low- and high-resolution microscopy, computer vision software, and mathematical modelling. This finding brings certain novelty to the field of CME and molecular condensation. A major revision is recommended to improve the MS.

We thank the reviewer for their kind assessment of our work.

Major concerns:

1. The complete picture of the two-stage concentration is still confusing. The authors try to convince that Dyn1xb is recruited to the plasma membrane (pm) first, then attracted to a nearby micro cluster instead of undergoing 3D diffusion. The direct evidence to back this claim is the directed motion of single Dyn1xb molecules toward a given center, which really describes a trap. Whether such directed motion exists is the key. Such motions might be extracted from the current data collection. The subtle changes in MSD and diffusion coefficients are nice additional evidence but insufficient to support the 2D trapping model because these quantities only describe random motions.

We thank the reviewer for this concern. We have addressed this concern through significant in-depth analysis of our sptPALM data in PC12 cells and hippocampal neurons (new Fig. 4 and Fig. 5, please also refer to the response to Reviewer 1, Comment 3). In brief, Dyn1 molecules that were diffusing on the membrane prior to their confinement into nanoclusters was analysed for all isoforms in PC12 and Hippocampal neurons. This analysis was used to characterise the trapping behaviour of dynamin1. Importantly, we do not hypothesise that active directed motion is the mechanism by which these clusters are formed. To clarify, the lateral trapping mechanism we describe, occurs stochastically via random diffusion of a two-dimensionally sampling molecule into a ‘trap’. We have copied the answer to reviewer 1 below:

“We thank the reviewer for this fair criticism. We have now performed two additional analyses to provide further evidence of lateral diffusion by Dyn1ab and Dyn1bb. Firstly, we analysed the cumulative count of single-particle trajectories (trajectories/ μm^2) for each mEos2-tagged Dyn1 isoform at the membrane of PC12 cells as a function of time (new Extended Data Fig. 3a-b). We observed that during control treatment, the cumulative count of Dyn1aa, Dyn1ab and Dyn1bb trajectories increased at an approximately linear rate (constant rate of cumulative count) over the duration of imaging. For Dyn1ab and Dyn1bb, upon Ba^{2+} stimulation, the gradient of the cumulative count rose rapidly, indicating a major recruitment onto the plasma membrane. However, during the second concentration phase in which the low fluorescent intensity area (LFA) decays whilst high fluorescent intensity area (HFA) rises (100 – 200 s onwards, see Fig. 1e-g). The rate of molecular detections plateaus across the entire plasma membrane (new Extended Data Fig. 3a-b), therefore indicating that during the second concentration phase, the total content of Dyn1ab and Dyn1bb remains approximately constant. This analysis demonstrates that in the second phase, when the re-organization of the diffraction-limited fluorescent intensity is observed, the total membrane concentration of Dyn1ab and Dyn1bb remains constant and therefore supports the lateral trapping hypothesis.

Secondly, we have extensively expanded the analysis of our single-particle tracking experiments. Here, we utilize our spatiotemporal clustering algorithm to derive regions of spatiotemporal clustering (i.e., finer regions of interest), and analysed those trajectories which intercept and enter nanoclusters in both space and time. Specifically for this analysis, we opted for measuring frame-to-frame displacement rather than diffusion coefficients, which allowed for the characterization of the stepwise changes in the displacements

as trajectories enter within a derived spatiotemporal nanocluster. We observe that Dyn1 isoforms can transition from a rapidly moving state, in which the molecule samples two-dimensional space of the plasma membrane, to a highly confined state in which the molecule becomes ‘trapped’ in a spatiotemporal nanocluster. This new analysis was performed on PC12 cells (new Fig. 4) and in hippocampal neurons (new Fig. 5). In addition, we found that the proportion of clustered trajectories in which molecules laterally diffuse prior to entering the nanocluster (and having its last detection localized within) is significantly greater for short-tail Dyn1bb and Dyn1ab isoforms compared to long-tail Dyn1aa. This strongly suggests that Dyn1bb and Dyn1ab exploit lateral trapping to a greater extent than Dyn1aa (new Fig. 4e). This demonstrates that molecules can undergo lateral trapping into spatiotemporal nanoclusters, and that isoforms which exhibit bi-phasic recruitment undergo these behaviours to a greater degree.”

2. The proposed phenomenon has multiple time scales that are hard to interpret. A time series of global intensity describes the global translocation of Dyn1xb at a time scale of minutes. However, the detailed lateral diffusion is measured over merely 0.2 seconds (why not longer?). It is reluctant to correlate these two processes with such a large disparity in time scale. The authors need to clarify the full picture of Dyn1xb condensation. It seems that Dyn1xb turnovers much faster in low-intensity areas but much slower in high-intensity areas. Or Dyn1xb can hardly be recruited to the low-intensity areas? Have these chemical turnovers much to do with the lateral physical motions? The relation between the slow change in global intensity pattern and the fast local motion of the Dyn1xb molecules is unclear.

It is important to clarify the different time scales chosen are appropriate for the two types of experiments requiring distinct lateral resolution with TIRF microscopy. Using Dyn1-GFP, we cannot differentiate individual molecules and the resolution is therefore low (250 nm) and we simply measure the overall intensity of Dyn1-GFP recruitment (Low Resolution, 50 Hz, 20ms for 320 seconds) at the ~100 nm z-depth of the evanescent field. The timescale requirement for such recruitment is low. To track the mobility of individual Dyn1 molecules, we photoconverted Dyn1-mEos2 molecules using sptPALM (acquisition 50 Hz, 20ms for 320 seconds). This allowed us to indeed track single molecules of dyn1 isoform following their recruitment to the plasma membrane and observe their clustering in space and time. It is important to note that the individual tracks of mEos2 only remain fluorescent for 8-20 frames before they are bleached due to the stochastic nature of single molecule fluorescence. We analysed the mobility of dynamin molecules within AP-2-positive endocytic pits (Fig. 6) and showed that it was more confined in these areas. The combination of the two main techniques have allowed us to track dynamin molecules as they are “mass-recruited” to the plasma membrane and track their behaviour at single molecule level using single molecule imaging.

3. After clearing up the full picture, the authors are strongly recommended to provide an interpretable schematic of the two-stage versus the single-stage condensation.

We thank the reviewer for their suggestion. We have now included a schematic illustrating the differences between bi-phasic recruitment and conventional single-phase recruitment (new Fig. 8).

4. The images with tracks, e.g. Fig2.a, are very crowded. It is not fully convincing that these reflect single molecule mobility. Are the tracks reflecting single Dyn1xb molecules or the superstructures of Dyn1xb oligomers? It is mentioned in the modeling session that these are likely dynamin rings. But no ring structure is seen. Can the intensity information from the tracking help clarify the observed structures? For example, the intensity distribution of the tracks could be applied to identify and filter large structures. This will help convince that the slower diffusion is due to trapping but not due to tracking larger objects. Also, a comparison between the intensity before and after stimulation is helpful. In other words, the correlation

between the lateral diffusion and the intensity of the tracks needs clarification. Ideally, the diffusion should not correlate with the intensity.

The tracks figure panels (such as those shown in Fig 2a) are a projection of every track that were obtained during the entire course of the 16,000 frame (50 Hz, 320 s). The individual frames of these movies track sparse Dyn1 molecules by tightly controlling 405 laser photoconversion of mEos2, to ensure that individual tracks of Dyn1 can be isolated. Importantly, this image alone could not show superstructures like ring structure of dynamin as these trajectories are presented in different time points. Our analysis of Dyn1 in areas of the membrane occupied by AP-2 further demonstrates that the mobility of Dyn1 is not uniform across the plasma membrane.

The fluorescence intensity before and after stimulation in Fig. 1 shows the recruitment of Dyn1-GFP to the membrane and the FI is plotted as a histogram. The single trajectory intensity in Fig. 2a (fourth panels), shows the localisation of single molecules on discrete areas of the membrane, but these do not always correlate with the diffusion co-efficient (third panels). However, our spatiotemporal nanocluster analysis, which can isolate trajectories that overlap in both space and time, has shown revealed areas with nanoclusters (with radius no larger than 0.15 μm) which contain Dyn1 molecules (See Fig. 3). Our new analysis discussed above has also shown the lateral trapping of single molecules.

5. The definition of ‘single clusters’, ‘single molecules’ and ‘areas’ needs clarification. It is hard to digest these concepts, understand how they are measured, and compare them to the model. Again, this is also a part of the full picture.

We apologise for the lack of clarity regarding those concepts. We have now revised our expressions, added new analysis and discussions as discussed above to make the full picture clearer. It will be easier to understand those concepts now.

6. For the model, a more comprehensive description of the variables, key rate parameters, and how the variables and parameters are related to the experiments are needed in the main text. Also, in addition to the average concentration, a snapshot or the dynamics of the intensity pattern of Dyn would be helpful to visually compare the model with the experiments and to directly support the two-stage condensation.

We have expanded the description of the model in the main text to motivate the variables in this minimal model and their connection to the experimental system by noting that in addition to Dyn1, we must minimally track a population of Dyn1 binders across the membrane (e.g. PIP2, Dyn1 SH3 binding partners), and a population that is localized to endocytic cluster sites (e.g. amphiphysin, intersectin). See lines 450-480.

For the parameters, we describe the rate constants in the Figure legend as they control specific protein interactions, and in the Methods explain the range-limits we imposed. Because our membrane recruiters (R and A) do not represent a single binding partner but rather a population of diverse partners, we could not use biochemical rates for specific domain-domain interactions as inputs for the model. We now further explain in the main text which model parameters are constrained by experiment (diffusion constants, cluster densities, intensity increases, initial relative intensities per isoform, and the time-evolution of the intensity increase within clusters) vs those parameters that we optimized to reproduce the observed experimental intensity vs time (concentrations of our variables and the rate constants).

Following your suggestion, we also added in plots to main Fig. 8 that show how the density of Dyn1 changes within and around the cluster sites as a function of time, both pre and post stimulation.

Finally, we implemented another version of our model that captures the single-particle dynamics of the Dyn1 molecules as they assemble at cluster sites. This model is also reaction-diffusion, and uses the same optimized parameters as the PDE model, but instead of having Dyn1 dimerization reactions to promote clustering, it explicitly gives Dyn1 4 sites so that it can spatially self-assemble. We include images of these simulations (Fig. 8d,e) and show their excellent quantitative agreement with the continuum models (Fig. 8f). This model more directly illustrates how both Dyn1 localization to cluster sites (by binding other proteins) and their assembly with one another drives rapid density increases at cluster sites.

Minor concerns:

1. The red dots and white boxes in Fig1ab are confusing. Are these examples of ROI? They need to be described in the main text for clarity.

We thank the reviewer for highlighting this issue. The red dots show areas of the cell that have a high fluorescent intensity at the time point the frame was taken, whereas the white boxes show representative regions of interest (ROIs) taken from either LFAs (outside of red dot regions) or HFAs (inside of red dot regions). These representative ROIs are now labelled with either “LFA” or “HFA” in Figure 1a and b for clarity. Additionally, we have clarified this point in the figure legend: “White box, representative ROI of LFA and HFA. Red area represents the HFA for this frame.”

2. Other recruitment factors, such as membrane curvature, should be discussed, e.g. <https://doi.org/10.1083/jcb.202109013>.

We have now added this point in the discussion (including the suggested reference):

“It is important to note that other recruitment factors, such as membrane curvature, may also play a role in the assembly of the endocytic machinery including dynamin. In the absence of clathrin, the induced membrane curvature is thought to be sufficient to recruit CME proteins and induce endocytosis (Cail, Shirazinejad et al. 2022). In addition, actin also plays a role in dynamin recruitment during CME (Merrifield, Feldman et al. 2002, Grassart, Cheng et al. 2014).” (Line 553-557)

3. References regarding CME hotspot need to be added, e.g. <https://doi.org/10.1111/j.1600-0854.2011.01273.x>.

We have now added this point in the discussion (including the suggested references):

“It has been previously demonstrated that CCPs can form repeatedly at predefined sites at varying times (hotspots), regulated by accessory proteins like AP2 that serve as nucleation factors (Nunez, Antonescu et al. 2011).” (Line 537-539)

4. A little more clarification on Dyn1 vs Dyn2 in different cell types is needed in the introduction before jumping to Dyn1.

We have now added this point in the introduction:

“There are 3 dynamin genes, dynamin 1 (Dyn1), dynamin 2 (Dyn2) and dynamin 3 (Dyn3). Dyn1 is neuronal specific, Dyn2 is ubiquitously expressed, and Dynamin 3 is expressed in a subset of cells including the testis.” (Line 61-63)

5. The selection criteria of FI areas need more clarification in the main text.

For further clarification, we have provided a more detailed explanation of the FI area selection method in the methods section (also see response to Reviewer 1, comment 7):

“The LFA refers to areas with comparatively less Dyn1-GFP fluorescence observed throughout the acquisition period, compared to that of HFA which exhibited high intensity Dyn1-GFP. Since the distribution of HFAs and LFAs varied dynamically, ROIs of equal size were meticulously chosen for each movie to ensure they were within either HFAs or LFAs throughout the duration of the acquisition.”

6. ‘AUC’ is not intuitive and not explained.

We have now modified the text from “AUC” to “Area Under the MSD Curve (AUC)”. Since it is not appropriate to do statistical analysis on the MSD curve directly, the AUC is used for statistical comparisons instead, see (Gormal, Padmanabhan et al. 2020) and (Kasula, Chai et al. 2016).

7. Terms like [D] need clarification when first introduced.

We thank the reviewer for pointing this out, we have now defined all variables immediately upon their first appearance in the manuscript, as well as in the figure legend of Fig. 8:

“This optimized model has initial concentrations for Dyn1 as $[D]_{mem} = 30$ copies/ μm^2 and $[D]_{sol} = 10$ μM , where the subscript denotes localization on membrane or in solution. Uniform recruiters are initially concentrated at $[R]_{mem} = 4.85$ copies/ μm^2 , and ‘activators’ are $[A]_{clus} = 207$ copies/ μm^2 . Optimal rates are $k_{mem} = 0.017$ $\mu\text{M}^{-1}\text{s}^{-1}$, $k_{frev} = 0.09$ $\mu\text{M}^{-1}\text{s}^{-1}$, $k_{memPost} = 0.0024$ $\mu\text{M}^{-1}\text{s}^{-1}$, $k_{dyn} = 0.035$ $\mu\text{M}^{-1}\text{s}^{-1}$, $k_b = 0.15$ s^{-1} , $h=10$ nm.”

Cail, R. C., C. R. Shirazinejad and D. G. Drubin (2022). "Induced nanoscale membrane curvature bypasses the essential endocytic function of clathrin." J Cell Biol **221**(7).

Cocucci, E., R. Gaudin and T. Kirchhausen (2014). "Dynamain recruitment and membrane scission at the neck of a clathrin-coated pit." Molecular biology of the cell **25**(22): 3595-3609.

Gormal, R. S., P. Padmanabhan, R. Kasula, A. T. Bademosi, S. Coakley, J. Giacomotto, A. Blum, M. Joensuu, T. P. Wallis and H. P. Lo (2020). "Modular transient nanoclustering of activated $\beta 2$ -adrenergic receptors revealed by single-molecule tracking of conformation-specific nanobodies." Proceedings of the National Academy of Sciences **117**(48): 30476-30487.

Grassart, A., A. T. Cheng, S. H. Hong, F. Zhang, N. Zenzer, Y. Feng, D. M. Briner, G. D. Davis, D. Malkov and D. G. Drubin (2014). "Actin and dynamain2 dynamics and interplay during clathrin-mediated endocytosis." J Cell Biol **205**(5): 721-735.

Imoto, Y., S. Raychaudhuri, Y. Ma, P. Fenske, E. Sandoval, K. Itoh, E. M. Blumrich, H. T. Matsubayashi, L. Mamer, F. Zarebidaki, B. Sohl-Kielczynski, T. Trimbuch, S. Nayak, J. H. Iwasa, J. Liu, B. Wu, T. Ha, T. Inoue, E. M. Jorgensen, M. A. Cousin, C. Rosenmund and S. Watanabe (2022). "Dynamain is primed at endocytic sites for ultrafast endocytosis." Neuron **110**(17): 2815-+.

Kasula, R., Y. J. Chai, A. T. Bademosi, C. B. Harper, R. S. Gormal, I. C. Morrow, E. Hosy, B. M. Collins, D. Choquet and A. Papadopoulos (2016). "The Munc18-1 domain 3a hinge-loop controls syntaxin-1A nanodomain assembly and engagement with the SNARE complex during secretory vesicle priming." Journal of Cell Biology **214**(7): 847-858.

Merrifield, C. J., M. E. Feldman, L. Wan and W. Almers (2002). "Imaging actin and dynamain recruitment during invagination of single clathrin-coated pits." Nat Cell Biol **4**(9): 691-698.

Nunez, D., C. Antonescu, M. Mettlen, A. Liu, S. L. Schmid, D. Loerke and G. Danuser (2011). "Hotspots organize clathrin-mediated endocytosis by efficient recruitment and retention of nucleating resources." Traffic **12**(12): 1868-1878.

Praefcke, G. J. and H. T. McMahon (2004). "The dynamin superfamily: universal membrane tubulation and fission molecules?" Nat Rev Mol Cell Biol **5**(2): 133-147.

Srinivasan, S., C. J. Burckhardt, M. Bhave, Z. Chen, P. H. Chen, X. Wang, G. Danuser and S. L. Schmid (2018). "A noncanonical role for dynamin-1 in regulating early stages of clathrin-mediated endocytosis in non-neuronal cells." PLoS Biol **16**(4): e2005377.

Taylor, M. J., D. Perrais and C. J. Merrifield (2011). "A high precision survey of the molecular dynamics of mammalian clathrin-mediated endocytosis." PLoS biology **9**(3): e1000604.

Wallis, T. P., A. Jiang, K. Young, H. Hou, K. Kudo, A. J. McCann, N. Durisic, M. Joensuu, D. Oelz, H. Nguyen, R. S. Gormal and F. A. Meunier (2023). "Super-resolved trajectory-derived nanoclustering analysis using spatiotemporal indexing." Nat Commun **14**(1): 3353.

Xue, J., M. E. Graham, A. E. Novelle, N. Sue, N. Gray, M. A. McNiven, K. J. Smillie, M. A. Cousin and P. J. Robinson (2011). "Calcineurin selectively docks with the dynamin Ixb splice variant to regulate activity-dependent bulk endocytosis." Journal of biological chemistry **286**(35): 30295-30303.

REVIEWER COMMENTS

Reviewer #1 (Remarks to the Author):

In this revised version, Jiang et al. have made significant efforts to address all the concerns raised by reviewers. A major challenge in the original manuscript was distinguishing between “Lateral diffusion model” and the traditional model from the dataset. The authors have now beautifully addressed this issue by measuring frame-to-frame displacement from the cluster's exterior to its interior. The current data is compelling in the claim of Lateral diffusion model. While there are still a few confusing aspects, as below, the revised manuscript has almost reached a level of completeness that justifies publication in Nature communications.

1. masks of nanoclusters within axon.

In Figure 5d,e, the authors created masks to measure the displacement of dynamin molecules both inside and outside the cluster. However, the relative size of these masks compared to the bouton is unclear. This detail is significant since many proteins exhibit a reduced diffusion coefficient upon entering the bouton from the axon, likely due to the crowded environment of presynapses such as synaptic vesicle clusters (Reshetniak et al. PMID 32627850). If the area of the mask covers large portion of the bouton, the observed shorter displacement within the masks could be attributed to this crowded environment. To provide clarity on this issue, the author should overlay the outline of the processes on the top panels of Figure 5d. Additionally, for greater data transparency and to aid reader's understanding, it would be beneficial to include more example traces showing both the mask and the outline of the processes in an extended figure. These changes should be feasible using the existing data set.

2. Image of Syt1-pHluorin.

The authors mention performing Syt1-pHluorin imaging to identify presynapses. While this experiment is clearly described in the text, Syt1 localizations are not included in the figures. Therefore, the corresponding Syt1-pHluorin images should be added alongside the low-resolution image and the enlarged tracking panel in Figure 5a, b, c.

3. Distinguish signals from Syt1-pHluorin and mEos2.

The emissions from pHluorin and unconverted mEos2 have nearly identical detection ranges. In addition, the fluorescence intensity of mEos2 looks consistently high across the processes. Consequently, it is unclear how the authors identified presynapses. The similar concern was also raised by Reviewer #2 and the authors made some comments with reviewer's figures. It should be also explained for the Syt1-pHluorin images. To enhance clarity for readers, the authors should include this explanation both in the text and in an extended figure.

Reviewer #2 (Remarks to the Author):

The authors have done a commendable job of addressing my concerns. I am happy to recommend publication. I have only some minor concerns.

1. In Figure S3, the authors show that the rate of cumulative count reaches a plateau, more prominently after Ba²⁺ stimulation. This strongly indicates that the 3D molecular recruitment was reduced at that phase. However, as one can imagine, this readout is a more average parameter and does not rule out variability between HFS and LFS zones. In other words, there could still be increased 3D recruitment in certain zones while the cell-wide average goes down. Can we confidently rule out that whatever has been shown through spatiotemporal 2D BOOSH analysis in Figures 4 and 5 is also not happening on the Z dimension? Because we know that 2D tracking has projection artifact issues and that some types of 3D motions get masked in 2D.

2. I am also intrigued by the answers provided by the authors on my minor concerns 3 and 4. The authors say neither Ba²⁺ nor K⁺ stimulation leads to ultrafast endocytosis. In addition, they also mention that the first phase of dynamic membrane recruitment is probably needed for bulk endocytosis, which is primarily relevant again during fast endocytosis, at least considering the time scale for their experiments. Given that we can't be sure if the clusters represent endocytic sites, I need help understanding how such a mechanism would benefit the cell. But that may be a matter of more considerable discussion in the field.

Reviewer #3 (Remarks to the Author):

The MS is significantly improved. The suggested mechanism of bi-phasic recruitment of Dyn1 is fully supported by evidence. The concerns I had are addressed by the authors.

Here I only have a couple of minor suggestions:

1. Typos, e.g. line 174
2. Sequence of figures. E.g. in line 203, the authors jump from Fig2 to Fig3sup without going through Fig3 first. This breaks the flow quite a bit. Rearranging the figures or having a more sequential context is suggested.

Overall, the MS is better and qualified for publication in Nature Communications.

REVIEWER COMMENTS

Reviewer #1 (Remarks to the Author):

In this revised version, Jiang et al. have made significant efforts to address all the concerns raised by reviewers. A major challenge in the original manuscript was distinguishing between “Lateral diffusion model” and the traditional model from the dataset. The authors have now beautifully addressed this issue by measuring frame-to-frame displacement from the cluster's exterior to its interior. The current data is compelling in the claim of Lateral diffusion model. While there are still a few confusing aspects, as below, the revised manuscript has almost reached a level of completeness that justifies publication in Nature communications.

We thank the reviewer for these kind comments.

1. masks of nanoclusters within axon.

In Figure 5d,e, the authors created masks to measure the displacement of dynamin molecules both inside and outside the cluster. However, the relative size of these masks compared to the bouton is unclear. This detail is significant since many proteins exhibit a reduced diffusion coefficient upon entering the bouton from the axon, likely due to the crowded environment of presynapses such as synaptic vesicle clusters (Reshetniak et al. PMID 32627850). If the area of the mask covers large portion of the bouton, the observed shorter displacement within the masks could be attributed to this crowded environment. To provide clarity on this issue, the author should overlay the outline of the processes on the top panels of Figure 5d. Additionally, for greater data transparency and to aid reader's understanding, it would be beneficial to include more example traces showing both the mask and the outline of the processes in an extended figure. These changes should be feasible using the existing data set.

To address the reviewer's concern and provide further insight/transparency, we have included a supplementary figure in which both representative outlines of processes and single-molecule detections made within the process are plotted over the spatiotemporal nanoclusters and trajectories from figure 5d (we would like to note that there was not enough space in our figure 5 to include these axon outlines directly to the traces displayed without compromising size and readability). As suggested by the reviewer, we have also included additional example trajectories in this figure, providing further clarity to readers (see below).

BOOSH algorithm, which utilizes 3-dimensional (spatial and temporal) density-based spatial clustering of applications with noise (DBSCAN) identifies clusters of Dyn1-mEos2 trajectories in space and time (Wallis, T.P. et al. Nat Commun 14, 3353 (2023))

The spatiotemporal nanoclusters of dynamin sometimes occupy a relatively small proportion of the presynaptic bouton, (cluster radius was approximately 0.1 which is small in comparison to the known sizes of presynaptic boutons). The ratio between the size of a bouton and nanocluster cannot provide evidence for, nor against the role of synapse-wide molecular crowding in the drop in mobility observed as Dyn1-mEos2 molecules enter a dynamin cluster. This is because there likely exists

inhomogeneity in 'crowdedness' inside an individual bouton (e.g., the crowded SV cluster, does not occupy the entirety of the presynaptic bouton).

Molecular crowding likely plays a role in hindering the motion of, and clustering, molecules at endocytic sites. Our spatiotemporal algorithm simply identifies regions of molecular clustering in space and time. Sites whereby dynamin exhibits clustering (and whereby dynamin shows reduced mobility), are indeed likely to be regions where there is heightened molecular crowding because a high abundance of molecules must be spatially localized in small regions for successful endocytosis.

Although molecular crowding is an interesting topic in itself, we are limited in tools to study it and therefore believe that this is outside of the scope of the current study.

New Extended data Fig. 6:

Extended data Fig. 6

 = BOOSH-derived spatiotemporal nanocluster

 = first detection in trajectory

2. Image of Syt1-pHluorin.

The authors mention performing Syt1-pHluorin imaging to identify presynapses. While this experiment is clearly described in the text, Syt1 localizations are not included in the figures. Therefore, the corresponding Syt1-pHluorin images should be added alongside the low-resolution image and the enlarged tracking panel in Figure 5a, b, c.

The low-resolution images (green) are the Syt1-pHluorin images, we have now changed the heading above the figure to make it clearer. Note, fluorescence in the green low-resolution image is contributed from both Syt1-pHluorin and mEos2 (green), this has been discussed below.

3. Distinguish signals from Syt1-pHluorin and mEos2.

The emissions from pHluorin and unconverted mEos2 have nearly identical detection ranges. In addition, the fluorescence intensity of mEos2 looks consistently high across the processes. Consequently, it is unclear how the authors identified presynapses. The similar concern was also raised by Reviewer #2 and the authors made some comments with reviewer's figures. It should be also explained for the Syt1-pHluorin images. To enhance clarity for readers, the authors should include this explanation both in the text and in an extended figure.

Our experimental method involves drawing ROIs for putative presynaptic nerve terminal structures based on the morphology (i.e., small varicosities along the axonal shaft) on the green channel. Though the green-channel does indeed include fluorescence from mEos2 (green) and Syt1-pHluorin, the small varicosities along axonal shafts representing presynapses are still easily determinable. We are therefore highly certain that the structures we draw ROIs around are indeed presynapses, this is also strengthened by the fact that only well-transfected hippocampal neurons with bright green fluorescence signal within processes were selected for imaging. This approach has previously been utilized extensively to identify synapses along axons (Longfield et al., (2023), PMC10638352).

Reviewer #2 (Remarks to the Author):

The authors have done a commendable job of addressing my concerns. I am happy to recommendation publication. I have only some minor concerns.

We thank the reviewer for this kind assessment of our work.

1. In Figure S3, the authors show that the rate of cumulative count reaches a plateau, more prominently after Ba²⁺ stimulation. This strongly indicates that the 3D molecular recruitment was reduced at that phase. However, as one can imagine, this readout is a more average parameter and does not rule out variability between HFS and LFS zones. In other words, there could still be increased 3D recruitment in certain zones while the cell-wide average goes down. Can we confidently rule out that whatever has been shown through spatiotemporal 2D BOOSH analysis in Figures 4 and 5 is also not happening on the Z dimension? Because we know that 2D tracking has projection artifact issues and that some types of 3D motions get masked in 2D.

Firstly, we agree with the reviewer that the current recruitment-rate analysis does not dissociate the variability of recruitment between HFA and LFA regions. Regardless, our results throughout the manuscript do not disprove 3D-recruitment of the short-tail isoforms, nor are we trying to refute this. In fact, theoretically it is likely that there is increased 3D recruitment to certain zones (HFA zones). This is simply due to the increased probability for a 3-dimensionally moving cytosolic molecule to directly associate with a cluster in these microscale regions due to the sheer abundance and density of clusters. What our results throughout the manuscript highlight is that 2-dimensional recruitment is a mechanism of recruitment that significantly enhances clustering, especially by the short-tail isoforms.

Secondly, the reviewer is concerned that the lateral-trapping analyses could be detecting cytosolic proteins that move very close to (but not tethered to) the plasma membrane prior to 3-dimensionally associating with a cluster site. We can not be confident that 100% of trajectories we detected exhibiting this 'lateral trapping' motion type did indeed tether (directly to or indirectly to) the plasma membrane and then move laterally to be trapped. However, the probability for a freely diffusing cytosolic molecule, which move extremely rapidly (e.g., cytosolic GFP diffuses at $\sim 25 \mu\text{m}^2 / \text{s}$), to consistently exist within the volume of an evanescent field ($<200\text{nm}$ thickness) for more than a few frames would be an extremely rare event. To illustrate this, below is a trajectory from a computational simulation of a single particle diffusing at $25 \mu\text{m}^2 / \text{s}$ within a 3-dimensional space for 1 second (left); and on (right) are trajectories made when this particle came within a $0.2 \mu\text{m}$ thickness area at the bottom of the volume. The average length of the particle trajectories made in the evanescent field is 0.00007 seconds - this is a blip in our experimental exposure rate.

We have previously demonstrated that cytosolic proteins were almost entirely undetectable in our TIRF system. Here, the membrane-binding deficient mutant (PH-PLC δ^{R40L}), which localizes to the cytosol, results in very little number of trajectories detected (Gormal et al., (2020); PMC7720173).

This highlights that the trajectories we detect are highly likely to be those events in which the particles become associated with the PM (directly or indirectly), then move laterally while associated to the PM. Nonetheless 3D recruitment strategy and 2D recruitment strategies are not mutually exclusive; both occur, but the degree to which they occur differ between the dynamin isoforms studied.

2. I am also intrigued by the answers provided by the authors on my minor concerns 3 and 4. The authors say neither Ba²⁺ nor K⁺ stimulation leads to ultrafast endocytosis. In addition, they also mention that the first phase of dynamic membrane recruitment is probably needed for bulk endocytosis, which is primarily relevant again during fast endocytosis, at least considering the time scale for their experiments. Given that we can't be sure if the clusters represent endocytic sites, I need help understanding how such a mechanism would benefit the cell. But that may be a matter of more considerable discussion in the field.

We thank the reviewer for this interesting question.

Although we do demonstrate colocalization of Dyn1 and Tf-647 – this does not occur for all Dyn1 molecules present at the plasma membrane. This was observed for all isoforms studied. Due to the dynamic nature of endocytosis, one hypothesis is that TF-647-negative Dyn1 puncta could represent regions on the PM where future endocytic events will take place. Another hypothesis is that these sites represent 'stores' of Dyn1 on the plasma membrane, whereby upon initiation of endocytosis at a nearby site, Dyn1 is released from this cluster and is then recruited directly to the newly forming endocytic site – this would bypass the need for recruitment of Dyn1 from the cytosol, thus reducing its search time and as a result enhancing the efficiency of endocytosis.

We would like to reemphasize that we have demonstrated that Dyn1bb-mEos2 mobility is significantly lower in AP2 positive clusters, further demonstrating that a degree of Dyn1bb-mEos2 clustering occurs at putative endocytic sites.

Reviewer #3 (Remarks to the Author):

The MS is significantly improved. The suggested mechanism of bi-phasic recruitment of Dyn1 is fully supported by evidence. The concerns I had are addressed by the authors.

We thank the reviewer for the kind assessment of our work.

Here I only have a couple of minor suggestions:

1. Typos, e.g. line 174

To the best of our knowledge, we have corrected these typos.

2. Sequence of figures. E.g. in line 203, the authors jump from Fig2 to Fig3sup without going through Fig3 first. This breaks the flow quite a bit. Rearranging the figures or having a more sequential context is suggested.

We thank the reviewer for this suggestion. We strongly believe that the current sequential context of the figures provides the best logical flow of our result.

First, in Figure 2, we introduce the readers to underlying single-molecule imaging data. Figure 2 contains plots of number of trajectories observed during acquisition. Extended data Figure 3 further delves into the rates at which these trajectories were observed and provide more evidence for the biphasic recruitment strategy (providing further evidence for data shown in Figure 1).

Overall, the MS is better and qualified for publication in Nature Communications.

We are excited to hear that the reviewer finds that our revisions have enhanced the MS.

REVIEWERS' COMMENTS

Reviewer #1 (Remarks to the Author):

The authors sufficiently addressed the remaining concerns.

Reviewer #2 (Remarks to the Author):

In the manuscript titled "Dynamin1 long- and short-tail isoforms exploit distinct recruitment and spatial patterns to form endocytic nanoclusters", Jiang et al. have presented convincing evidence to show that during ultrafast or fast endocytosis, dynamin short and long-tail isoforms follow distinct recruitment patterns. The short-tailed isoforms get recruited in the plasma membrane and move laterally to form clusters of dynamin that are predicted to represent sites of endocytic initiation. However, substantial variability in these clusters needs to be addressed further. This sequential recruitment of dynamin to PM and lateral diffusion represents a form of information compression where dynamin can find the target endocytic locations at a much shorter timescale, matching the process's timescale.

This is an exciting way to analyze the phenomena and has opened up multiple questions to be followed in the future. Experiments were clearly thought out, and robust statistical analysis was performed to support conclusions. Overall, this study broadens the understanding of molecular and biophysical principles that drive the important phenomena of vesicle cycling in neurons.

The authors also answered my concerns very well, and I support the publication in the current version, as suggested by other referees.